# Convergence Bound and Critical Batch Size of Muon Optimizer

## Abstract

Muon, a recently proposed optimizer that leverages the inherent matrix structure of neural network parameters, has demonstrated strong empirical performance, indicating its potential as a successor to standard optimizers such as AdamW. This paper presents theoretical analysis to support its practical success. We provide convergence proofs for Muon across four practical settings, systematically examining its behavior with and without the inclusion of Nesterov momentum and weight decay. We then demonstrate that the addition of weight decay ensures almost-sure boundedness of the parameter and gradient norms—without relying on the commonly imposed bounded-gradient assumption—and clarify the interplay between the weight decay coefficient and the learning rate. Finally, we derive a lower bound on the critical batch size for Muon—the batch size that minimizes the stochastic first-order oracle (SFO) complexity of training. Because the resulting formula involves problem-dependent quantities that are not directly observable (gradient variance, target precision, effective rank), it does not predict the critical batch size in absolute terms; rather, it reveals how the hyperparameters $\beta$ (momentum) and $\lambda$ (weight decay) govern the qualitative scaling of this value. Our experiments validate these hyperparameter-dependent predictions across workloads including image classification and language modeling.

## 1 Introduction

Optimization algorithms are fundamental to the training of deep neural networks (DNNs). Since the introduction of stochastic gradient descent (SGD) (Robbins & Monro, 1951), numerous optimizers have been proposed to accelerate convergence. Among these, adaptive gradient algorithms such as Adam (Kingma & Ba, 2015) and its subsequent variant AdamW (Loshchilov & Hutter, 2019) have emerged as the de facto standard in modern deep learning, valued for their rapid convergence and robust performance across a wide range of tasks. A common characteristic of these widely used first-order optimizers is that they treat the weight parameters of neural networks, which are inherently matrices or higher-order tensors, as high-dimensional vectors. However, this vector-based perspective, while effective, disregards the underlying geometric and algebraic structure within the parameter matrices.

The recently proposed Muon optimizer (Jordan et al., 2024) introduces a distinct paradigm that departs from the conventional vector-based viewpoint. The core idea of Muon is to preserve and leverage the intrinsic matrix structure of the network parameters. Instead of using the gradient vector, Muon computes its search direction by orthogonalizing the gradient momentum matrix. Specifically, for a given momentum matrix $C_t \in \mathbb{R}^{m \times n}$, the search direction $O_t \in \mathbb{R}^{m \times n}$ is the orthogonal matrix that is closest in Frobenius norm, found by solving

$$O_t := \underset{O \in \{O \in \mathbb{R}^{m \times n}: O^\top O = I_n\}}{\operatorname{argmin}} \|O - C_t\|_{\mathrm{F}}, \tag{1}$$

where $I_n$ denotes a $n \times n$ identity matrix. As established by classic matrix theory, if the singular value decomposition (SVD) of $C_t$ is $U_t S_t V_t^\top$, the optimal search direction is simply $O_t = U_t V_t^\top$, where $U_t \in \mathbb{R}^{m \times r}, S_t \in \mathbb{R}^{r \times r}, V_t \in \mathbb{R}^{n \times r}$, and $r > 0$ is the rank of $C_t$. This process of gradient orthogonalization is

aimed at finding a search direction that is independent of the gradient's magnitude, potentially leading to more stable and effective training dynamics. While SVD provides an exact analytical solution, its high computational cost renders it impractical for the large matrices found in modern DNNs. As detailed by Bernstein (2025), the key computational innovation of Muon is the use of the Newton-Schulz iteration (Bernstein & Newhouse, 2024; Higham, 2008; Björck & Bowie, 1971; Kovarik, 1970), a classic and remarkably efficient numerical method, to approximate this orthogonalization. This iterative algorithm enables Muon to compute the search direction without performing an explicit SVD, making it a computationally feasible optimizer for large-scale applications. This elegant fusion of a novel optimization perspective with a powerful numerical technique positions Muon as a promising, theoretically grounded alternative to existing optimizers.

Several studies have reported the strong empirical performance of Muon. Jordan et al. (2024) showed that Muon outperforms Shampoo (Gupta et al., 2018) and SOAP (Vyas et al., 2025), both on a per-step and wall-clock basis. Liu et al. (2025b) showed that by dividing the parameters and updating them with SOAP and Muon, optimization performance equivalent to or better than SOAP can be achieved while substantially reducing memory usage. Liu et al. (2025a) demonstrated that Muon is effective for training large-scale LLMs and suggested it has the potential to replace AdamW as the standard optimizer. AI et al. (2025) showed that Muon expands AdamWs Pareto frontier on the compute-time plane, enlarging the practitioners flexibility in resource allocation. However, the theoretical understanding of Muon's convergence behavior remains underdeveloped, and a formal justification for its strong performance over AdamW is still lacking. This work aims to bridge this gap by providing a rigorous convergence analysis of Muon.

Furthermore, since the batch size is a critical hyperparameter for managing computational costs in large-scale training, we also consider Muon's critical batch size. This is defined as the batch size that minimizes the computational cost of training. In other words, the critical batch size is the point at which further increases in batch size yield diminishing returns in hardware throughput (i.e., the number of samples processed per unit time). By understanding and utilizing the critical batch size, one can maximize GPU utilization, thereby shortening training times and reducing overall computing costs. Therefore, to theoretically understand Muon and maximize its performance, analyzing its critical batch size is essential. Following previous studies, we aim to clarify the critical batch size of Muon.

Our main contributions are as follows:

- We present a convergence analysis for four variants of Muon—with and without Nesterov momentum and with and without weight decay (Theorems 3.1 and 3.2). We showed that the upper bound on the average expected gradient norm is as follows:

$$
\frac{1}{T} \sum_{t=0}^{T-1} \mathbb{E} \left[ \|\nabla f(W_t)\|_* \right] = \begin{cases} \mathcal{O}\left( \dfrac{1}{\eta T} + \sqrt{\dfrac{(1-\beta)r_1}{b}} + \bar{r}\eta \right) & \text{without weight decay,} \\[2ex] \mathcal{O}\left( \dfrac{1}{\eta T} + \sqrt{\dfrac{(1-\beta)r_1}{b}} + \hat{r}\eta \right) & \text{with weight decay,} \end{cases}
$$

  where $W_t \in \mathbb{R}^{m \times n}$, $\eta > 0$ is a learning rate, $\beta \in [0,1)$ is momentum, $b$ is the batch size, $r_1 \leq n$ is maximum rank of $C_t - \nabla f(W_t)$, $r_2 \leq n$ is maximum rank of $O_t$, $r_3 \leq n$ is maximum rank of $W_t$, $\bar{r} := \max\{r_1, r_2\}$, and $\hat{r} := \max\{r_1, r_2, r_3\}$. For detailed definitions, see Section 2 and Theorems 3.1 and 3.2. The variant combining both Nesterov momentum and weight decay is of particular interest as it mirrors common practical settings. Our analysis thus offers direct insights into Muon's real-world behavior.

- We prove that incorporating weight decay ensures almost-sure boundedness of both the parameter and gradient norms (Propositions 3.1 and 3.2) without assuming bounded gradients, and present experimental results to support this finding. We also show that for Muon to converge, the learning rate $\eta$ and weight decay coefficient $\lambda$ must satisfy $\eta \leq \frac{1}{\lambda}$, a condition supported by our experimental results (see Figure 1).

- We derive a lower bound on the critical batch size for the four Muon variants (Proposition 4.3). For example, for Muon with Nesterov momentum and weight decay we show

$$b^{\star}_{\text{Muon}} > \frac{9(1 - \beta)(1 + \sqrt{2}\beta)^2 r_1 \sigma^2}{(1 - \lambda)^2 \epsilon^2},$$

where $\beta \in (0, 1]$ is momentum, $\lambda > 0$ is the weight decay coefficient, $\sigma^2 > 0$ is the variance of the stochastic gradient, $r_1$ is the maximum rank of $C_t - \nabla f(W_t)$, and $\epsilon > 0$ is the stopping threshold. Because the formula contains problem-dependent quantities ($\sigma^2$, $r_1$, $\epsilon$) that are not directly controlled by the practitioner, it does not predict the absolute value of the critical batch size. Instead, it identifies the qualitative scaling law: how the hyperparameters $\beta$ and $\lambda$ govern $b^{\star}$, and how the ratio between variants depends only on $\beta$ and $\lambda$ (the unknown quantities cancel). Our experiments validate these hyperparameter-dependent predictions (see Figures 3–4).

## 2 Preliminaries

### 2.1 Notations and Definition

Let $\mathbb{N}$ be the set of nonnegative integers. For $p \in \mathbb{N} \setminus \{0\}$, define $[p] := \{1, 2, \ldots, p\}$. Let $\mathbb{R}^d$ be a $d$-dimensional Euclidean space. We use lowercase letters for scalars (e.g., $a \in \mathbb{R}$), bold lowercase letters for vectors (e.g., $\boldsymbol{a} \in \mathbb{R}^d$), and uppercase letters for matrices (e.g., $A \in \mathbb{R}^{m \times n}$). $\boldsymbol{a}^{\top} \in \mathbb{R}^{1 \times d}$ and $A^{\top} \in \mathbb{R}^{n \times m}$ denote the transposes of $\boldsymbol{a} \in \mathbb{R}^d$ and $A \in \mathbb{R}^{m \times n}$, respectively. For a square matrix $A = (a_{ij}) \in \mathbb{R}^{n \times n}$, the trace is defined as $\text{tr}(A) := \sum_{i=1}^n a_{ii}$. For all vectors $\boldsymbol{x}, \boldsymbol{y} \in \mathbb{R}^d$, the Euclidean inner product is defined as $\langle \boldsymbol{x}, \boldsymbol{y} \rangle_2 := \boldsymbol{x}^{\top} \boldsymbol{y}$ and the Euclidean norm is defined as $\|\boldsymbol{x}\|_2 := \sqrt{\langle \boldsymbol{x}, \boldsymbol{x} \rangle_2}$. For all matrices $A, B \in \mathbb{R}^{m \times n}$, the Frobenius inner product is defined as $\langle A, B \rangle_{\text{F}} := \text{tr}(A^{\top} B)$, the Frobenius norm defined as $\|A\|_{\text{F}} := \sqrt{\langle A, A \rangle_{\text{F}}}$, the operator norm defined as $\|A\|_{\text{op}} := \max_{\|\boldsymbol{x}\|_2 \leq 1} \|A\boldsymbol{x}\|_2$, and the nuclear norm defined as $\|A\|_* := \sum_{i=1}^{\min\{m,n\}} s_i(A)$, where $s_i(A)$ denotes the $i$-th singular value of $A$. The model is parameterized by a matrix $W \in \mathbb{R}^{m \times n}$ ($m \geq n$), which is optimized by minimizing the empirical loss function $f(W) := \frac{1}{N} \sum_{i \in [N]} f_i(W)$, where $N \in \mathbb{R}$ is the number of training samples and $f_i(W)$ denotes the loss associated with the $i$-th training sample $z_i$ ($i \in [N]$). We define $W^{\star} := \arg\min_{W \in \mathbb{R}^{m \times n}} f(W)$. Let $\xi$ be a random variable that does not depend on $W \in \mathbb{R}^{m \times n}$, and let $\mathbb{E}_{\xi}[X]$ denote the expectation with respect to $\xi$ of a random variable $X$. $\xi_{t,i}$ is a random variable generated from the $i$-th sampling at time $t$, and $\xi_{t,i}$ and $\xi_{t,j}$ are independent ($i \neq j$). $\boldsymbol{\xi}_t := (\xi_{t,1}, \xi_{t,2}, \ldots, \xi_{t,b})^{\top}$ is independent of sequence $(W_k)_{k=0}^t \subset \mathbb{R}^{m \times n}$ generated by Muon (Algorithm 1), where $b$ ($\leq N$) is the batch size. The independence of $\boldsymbol{\xi}_0, \boldsymbol{\xi}_1, \ldots$ allows us to define the total expectation $\mathbb{E}$ as $\mathbb{E} = \mathbb{E}_{\boldsymbol{\xi}_0} \mathbb{E}_{\boldsymbol{\xi}_1} \cdots \mathbb{E}_{\boldsymbol{\xi}_t}$. Let $\mathsf{G}_{\xi}(W)$ be the stochastic gradient of $f(\cdot)$ at $W \in \mathbb{R}^{m \times n}$. The mini-batch $\mathcal{S}_t$ consists of $b$ samples at time $t$, and the mini-batch stochastic gradient of $f(W_t)$ for $\mathcal{S}_t$ is defined as $\nabla f_{\mathcal{S}_t}(W_t) := \frac{1}{b} \sum_{i \in [b]} \mathsf{G}_{\xi_{t,i}}(W_t) = \frac{1}{b} \sum_{i \in \mathcal{S}_t} \nabla f_i(W_t)$.

Algorithm 1 presents the most common variant of Muon, which incorporates Nesterov momentum and weight decay. Our implementation of Nesterov momentum and decoupled weight decay (Loshchilov & Hutter, 2019) follows the original formulations of (Jordan et al., 2024). Specifically, the Nesterov step $C_t = \beta M_t + (1 - \beta) \nabla f_{\mathcal{S}_t}(W_t)$ is the "accelerated gradient" form of Nesterov momentum, in which the gradient is blended with the momentum buffer rather than computed at a look-ahead point, following the reparametrization of Sutskever et al. (2013) that is standard in modern deep learning frameworks, and adopted by Dozat (2016) for Adam. It is equivalent to the classical Nesterov form under a change of variables. Muon optimizers with both Nesterov momentum and weight decay are often used in practice (e.g., Jordan et al., 2024; AI

---

**Algorithm 1** Muon

**Require:** $\eta, \lambda > 0, \beta \in [0, 1), M_{-1} := \boldsymbol{0}, W_0 \in \mathbb{R}^{m \times n}$
    **for** $t = 0$ to $T - 1$ **do**
        $M_t := \beta M_{t-1} + (1 - \beta) \nabla f_{\mathcal{S}_t}(W_t)$
        **if** (Nesterov = True) **then**
            $C_t := \beta M_t + (1 - \beta) \nabla f_{\mathcal{S}_t}(W_t)$
        **else**
            $C_t := M_t$
        **end if**
        $O_t := \text{NewtonSchulz5}(C_t)$
        **if** (weight decay = True) **then**
            $W_{t+1} := (1 - \eta\lambda) W_t - \eta O_t$
        **else**
            $W_{t+1} := W_t - \eta O_t$
        **end if**
    **end for**
    **return** $W_T$

et al., 2025). NewtonSchulz5($\cdot$) receives $C_t$, sets $X_0 := C_t/\|C_t\|_{\mathrm{F}}$, performs the following iterations from $k = 0$ to 4, and returns $X_5$.

$$X_{k+1} := aX_k + b(X_k X_k^\top)X_k + c(X_k X_k^\top)^2 X_k, \tag{2}$$

where $a = 3.4445, b = -4.7750, c = 2.0315$. The sequence $(X_k)_{k\in\mathbb{N}}$ converges to $O_t$ defined as Eq. (1) (Bernstein & Newhouse, 2024; Higham, 2008). Our main theoretical analysis assumes that $O_t := $ NewtonSchulz5($C_t$) satisfies Eq. (1). Therefore, there may be a gap between the Muon we consider theoretically and its practical implementation. To quantitatively assess the impact of the approximation error introduced by the NewtonSchulz iteration on the convergence rate of Muon, we provide an analysis of Muon using the approximate solution obtained via the NewtonSchulz iteration in Section 3.3.

## 2.2 Assumptions

We make the following standard assumptions:

**Assumption 2.1.** *The function $f \colon \mathbb{R}^{m\times n} \to \mathbb{R}$ is $L$-smooth, i.e., for all $A, B \in \mathbb{R}^{m\times n}$,*

$$\|\nabla f(A) - \nabla f(B)\|_{\mathrm{F}} \leq L\|A - B\|_{\mathrm{F}}.$$

**Assumption 2.2.** (i) *For all $t$ and all $i$,*

$$\mathbb{E}_{\xi_{t,i}}\left[\mathsf{G}_{\xi_{t,i}}(W_t)\right] = \nabla f(W_t).$$

(ii) *There exists a nonnegative constant $\sigma^2$ such that, for all $t$ and all $i$,*

$$\mathbb{E}_{\xi_{t,i}}\left[\|\mathsf{G}_{\xi_{t,i}}(W_t) - \nabla f(W_t)\|_{\mathrm{F}}^2\right] \leq \sigma^2.$$

# 3 Analysis of Muon's convergence

## 3.1 Muon without weight decay

We now present a convergence analysis of Muon (Algorithm 1) without weight decay. The proofs of Theorem 3.1(i) and (ii) are in Appendix B.

**Theorem 3.1.** *Suppose Assumptions 2.1 and 2.2 hold. Then, for all $t \in \mathbb{N}$,*

(i) for Muon without Nesterov and without Weight Decay,

$$\frac{1}{T}\sum_{t=0}^{T-1}\mathbb{E}\left[\|\nabla f(W_t)\|_*\right] \leq \frac{f(W_0)}{\eta T} + \frac{8\sqrt{r_1}\Delta}{(1-\beta)T} + 2\sqrt{2(1-\beta)}\sqrt{\frac{r_1\sigma^2}{b}} + \frac{4L\sqrt{r_1 r_2}\eta}{1-\beta} + \frac{Lr_2\eta}{2}$$

$$= \mathcal{O}\left(\frac{1}{\eta T} + \sqrt{\frac{(1-\beta)r_1}{b}} + \bar{r}\eta\right),$$

(ii) for Muon with Nesterov and without Weight Decay,

$$\frac{1}{T}\sum_{t=0}^{T-1}\mathbb{E}\left[\|\nabla f(W_t)\|_*\right] \leq \frac{f(W_0)}{\eta T} + \frac{8\sqrt{r_1}\beta\Delta}{(1-\beta)T} + 2(1+\sqrt{2}\beta)\sqrt{1-\beta}\sqrt{\frac{r_1\sigma^2}{b}} + \frac{4L\beta\sqrt{r_1 r_2}\eta}{1-\beta} + \frac{Lr_2\eta}{2}$$

$$= \mathcal{O}\left(\frac{1}{\eta T} + \sqrt{\frac{(1-\beta)r_1}{b}} + \bar{r}\eta\right),$$

*where $\Delta := \|M_0 - \nabla f(W_0)\|_{\mathrm{F}}$, $r_1 := \sup_t \mathrm{rank}(C_t - \nabla f(W_t))$, $r_2 := \sup_t \mathrm{rank}(O_t)$, and $\bar{r} := \max\{r_1, r_2\}$.*

Theorem 3.1 shows that Muon, both with and without Nesterov momentum, converges to a neighborhood of a stationary point, and achieves same upper bounds on convergence. Note that some terms in the upper bound are slightly smaller when Nesterov momentum is used. The term $\Delta$ arises from the initialization $\boldsymbol{m}_{-1} := \boldsymbol{0}$ used in Algorithm 1, and its effect diminishes as the number of steps $T$ increases. From a theoretical perspective, initializing $\boldsymbol{m}_0 := \boldsymbol{0}$ or $\boldsymbol{m}_0 := \nabla f_{\mathcal{S}_0}(W_0)$ would yield a more refined upper bound for $\Delta$. However, note that the Muon implementation (Jordan et al., 2024) adopts $\boldsymbol{m}_{-1} := \boldsymbol{0}$.

**Interpretation of the convergence rate.** Theorem 3.1 suggests that setting the learning rate small and the momentum and batch size large results in a tighter bounds. For example, using $\eta = T^{-\frac{3}{4}}$ and $1 - \beta = T^{-\frac{1}{2}}$ yields $\frac{1}{T} \sum_{t=0}^{T-1} \mathbb{E}\left[\|\nabla f(W_t)\|_*\right] = \mathcal{O}\left(T^{-\frac{1}{4}}\right)$. This rate matches the standard rate for nonconvex SGD-type methods, although the convergence criteria are different. For example, under exactly the same assumptions as ours, Liu et al. (2020) show that SGD with momentum using a learning rate $\eta = T^{-\frac{1}{2}}$ achieves $\frac{1}{T} \sum_{t=0}^{T-1} \mathbb{E}\left[\|\nabla f(\boldsymbol{\theta}_t)\|_2\right] = \mathcal{O}\left(T^{-\frac{1}{4}}\right)$. Note that in the matrix context, the Euclidean norm of the vectorized parameters corresponds to the Frobenius norm. Since $\|\nabla f(W_t)\|_{\mathrm{F}} \leq \|\nabla f(W_t)\|_*$, a convergence guarantee in terms of the nuclear norm is stronger than that in terms of the Frobenius norm. Therefore, compared to conventional methods such as SGD with momentum, which effectively ignore the matrix structure through vectorization, Muon respects the matrix structure and is shown to converge under a stricter analytical criterion given by the nuclear norm. This suggests that Muon provides a more structure aware and potentially more informative convergence guarantee in practice.

**Interpretation of the role of rank $r_1$.** The distinguishing feature of Muon's bound lies in the *variance term* $\sqrt{(1-\beta)r_1/b}$, which depends on the rank $r_1 = \sup_t \mathrm{rank}(C_t - \nabla f(W_t))$ rather than the full dimension $mn$. In practical deep networks, the effective rank of gradient matrices is often much smaller than the full dimension (i.e., $r_1 \ll n$), particularly in over-parameterized regimes where gradients exhibit low-rank structure. Recent empirical evidence supports this: Ahn et al. (2025) showed that a rank-fraction of $1/4$ (i.e., $r_1 \approx n/4$) suffices to match full-rank Muon performance in LLMs up to 3B parameters, and that larger models tolerate even lower rank fractions. This suggests that Muon's bound may be tighter than bounds for vector-based optimizers whose variance terms scale with the full parameter count. To make this comparison concrete, Appendix I measures $r_1$ directly on a controlled FashionMNIST MLP and finds $r_1 \leq 329$ for the $2048 \times 784$ layer, compared to $d = mn \approx 1.6 \times 10^6$. Since the SGD-type variance term scales with $\sqrt{d/b}$ while the Muon bound scales with $\sqrt{r_1/b}$, this translates to a variance-term reduction factor of $\sqrt{d/r_1} \gtrsim 70 \times$ at fixed $b$; equivalently, matching Muon's variance contribution would require SGD to use a batch roughly $d/r_1 \gtrsim 4.9 \times 10^3$ times larger. We emphasize, however, that this is a property of the *upper bound*, not a proof of strict superiority; the actual convergence behavior depends on problem-specific constants that are difficult to compare across optimizers. The empirical advantage of Muon over AdamW observed in our experiments (Figures 2–5) is consistent with this interpretation but may also arise from other mechanisms (e.g., implicit regularization from gradient orthogonalization) not captured by the worst-case bound.

## 3.2 Muon with weight decay

The following proposition establishes a key result for Muon (Algorithm 1) with weight decay. The proofs of Propositions 3.1 and 3.2 are in Appendix C.

**Proposition 3.1.** *Suppose Assumptions 2.1 and 2.2 hold, and that Muon is run with $\eta \leq \frac{1}{\lambda}$. Then, for all $t \in \mathbb{N}$,*

$$\|W_t\|_{\mathrm{F}} \leq \begin{cases} (1 - \eta\lambda)^t \|W_0\|_{\mathrm{F}} + \frac{\sqrt{r_2}}{\lambda} & \text{if } \eta < \frac{1}{\lambda}, \\ \frac{\sqrt{r_2}}{\lambda} & \text{if } \eta = \frac{1}{\lambda}, \end{cases}$$

*where $r_2 := \sup_t \mathrm{rank}(O_t)$.*

**Proposition 3.2.** *Suppose Assumptions 2.1 and 2.2 hold, and that Muon is run with $\eta \leq \frac{1}{\lambda}$. Then, for all $T \in \mathbb{N}$,*

$$\|\nabla f(W_t)\|_{\mathrm{F}} \leq \begin{cases} L(1-\eta\lambda)^t\|W_0\|_{\mathrm{F}} + \frac{L\sqrt{r_2}}{\lambda} + L\|W^\star\|_{\mathrm{F}} & \text{if } \eta < \frac{1}{\lambda}, \\ \frac{L\sqrt{r_2}}{\lambda} + L\|W^\star\|_{\mathrm{F}} & \text{if } \eta = \frac{1}{\lambda}, \end{cases}$$

*where $r_2 := \sup_t \mathrm{rank}(O_t)$.*

Proposition 3.1 establishes that when $\eta \leq \frac{1}{\lambda}$, weight decay ensures the parameter norm remains almost surely bounded. Furthermore, the upper bound decreases monotonically with $t$, converging to $\frac{\sqrt{r_2}}{\lambda}$ as $t \to \infty$. The bound is minimized uniformly across all $t$ when $\eta = \frac{1}{\lambda}$. Proposition 3.2 extends the result of Proposition 3.1 to the full gradient norm, which is likewise almost surely bounded. This bound decreases monotonically with $t$, converging to $\frac{L\sqrt{r_2}}{\lambda} + L\|W^\star\|_{\mathrm{F}}$ as $t \to \infty$. From these results, Corollary 3.1 establishes an almost surely bound of Muon. In both cases, the upper bounds are minimized when $\eta = \frac{1}{\lambda}$. Under He initialization, the expected operator norm $\|W_0\|_{\mathrm{op}}$ is approximately $\sqrt{2}$ for a square matrix, so the condition $\eta \leq 1/\lambda$ is easily satisfied for practical weight decay values ($\lambda \leq 0.2$ in all our experiments). This condition is a consequence of the proof technique (ensuring the iterate norms remain bounded); we conjecture it is not tight.

**Corollary 3.1.** *Suppose Assumptions 2.1 and 2.2 hold and $\eta \leq \frac{1}{\lambda}$. Then, for all $T \in \mathbb{N}$,*

$$\frac{1}{T}\sum_{t=0}^{T-1} \|\nabla f(W_t)\|_{\mathrm{F}} \leq \begin{cases} \frac{L\|W_0\|_{\mathrm{F}}}{\eta\lambda T} + \frac{L\sqrt{r_2}}{\lambda} + L\|W^\star\|_{\mathrm{F}} & \text{if } \eta < \frac{1}{\lambda}, \\ \frac{L\sqrt{r_2}}{\lambda} + L\|W^\star\|_{\mathrm{F}} & \text{if } \eta = \frac{1}{\lambda}, \end{cases}$$

*where $r_2 := \sup_t \mathrm{rank}(O_t)$.*

While these results suggest that a larger weight decay $\lambda$ yields a tighter bound, the condition $\eta \leq \frac{1}{\lambda}$ necessitates a smaller learning rate $\eta$. These desirable properties stem from the fact that Muon's search direction is inherently bounded. A key advantage of this feature is that our analysis does not rely on the common-and often restrictive-assumption of bounded gradients. We emphasize that the advantage of weight decay demonstrated by Propositions 3.1 and 3.2 is not a tighter asymptotic convergence rate: the rate $\mathcal{O}(1/(\eta T) + \sqrt{(1-\beta)r_1/b} + \hat{r}\eta)$ is the same with or without weight decay (compare Theorems 3.1 and 3.2), and the explicit $1/(1-\lambda)$ prefactor appearing in Theorem 3.2 in fact *slightly loosens* the upper bound. The practical benefit of weight decay is instead the almost-sure boundedness of $\|W_t\|_{\mathrm{F}}$ and $\|\nabla f(W_t)\|_{\mathrm{F}}$ established above, which removes the need for the common (and often restrictive) bounded-gradient assumption and provides implicit regularization during training.

The following is a convergence analysis of Muon (Algorithm 1) with weight decay (the proofs of Theorems 3.2(i) and (ii) are in Appendix C).

**Theorem 3.2.** *Suppose Assumptions 2.1 and 2.2 hold, and that Muon is run with $\eta \leq \frac{1}{\lambda}$. If $\lambda < \min\left\{\frac{1}{\|W_0\|_{\mathrm{op}}}, 1\right\}$, then, for all $T \in \mathbb{N}$,*

(i) for Muon without Nesterov and with Weight Decay,

$$\frac{1}{T}\sum_{t=0}^{T-1} \mathbb{E}\left[\|\nabla f(W_t)\|_*\right] \leq \frac{f(W_0)}{(1-\lambda)\eta T} + \frac{8\sqrt{r_1}\Delta}{(1-\beta)T} + \frac{2\sqrt{2(1-\beta)}}{1-\lambda}\sqrt{\frac{r_1\sigma^2}{b}} + \frac{4L\sqrt{r_1 r_2}\eta}{(1-\beta)(1-\lambda)} + \frac{(r_2 + \lambda^2 r_3)L\eta}{1-\lambda}$$

$$= \mathcal{O}\left(\frac{1}{\eta T} + \sqrt{\frac{(1-\beta)r_1}{b}} + \hat{r}\eta\right),$$

(ii) for Muon with Nesterov and with Weight Decay,

$$\frac{1}{T}\sum_{t=0}^{T-1} \mathbb{E}\left[\|\nabla f(W_t)\|_*\right] \leq \frac{f(W_0)}{(1-\lambda)\eta T} + \frac{8\beta\sqrt{r_1}\Delta}{(1-\beta)T} + \frac{2(1+\sqrt{2}\beta)\sqrt{1-\beta}}{1-\lambda}\sqrt{\frac{r_1\sigma^2}{b}} + \frac{4L\beta\sqrt{r_1 r_2}\eta}{(1-\beta)(1-\lambda)} + \frac{(r_2 + \lambda^2 r_3)L\eta}{1-\lambda}$$

$$= \mathcal{O}\left(\frac{1}{\eta T} + \sqrt{\frac{(1-\beta)r_1}{b}} + \hat{r}\eta\right),$$

*where* $\Delta := \|M_0 - \nabla f(W_0)\|_F$, $r_1 := \sup_t \text{rank}(C_t - \nabla f(W_t))$, $r_2 := \sup_t \text{rank}(O_t)$, $r_3 := \sup_t \text{rank}(W_t)$, *and* $\hat{r} := \max\{r_1, r_2, r_3\}$.

Similar conclusions follow from Theorem 3.2, which again demonstrate a modest advantage from incorporating Nesterov momentum. These results build on Propositions 3.1 and 3.2 and therefore inherit the assumption that $\eta \leq \frac{1}{\lambda}$. In other words, for Muon with weight decay to attain the stated convergence rate, it must satisfy $\eta \leq \frac{1}{\lambda}$. Practically speaking, since the weight decay coefficient $\lambda$ is typically less than 1, this assumption is realistic and does not materially constrain the choice of learning rate. Furthermore, Theorem 3.2 requires that the weight decay coefficient satisfy $\lambda < \min\left\{\frac{1}{\|W_0\|_{op}}, 1\right\}$, which recommends using a small $\lambda$. In practice, $\lambda$ is often used with values much smaller than 1, and this assumption can also be considered realistic. For practical reference, under He initialization (He et al., 2015) the operator norm $\|W_0\|_{op}$ is $\mathcal{O}(1)$ for each layer: we measured $\|W_0\|_{op} \approx 0.76$ for 3×3 convolution layers and $\approx 1.41$ for 1×1 shortcuts in ResNet-18, giving $1/\max_\ell \|W_0^{(\ell)}\|_{op} \approx 0.71$. Hence the condition $\lambda < 1/\|W_0\|_{op}$ is satisfied for all weight decay values used in our experiments ($\lambda \leq 0.5$). Compared to Theorem 3.1, combining Muon and weight decay multiplies several terms in the upper bound by $\frac{1}{1-\lambda}$, which *slightly loosens* the rate bound rather than tightening it. When $\lambda$ is sufficiently small we have $\frac{1}{1-\lambda} \approx 1$, so the bound does not grow excessively large even with weight decay. Consistent with the discussion after Propositions 3.1–3.2, the faster empirical convergence of the weight-decay variant observed in Figure 2 should therefore be attributed to the almost-sure boundedness of $\|W_t\|_F$ and $\|\nabla f(W_t)\|_F$ (i.e., implicit regularization), not to a smaller leading constant in the convergence rate. As in Theorem 3.1, these bounds represent convergence to a neighborhood of a stationary point, suggesting that setting the learning rate small and the momentum and batch size large results in a smaller upper bound.

### 3.3 Muon with Newton-Schulz

In practice, the search direction $O_t$ in Muon is approximated via the Newton-Schulz iteration, whereas in the previous sections we have assumed that $O_t$ is computed exactly via SVD. In this section, we show that the approximation error of $O_t$ introduced by the Newton-Schulz iteration has a limited effect on the convergence of Muon. Let $O_t$ be the exact orthogonalization matrix obtained via SVD, and let $\tilde{O}_t^{(k)}$ be the approximate matrix after $k$ iterations of the Newton-Schulz iteration (2). According to Kim & hwan Oh (2026, Theorem 2), for all $k \in \mathbb{N}$, there exists $\delta_{(k)} > 0$, such that $\|\tilde{O}_t^{(k)} - O_t\|_{op} \leq \delta_{(k)} \leq c_0^{3^k}$, where $c_0 < 1$. Using this fact, the following proposition characterizes the effect of the approximation error introduced by the Newton-Schulz iteration on the convergence of Muon with Nesterov momentum and weight decay. We present the analysis for this variant, as it is the most involved, and omit the results for the other three variants. Note that similar results can be derived for the other variants as well. The proof of Proposition 3.3 is in Appendix D.

**Proposition 3.3.** *Suppose Assumptions 2.1 and 2.2 hold, and that Muon is run with $\eta \leq \frac{1}{\lambda}$. If $\lambda < \min\left\{\frac{1}{\|W_0\|_{op}}, 1 - \delta_{(k)}\right\}$, then, for all $T \in \mathbb{N}$, for Muon with Nesterov, Weight Decay and Newton-Schulz iteration,*

$$\frac{1}{T}\sum_{t=0}^{T-1}\mathbb{E}\left[\|\nabla f(W_t)\|_*\right] \leq \frac{f(W_0) - f(W_T)}{(1 - \lambda - \delta_{(k)})\eta T} + \frac{8\beta\sqrt{r_1}\|M_0 - \nabla f(W_0)\|_F}{(1-\beta)(1 - \lambda - \delta_{(k)})T} + \frac{2(1+\sqrt{2}\beta)\sqrt{1-\beta}}{1 - \lambda - \delta_{(k)}}\sqrt{\frac{r_1\sigma^2}{b}}$$

$$+ \frac{4L\beta\eta\sqrt{r_1 r_2}}{(1-\beta)(1 - \lambda - \delta_{(k)})} + \frac{(2r_2 + \lambda^2 r_3)L\eta}{1 - \lambda - \delta_{(k)}} + \frac{2r_2\delta_{(k)}^2 L\eta}{1 - \lambda - \delta_{(k)}}$$

$$= \mathcal{O}\left(\frac{1}{\eta T} + \sqrt{\frac{r_1(1-\beta)}{b}} + \eta\hat{r}\right).$$

*where* $\Delta := \|M_0 - \nabla f(W_0)\|_F$, $r_1 := \sup_t \text{rank}(C_t - \nabla f(W_t))$, $r_2 := \sup_t \text{rank}(O_t)$, $r_3 := \sup_t \text{rank}(W_t)$, *and* $\hat{r} := \max\{r_1, r_2, r_3\}$.

Comparing Proposition 3.3 with Theorem 3.2(ii), we observe that the Newton-Schulz approximation introduces an additional $\delta_{(k)}^2$-dependent term, while the existing coefficients are also mildly affected through the

factor $1 - \lambda - \delta_{(k)}$. Since $\delta_{(k)}$ decays exponentially with $k$, it is small for the fixed number of Newton-Schulz iterations used in practice, and its impact is therefore limited. In fact, according to Shen et al. (2025), the experimental performance of the SVD-based Muon is equivalent to that of the Newton-Schulz-based version, with the main difference being the higher computational cost of the SVD procedure. Our own experiments comparing $k=3$ and $k=5$ Newton-Schulz iterations across four network widths (Appendix, Table 8) also show that while $k=5$ reduces the approximation error, the test accuracy gain is at most +0.15 percentage points, confirming that the gap is negligible in practice.

## 4    Analysis of Muon's critical batch size

Our theoretical analysis characterizes the critical batch size as a function of the gradient variance $\sigma^2$ and optimization hyperparameters. While explicitly modeling the dependence of $\sigma^2$ on model width or depth is beyond the scope of this single-matrix analysis, our results establish the fundamental relationship $b^\star \propto \sigma^2$. This suggests that scaling behaviors observed in larger models are mediated through changes in their gradient noise properties. Indeed, recent large-scale empirical studies report that Muon remains efficient at increasingly large batch sizes for large language models (Liu et al., 2025a); our theory supports this observation, predicting that if larger models entail distinct gradient variance characteristics, the critical batch size will shift accordingly. With this motivation, we now formalize the notion of the critical batch size used in our analysis.

We next introduce the concept of the critical batch size, defined as the batch size that minimizes computational complexity. This complexity is measured in terms of the stochastic first-order oracle (SFO) complexity, which is the total number of stochastic gradient computations. Since the optimizer computes $b$ stochastic gradients per step, an optimizer that runs for $T$ steps with batch size $b$ incurs a total of $Tb$ SFO complexity. Empirically, for batch sizes up to a certain threshold $b^\star$ (the critical batch size), the number of training steps $T$ required to train a DNN scales inversely with $b$ (Shallue et al., 2019; Ma et al., 2018; McCandlish et al., 2018). Beyond $b^\star$, increasing the batch size yields diminishing returns in reducing $T$. The critical batch size is therefore the batch size that minimizes SFO complexity $Tb$. Prior work has shown that $b^\star$ depends on both the optimizer (Zhang et al., 2019) and dataset size (Zhang et al., 2025) and has established a theoretical framework for proving its existence and estimating its lower bound (Sato & Iiduka, 2023; Imaizumi & Iiduka, 2024). Concurrently, Naganuma et al. (2026) derived a non-Euclidean gradient noise scale tailored to spectral descent methods (including Muon) and proposed an adaptive batch-size schedule based on this noise scale. Their analysis characterizes the critical batch size of the spectral gradient step itself but does not consider the effects of Nesterov momentum or weight decay. In contrast, our work adopts the SFO-minimization framework and aims to clarify the relationship between the Muon hyperparameters $\beta$ and $\lambda$ and its critical batch sizes.

### 4.1    Relationship between batch size and number of steps needed for training

Suppose Assumptions 2.1 and 2.2 hold. Then, by Theorems 3.1 and 3.2, the following inequality holds:

$$\frac{1}{T} \sum_{t=0}^{T-1} \mathbb{E}\left[\|\nabla f(W_t)\|_{\mathrm{F}}\right] \leq \frac{X}{T} + \frac{Y}{\sqrt{b}} + Z,$$

where $X, Y, Z > 0$ are nonnegative constants. Let $\epsilon > 0$ be an arbitrarily fixed threshold.

$$\exists T, \exists b : \frac{X}{T} + \frac{Y}{\sqrt{b}} + Z < \epsilon, \tag{3}$$

The relationship between $b$ and the number of steps $T_b$ satisfying Eq. (3) is as follows:

**Proposition 4.1.** *Suppose Assumptions 2.1 and 2.2 hold and let Muon be the optimizer under consideration. Then, $T_b$ defined by*

$$T_b := \frac{X\sqrt{b}}{(\epsilon - Z)\sqrt{b} - Y} < T \ \ for \ \ \sqrt{b} > \frac{Y}{\epsilon - Z}, \tag{4}$$

*satisfies Eq. (3). In addition, the function $T_b$ defined by Eq. (4) is monotone decreasing and convex for $\sqrt{b} > \frac{Y}{\epsilon - Z}$.*

*Proof.* According to Eq.(4), Muon satisfies Eq.(3). For $\sqrt{b} > \frac{Y}{\epsilon - Z}$, we have

$$\frac{\mathrm{d}T_b}{\mathrm{d}b} = \frac{-XY}{2\sqrt{b}\left\{(\epsilon - Z)\sqrt{b} - Y\right\}^2} \leq 0, \quad \frac{\mathrm{d}^2 T_b}{\mathrm{d}b^2} = \frac{XY\left\{3(\epsilon - Z)\sqrt{b} - Y\right\}}{4b\sqrt{b}\left\{(\epsilon - Z)\sqrt{b} - Y\right\}^3} \geq 0.$$

Therefore, $T_b$ is monotone decreasing and convex for $\sqrt{b} > \frac{Y}{\epsilon - Z}$. This completes the proof. □

Proposition 4.1 implies that the required number of steps $T_b$ decreases as $\sqrt{b}$ increases. In fact, the reduction in the required number of steps achieved by increasing the batch size has a ceiling. The next section will show that a critical batch size exists that minimizes computational complexity.

### 4.2 Existence of a critical batch size

The critical batch size minimizes the computational complexity for training. Here, we use SFO complexity as a measure of computational complexity. Since the stochastic gradient is computed $b$ times per step, SFO complexity is defined as

$$T_b b = \frac{Xb\sqrt{b}}{(\epsilon - Z)\sqrt{b} - Y}. \tag{5}$$

The following theorem guarantees the existence of critical batch sizes that are global minimizers of $T_b b$ defined by Eq. (5).

**Proposition 4.2.** *Suppose that Assumptions 2.1 and 2.2 hold and consider Muon. Then, there exists*

$$b_{Muon}^{\star} > \frac{9Y^2}{4\epsilon^2} \tag{6}$$

*such that $b_{Muon}^{\star}$ minimizes SFO complexity $T_b b$.*

*Proof.* From Eq.(6), we have that, for $\sqrt{b} > \frac{Y}{\epsilon - Z}$,

$$\frac{\mathrm{d}T_b b}{\mathrm{d}b} = \frac{X\sqrt{b}\left\{2(\epsilon - Z)\sqrt{b} - 3Y\right\}}{2\left\{(\epsilon - Z)\sqrt{b} - Y\right\}^2}, \quad \frac{\mathrm{d}^2 T_b b}{\mathrm{d}b^2} = \frac{XY\left\{3Y - (\epsilon - Z)\sqrt{b}\right\}}{4\sqrt{b}\left\{(\epsilon - Z)\sqrt{b} - Y\right\}^3}.$$

Since $\frac{\mathrm{d}^2 T_b b}{\mathrm{d}b^2} > 0$ holds when $\sqrt{b} < \frac{3Y}{\epsilon - Z}$ and $\frac{\mathrm{d}^2 T_b b}{\mathrm{d}b^2} \leq 0$ holds when $\sqrt{b} \geq \frac{3Y}{\epsilon - Z}$, $T_b b$ is a convex function for $\frac{Y}{\epsilon - Z} < \sqrt{b} < \frac{3Y}{\epsilon - Z}$. In addition, from $\frac{\mathrm{d}T_b b}{\mathrm{d}b} = 0$, we find that $\sqrt{b_{\text{Muon}}^{\star}} = \frac{3Y}{2(\epsilon - Z)}$ is a global minimizer of $T_b b$. Therefore, we have

$$\sqrt{b_{\text{Muon}}^{\star}} > \frac{3Y}{2\epsilon}, \quad \text{i.e.,} \quad b_{\text{Muon}}^{\star} > \frac{9Y^2}{4\epsilon^2}$$

This completes the proof. □

In Proposition 4.2, we obtained a lower bound for the critical batch size using $\frac{3Y}{2(\epsilon - Z)} > \frac{3Y}{2\epsilon}$. We provide additional context regarding the tightness of this inequality. In any variant we are considering, the term $Z$ is expressed using the parameter dimension $n$ and the learning rate $\eta$. For example, in the case of Muon with Nesterov momentum and weight decay, $Z = \frac{4L\beta\sqrt{r_1 r_2}\eta}{(1-\beta)(1-\lambda)} + \frac{(r_2 + \lambda^2 r_3)L\eta}{1-\lambda} = \mathcal{O}(\hat{r}\eta)$. Therefore, by choosing a sufficiently small learning rate $\eta$, the term $Z$ also becomes sufficiently small, and this inequality can be considered tight. On the basis of Theorems 3.1 and 3.2 and Proposition 4.2, we derive the following proposition, which gives $b_{\text{Muon}}^{\star}$.

Table 1: Approximate lower bound of critical batch size $b^\star_{\text{Muon}}$ computed with $\beta = 0.95$ and $\lambda = 0.0625$.

| | w/o weight decay | w/ weight decay |
|---|---|---|
| w/o Nesterov | $\dfrac{18(1-\beta)r_1\sigma^2}{\epsilon^2} \approx 0.9 \times \dfrac{r_1\sigma^2}{\epsilon^2}$ | $\dfrac{18(1-\beta)r_1\sigma^2}{(1-\lambda)^2\epsilon^2} \approx 1.0 \times \dfrac{r_1\sigma^2}{\epsilon^2}$ |
| w/ Nesterov | $\dfrac{9(1-\beta)(1+\sqrt{2}\beta)^2 r_1\sigma^2}{\epsilon^2} \approx 2.47 \times \dfrac{r_1\sigma^2}{\epsilon^2}$ | $\dfrac{9(1-\beta)(1+\sqrt{2}\beta)^2 r_1\sigma^2}{(1-\lambda)^2\epsilon^2} \approx 2.81 \times \dfrac{r_1\sigma^2}{\epsilon^2}$ |

**Proposition 4.3.** *Suppose Assumptions 2.1 and 2.2 hold. Then, for a given precision $\epsilon$, the lower bound of the critical batch size for Muon is shown in Table 1.*

To connect Table 1 with the proxy $X/T + Y/\sqrt{b} + Z < \epsilon$, the correspondence for Muon with Nesterov and weight decay (Theorem 3.2(ii)) is $Y = \frac{2(1+\sqrt{2}\beta)\sqrt{1-\beta}}{1-\lambda}\sqrt{r_1\sigma^2}$ and $Z = \frac{4L\beta\sqrt{r_1 r_2}\eta}{(1-\beta)(1-\lambda)} + \frac{(r_2 + \lambda^2 r_3)L\eta}{1-\lambda}$. The critical batch size $b^\star > 9Y^2/(4\epsilon^2)$ then yields the formula in Table 1. The results in Table 1 indicate that, for a given accuracy $\epsilon$, combining Nesterov momentum significantly increases the critical batch size *lower bound* for Muon. In contrast, the increase in the critical batch size lower bound achieved by combining weight decay is modest when using sufficiently small values of $\lambda$. Furthermore, in all cases, $Y^2$ becomes smaller as $\beta \to 1$ (e.g., $Y^2 = (1-\beta)(1+\sqrt{2}\beta)^2 \to 0$), suggesting that the larger the momentum $\beta$, the smaller the critical batch size $b^\star_{\text{Muon}}$. We stress that the formulas in Table 1 are lower bounds derived from the convergence upper bound and contain the problem-dependent quantities $\sigma^2$, $\epsilon$, and $r_1$. Accordingly, they should be read as *qualitative scaling laws* that identify which hyperparameters govern $b^\star$ and how, rather than as numerically predictive estimates. When comparing two Muon variants on the same workload, the unknown quantities cancel and the ratio depends only on $\beta$ and $\lambda$ (see Section 5). Although Table 1 is instantiated at $\lambda = 0.0625$ for readability, the $1/(1-\lambda)^2$ factor produces a substantially larger shift for larger $\lambda$: the predicted CBS ratio between the Nesterov+weight-decay and Nesterov-only variants scales as $1/(1-\lambda)^2$, yielding $\approx 1.56$ at $\lambda = 0.2$ and $\approx 4.0$ at $\lambda = 0.5$. The widened-range ResNet-18/CIFAR-10 sweep in Appendix F.4 (Table 13; 66 runs, $b \in \{32, \ldots, 4096\}$) confirms this quantitatively: the empirical CBS shifts from $\approx 1024$ for the Nesterov-only variant to $\approx 4096$ at $\lambda = 0.5$, i.e., a ratio of $4096/1024 = 4.0$ that matches the predicted $1/(1-0.5)^2 = 4.0$ within the power-of-two grid. In the next section, we provide numerical experiments focusing on this relationship between the critical batch sizes and hyperparameters $\beta$ and $\lambda$ (See Figures 3–6).

**Novelty of the CBS analysis for Muon.** We emphasize that, to our knowledge, the CBS derivation for Muon is among the first to address three key aspects compared to existing CBS analyses for SGD-type optimizers (Sato & Iiduka, 2023; Imaizumi & Iiduka, 2024). First, our CBS lower bound formula suggests a *rank-dependent* scaling $b^\star \propto r_1$, where $r_1$ is the rank of the momentum error matrixand does not occur in SGD analysis Imaizumi & Iiduka (2024). This structural difference may stem from Muon's matrix-recognized update rule. Second, the momentum parameter $\beta$ appears explicitly in the CBS lower bound formula (e.g., $b^\star \propto (1-\beta)(1+\sqrt{2}\beta)^2$ for the Nesterov variant), suggesting how momentum tuning may affect the optimal batch size—a relationship not emphasized in prior CBS results. Third, under our current analysis, we provide a two-level validation framework that combines a controlled full-Muon toy experiment designed to match the theoretical assumptions (with proxy fit $R^2 \approx 0.914$) with practical-scale hybrid experiments across vision and language workloads, offering a comprehensive validation approach for a matrix-aware optimizer.

## 5 Numerical Experiments

We evaluate Muon on three workloads: (i) ResNet-18 on CIFAR-10, (ii) VGG-16 on CIFAR-100, and (iii) Llama3.1 (320M) on the C4 corpus. We first analyze convergence and critical batch size on CIFAR-10 with ResNet-18, then report languagemodeling results on C4. Results for VGG-16 on CIFAR-100 are summarized in Appendix H, where we observe the same qualitative trends.

The experiments are organized into three complementary tiers. First, a *controlled full-Muon setting* (Teacher-Student task, Appendix F) that exactly matches the theoretical assumptions serves as a direct, quantitative

test of the convergence proxy and CBS lower bound formulas. Second, *practical-scale vision experiments* (ResNet/CIFAR, VGG/CIFAR-100) test whether qualitative predictions (CBS ordering among variants, $\beta$–CBS monotonicity, and weight-decay effects) transfer to realistic architectures under the standard hybrid optimizer. Third, a *language-modeling workload* (Llama/C4) probes whether the key trends generalize to a substantially different domain, serving as a practical relevance check rather than a strict theory validation.

**Experimental Setup** For CIFAR-10 and CIFAR-100 experiments, unless otherwise stated, we tuned the learning rate by grid search at a base batch size of 512 and applied square-root scaling for Muon and AdamW; for Momentum SGD we tried both square-root and linear scaling. Each configuration was run five times with different seeds and we report mean and standard deviation. For C4 dataset [1] we trained Llama3.1 (320M) with sequence length 2048 and batch sizes from 64 to 8192. SFO complexity is measured as *steps × batch size*. Further details of the experimental protocol are in Appendix E. Details of the tuning protocol and compute fairness considerations are provided in Appendix E.

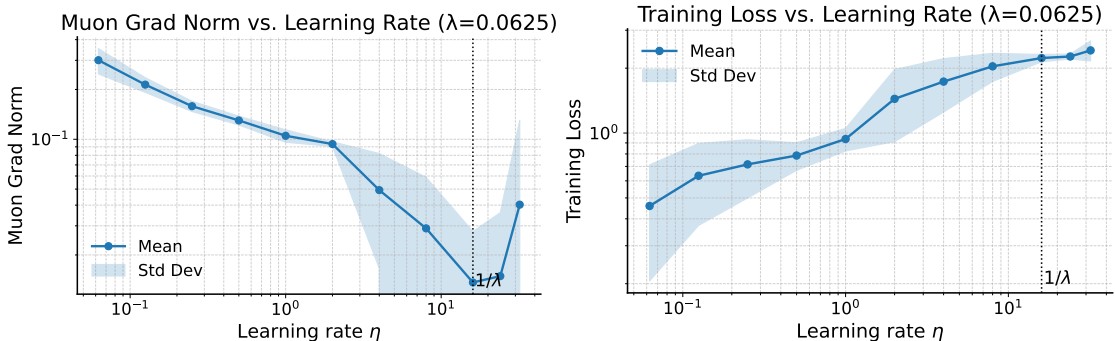

Figure 1: Empirical validation of the stability condition in Proposition 3.2. Final gradient norm (left) and training loss (right) for ResNet-18 on CIFAR-10 with Muon at $\lambda$=0.0625. The dashed line shows $\eta$=1/$\lambda$. Training is most stable near this value.

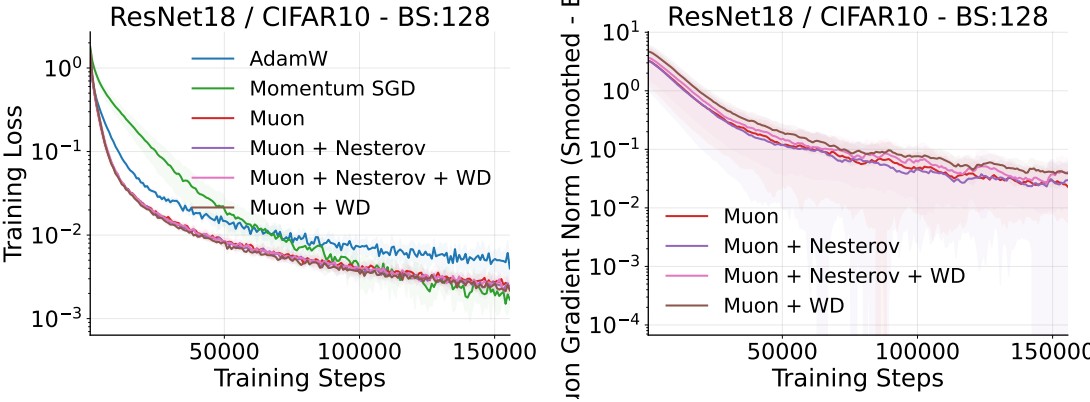

Figure 2: Convergence rate comparison for ResNet-18 on CIFAR-10 with batch size 128 trained for 400 epochs. Training loss (left) and smoothed gradient norm (right) over steps. Muon with Nesterov momentum and weight decay converges the fastest.

In the vision workloads, we follow common practice and use a hybrid optimizer (Muon on matrix-shaped parameter blocks and AdamW on the remaining parameters). This reflects the standard deployment of Muon: since the original release (Jordan et al., 2024), Muon has been applied exclusively to matrix-shaped (2D) parameters while a separate optimizer (typically AdamW) handles biases, normalization layers, and embedding vectors. We therefore adopt the same hybrid configuration in our practical-scale experiments to ensure our results are directly relevant to practitioners.

---

[1]https://huggingface.co/datasets/allenai/c4

**Two-level validation strategy: full Muon vs. hybrid Muon.** Our theory analyzes a single matrix parameter updated entirely by Muon, whereas practical implementations use a hybrid optimizer (Muon on matrix-shaped blocks, AdamW on the rest). To bridge this gap, we employ a two-level strategy: (i) a *controlled full-Muon setting* (Appendix F) that exactly matches the theoretical assumptions, confirming the convergence proxy $X/T + Y/\sqrt{b} + Z$ ($R^2 \approx 0.914$) and CBS scaling $b^\star \propto 1/\varepsilon^2$; and (ii) *practical-scale hybrid experiments* (ResNet/CIFAR, VGG/CIFAR, Llama/C4) showing that qualitative predictions (CBS ordering, $\beta$–CBS monotonicity) transfer to the hybrid regime. This transfer is expected because the Muon component dominates, gradient noise scales are highly correlated across layer types (Gray et al., 2024), and the CBS predictions depend only on hyperparameter ratios, not problem-specific constants.

**Theory-aligned stopping proxy: gradient-norm threshold.** Our theory upper-bounds the average expected full-gradient norm. To better align experiments with this proxy, we additionally report a gradient-normbased stopping metric. For each run, we track the Frobenius norm of the (mini-batch) gradient, $g_t := \|\nabla f_{\mathcal{S}_t}(W_t)\|_{\mathrm{F}}$, and define a smoothed estimate $\tilde{g}_t$ via an exponential moving average (EMA) over steps.[2] Given a target threshold $\epsilon$, we define the stopping time $T_\varepsilon(b)$ as the first step such that $\tilde{g}_t \leq \epsilon$, and report both steps $T_\varepsilon(b)$ and SFO complexity $b \cdot T_\varepsilon(b)$. We emphasize that our original loss/accuracy targets remain useful for practitioner-facing comparisons, while the gradient-norm criterion is introduced specifically to validate the theory-aligned proxy.

**Rationale for workload-specific stopping targets.** Our experiments use different stopping targets across workloads: 90% test accuracy and 95% training accuracy for ResNet-18/CIFAR-10 (Figure 3), the training loss for Llama3.1 (320M)/C4 (Figure 5), and the gradient-norm threshold $\varepsilon$ for the controlled full-Muon toy (Appendix F). These choices are not arbitrary but follow a single rule: *for each workload, we choose the most commonly reported metric in that domain, selecting a threshold within the regime where all compared optimizers reach the target so that SFO is well-defined.* For vision classification, test accuracy is the de facto benchmark metric and 90% on CIFAR-10 sits in the well-trained regime for ResNet-18 (near but below the $\approx 93\%$ ceiling), so every optimizer in our sweep reaches it; training accuracy at 95% additionally removes any generalization-gap confound. For language modeling, test accuracy is not standard; we therefore use training loss, which is the reported metric in the Muon-LLM literature (Jordan et al., 2024; Liu et al., 2025a; AI et al., 2025). For the controlled toy, the theoretical quantity itself is the full-gradient norm, so the $\varepsilon$-threshold criterion directly matches the theorem statement. To verify that our qualitative conclusions are not an artifact of any particular target, Appendix F.5 (Table 14) sweeps 13 different stopping criteria spanning test accuracy, training accuracy, training loss, and EMA gradient norm on a common workload, and shows that the SFO-minimizing batch size consistently falls in the range 32–256, with tighter thresholds shifting the CBS to larger values as $b^\star \propto 1/\varepsilon^2$ predicts. This robustness supports the workload-specific choices adopted in the main text.

**Convergence Analysis** We empirically validated the stability condition from Proposition 3.2. Figure 1 shows final gradient norm and training loss for ResNet-18 on CIFAR-10 across learning rates $\eta$ at fixed weight decay $\lambda=0.0625$. The vertical dashed line marks the threshold $\eta=1/\lambda=16.0$. The lowest gradient norm occurs near this threshold; for larger $\eta$ training becomes unstable. The same behavior holds for other values of $\lambda$ (Appendix H). This result validates a key practical implication of Propositions 3.1 and 3.2: the condition $\eta \leq 1/\lambda$ is not merely a theoretical convenience but a sharp stability boundary. Practitioners tuning Muon with weight decay should therefore set the learning rate at or below $1/\lambda$.

---

[2]

For large-scale workloads, computing the full gradient is impractical; we therefore use the mini-batch gradient norm as a practical surrogate for the full gradient and report EMA-smoothed curves for stability. We emphasize that while the stochastic gradient is an unbiased estimator of the true gradient, its *norm* is not an unbiased estimator of the true gradient norm; accordingly, we treat this quantity as a calibrated surrogate rather than an exact proxy. In our controlled full-Muon experiments (Appendix F), the Pearson correlation between the full-batch gradient norm and the mini-batch gradient norm exceeds 0.99 across all tested batch sizes and seeds (Table 10), confirming that the surrogate faithfully tracks the theoretical quantity. Theoretically, the CBS is the batch size that minimizes the computational cost required for the true gradient norm to reach a prescribed threshold. Empirically, however, the batch size that minimizes a given computational cost falls in the same regime whether measured via full-gradient norm, mini-batch gradient norm, or task-level metrics such as accuracy and loss (see Table 14). In a controlled toy setting (Appendix F), we also compute the full-batch gradient norm to directly match the theoretical quantity. The EMA window (100 steps) is chosen to balance noise suppression and responsiveness; preliminary tests with windows of 50 and 200 steps yielded CBS estimates within the same power-of-two grid point.

We next compared the four Muon variants with AdamW and Momentum SGD. Figure 2 shows that Muon with Nesterov momentum and weight decay attains the fastest decrease in both loss and gradient norm. From Theorems 3.1 and 3.2, the four variants of Muon exhibit almost identical convergence rates, making it difficult to clearly compare their convergence speeds based on the theory alone. On the other hand, Muon with weight decay consistently demonstrates better performance than its counterpart without weight decay, which can be explained by Propositions 3.1 and 3.2. In particular, weight decay guarantees the boundedness and monotonicity of both the parameter norm and the gradient norm, improving stability and thereby leading to better convergence behavior in practice.

**Critical Batch Size**  We measured the number of steps and SFO needed to reach 90% test accuracy and 95% training accuracy. Figure 3 shows that Muon scales better with batch size than the baselines. SFO is the lowest for Muon over the entire range, and Nesterov shifts the SFO-minimizing batch size to larger values. Momentum SGD requires more steps within the same schedule but eventually reaches comparable accuracy (Appendix H).

Table 1 predicts that, among the four Muon variants, adding Nesterov momentum increases the CBS coefficient more than adding weight decay alone. In Figure 3 (Right), the SFO-minimizing batch sizes of the four Muon variants are close to each other, with slight differences that are directionally consistent with the theoretical ordering. Across all tested batch sizes, Muon achieves fewer steps and lower SFO than AdamW. We note that the SFO curve is relatively flat near its minimum for all Muon variants, so several neighboring batch sizes yield near-optimal efficiency. In this flat regime, the small inter-variant CBS differences predicted by Table 1 are well within the plateau width, which is why the four curves largely overlap. A companion FashionMNIST sweep at batch sizes down to $b=4$ (Table 12 in Appendix F.3) quantifies this: at the optimal batch size the four Muon variants agree to within 4.6%, and across the plateau they lie within a $1.58\times$ band; the controlled full-Muon experiment, which eliminates the hybrid-optimizer confound, recovers a statistically significant ordering consistent with the theory. To demonstrate that the CBS differences become consequential when hyperparameters are varied more widely, we conducted an additional experiment on the same ResNet-18/CIFAR-10 setup sweeping Nesterov on/off, $\beta \in \{0.90, 0.95\}$, and $\lambda \in \{0, 0.2, 0.5\}$ across batch sizes up to 4096 (66 runs; details in Appendix F.4). In this setting the CBS shifts from $\approx 128$ (w/o Nesterov) to $\geq 4096$ (w/ Nesterov, $\lambda=0.5$), and the weight-decay CBS ratio between $\lambda=0.5$ and $\lambda=0$ equals 4.0, matching the predicted $1/(1-0.5)^2$ exactly.

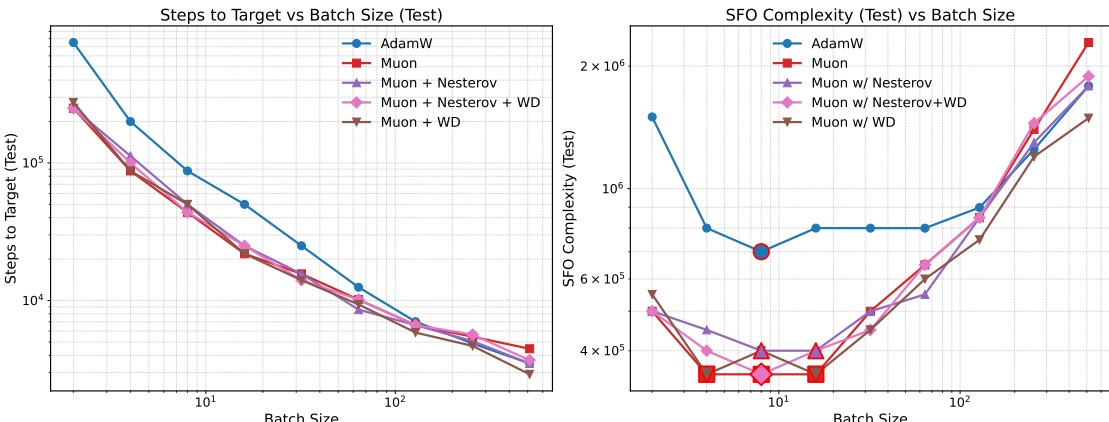

Figure 3: Batch-size scaling and SFO on ResNet-18/CIFAR-10. (Left) Steps to reach 90% test accuracy. (Right) SFO to reach 95% training accuracy. Muon achieves fewer steps and lower SFO than AdamW across all tested batch sizes.

**Role of rank $r_1$ and unknown quantities in the CBS lower bound formula.**  The CBS lower bound formulas in Table 1 contain several problem-dependent quantities that are not directly controlled by the practitioner: the gradient variance $\sigma^2$, the target precision $\epsilon$, and the maximum rank $r_1 := \sup_t \mathrm{rank}(C_t - \nabla f(W_t))$, where $r_1 \leq n$. Because the primary goal of our analysis is to clarify how the *hyperparameters $\beta$* and $\lambda$ govern the critical batch size, the quantities $\sigma^2$, $\epsilon$, and $r_1$ play the role of task-dependent constants

that cancel when comparing different Muon variants on the same workload. For instance, the ratio of CBS with Nesterov to CBS without Nesterov is $(1 + \sqrt{2}\beta)^2/2$, which depends only on $\beta$. Our experiments in Figures 3–6 are designed to test precisely these hyperparameter-dependent predictions and do not require knowledge of $r_1$ or $\sigma^2$. We note that recent work by Ahn et al. (2025) provides indirect evidence that $r_1 \ll n$ in practice: their Dion optimizer achieves near-full-rank Muon performance with only $r_1 \approx n/4$ in LLMs up to 3B parameters, suggesting that the low-rank structure assumed in our bound is realistic. We furthermore *directly* measure the rank and time-profile of $C_t - \nabla f(W_t)$ on a controlled FashionMNIST MLP in Appendix I: the hard rank saturates within $\approx 500$ steps at $\leq 0.42 \min(m, n)$ and does not exhibit unbounded growth with $T$, and the stable rank stays $\approx 1.15$–$1.25$, confirming that the worst-case $r_1 \leq \min(m, n)$ used in the proof is not attained in practice. The same appendix shows that the $\sqrt{(1 - \beta)r_1/b}$ variance term saturates quickly in $b$ while the accompanying $\mathcal{O}(\eta L \hat{r})$ floor can be suppressed by a diminishing step-size schedule, so large batches are not required for convergence. A detailed decomposition of how $r_1$, $\sigma^2$, and $\varepsilon$ individually affect the CBS is provided in Appendix G.

**Tightness of the CBS lower bound.** Proposition 4.2 provides a lower bound on the critical batch size via $b_{\mathrm{Muon}}^{\star} > 9Y^2/(4\epsilon^2)$, where the gap arises from bounding $\epsilon - Z$ by $\epsilon$. As noted in Section 4.2, $Z = \mathcal{O}(\hat{r}\eta)$ and becomes negligible for sufficiently small learning rates. In our controlled toy experiment (Appendix F), the fitted value of $Z$ is close to zero ($\hat{Z} \approx -0.017$, effectively negligible), and the simplified model without $Z$ still achieves $R^2 \approx 0.853$, empirically supporting that the lower bound is reasonably tight in practice.

**Width-varying CBS: empirical evidence for rank scaling.** To probe the rank dependence, we varied the ResNet-18 width multiplier $w \in \{0.125, \dots, 3.0\}$ on CIFAR-10. The effective rank increases monotonically with width, and the SFO complexity at the optimal batch size scales as SFO $\propto r_1^{-0.99}$, consistent with $b^{\star} \propto r_1$. Direct measurement of the gradient variance $\sigma^2$ shows that $r_1 \cdot \sigma^2$ is approximately constant across widths, explaining the flat CBS–rank scaling. Full results, including CBS estimates, gradient variance decomposition, and predicted vs. empirical CBS, are reported in Appendix G.

**Critical batch size under the gradient-norm stopping proxy (controlled full Muon).** As the first level of our validation strategy, we run a controlled experiment where Muon is applied to *all* parameters (full Muon) on a Teacher-Student Tanh Regression task (Appendix F). This setting exactly matches the theoretical assumptions: a single matrix parameter updated entirely by Muon, with no hybrid optimizer confound. We define the empirical SFO as $b \cdot T_\varepsilon(b)$ where $T_\varepsilon(b)$ is the first time the EMA-smoothed gradient norm drops below $\epsilon$. The EMA window (smoothing coefficient $\alpha = 0.01$, corresponding to an effective window of $\approx 100$ steps) was chosen to be long enough to suppress stochastic fluctuations while remaining shorter than the convergence timescale; varying $\alpha$ by a factor of two did not materially change the identified CBS. We observe a clear U-shaped SFO—batch curve and a well-defined minimizer $b^{\star}$ (e.g., $b^{\star} = 32$ for $\epsilon = 0.08$), providing evidence that a critical batch size exists when using the theory-aligned gradient-norm criterion.

**Critical batch size under the gradient-norm stopping proxy (controlled full Muon).** We further quantify how well the theoretical decomposition matches observations by fitting

$$\bar{g}(T, b) \approx \frac{X}{T} + \frac{Y}{\sqrt{b}} + Z$$

to the measured average gradient norms collected across multiple $(T, b)$ pairs in the controlled full-Muon MLP setting. A linear regression in the features $(1/T, 1/\sqrt{b}, 1)$ achieves a high goodness-of-fit (e.g., $R^2 = 0.914$), supporting that the proxy captures the dominant scaling with $T$ and $b$. From the fitted coefficients, we obtain $\hat{Y}$ and $\hat{Z}$ and compute a *predicted* critical batch size via

$$b_{\mathrm{pred}}^{\star} = \frac{9\hat{Y}^2}{4(\epsilon - \hat{Z})^2},$$

which matches the empirically SFO-minimizing batch size in the regime $\epsilon > \hat{Z}$. Moreover, when $\epsilon$ is not too close to the floor $\hat{Z}$, we observe the predicted scaling $b^{\star} \propto 1/(\epsilon - \hat{Z})^2$, consistent with the theoretical prediction from Proposition 4.2.

**Additional CBS validation.** We validated several additional predictions of the CBS theory. (i) Varying the target accuracy across four levels confirms $b^{\star} \propto 1/\varepsilon^2$ (log-log exponent 1.02, $R^2 = 0.984$). (ii) A three-optimizer comparison (Muon, AdamW, Shampoo) shows that Muon achieves the lowest SFO at every batch size (Appendix H.1).

(iii) Wall-clock measurements on a single A100 GPU confirm that Muon's per-step cost ($\approx 16$ ms) is moderate compared to AdamW ($\approx 8.5$ ms) and lower than Shampoo ($\approx 22$ ms; Appendix H.1). (iv) To verify that the CBS conclusions are robust to the choice of stopping criterion, we analyzed FashionMNIST (16 seeds) training logs under stopping criteria spanning four metric types: test accuracy, training accuracy, training loss, and EMA-smoothed gradient norm (Appendix F.5, Table 14). Across all criteria, the SFO-minimizing batch size falls in the range 32–256. The EMA gradient norm sweep yields the most direct evidence for the theoretical prediction: the CBS shifts monotonically as $32 \to 64 \to 128 \to 256$ (for $\varepsilon=0.60, 0.55, 0.40, 0.32$), supporting $b^\star \propto 1/\varepsilon^2$. (v) A controlled comparison of exact SVD vs. NS($k=3$) vs. NS($k=5$) orthogonalization in the full-Muon setting (10 seeds) confirms that all three methods converge to comparable gradient norms despite differing approximation errors, with NS error $< 0.04$ across all batch sizes (Appendix F.2, Table 11). (vi) An additional CBS experiment on ResNet-18/CIFAR-10 (66 runs, batch sizes 32–4096, 1–3 seeds) sweeping Nesterov on/off, $\beta \in \{0.90, 0.95\}$, and $\lambda \in \{0, 0.2, 0.5\}$ demonstrates clear CBS separation across configurations (Appendix F.4, Table 13). The CBS ordering, w/o Nesterov ($b^\star \approx 128$) < w/ Nesterov ($b^\star \approx 1024$) < w/ Nesterov, $\lambda=0.2$ ($b^\star \approx 2048$) < w/ Nesterov, $\lambda=0.5$ ($b^\star \geq 4096$), matches the direction predicted by Table 1, and the weight-decay CBS ratio between $\lambda=0.5$ and $\lambda=0$ equals 4.0, agreeing with the predicted $1/(1-0.5)^2$. This directly substantiates the boundedness/stability interpretation of the $1/(1-\lambda)^2$ factor predicted by the theory. In summary, the CBS range remains stable across stopping metrics (accuracy, loss, gradient norm), and tighter thresholds consistently shift the CBS to larger values. Both observations are consistent with the theoretical prediction $b^\star \propto 1/\varepsilon^2$.

**Effect of $\beta$ on Muon's Critical Batch Size**    Theory in Section 4.2 predicts that the critical batch size decreases as $\beta \to 1$. Figure 4 confirms this trend for ResNet-18/CIFAR-10, regardless of weight decay or Nesterov. Quantitatively, the CBS lower bound formulas in Table 1 contain the factor $(1-\beta)$ (or $(1-\beta)(1+\sqrt{2}\beta)^2$ with Nesterov), both of which vanish as $\beta \to 1$, driving the critical batch size toward zero. In Figure 4, the SFO-minimizing batch size shifts leftward as $\beta$ increases from 0.7 to 0.999, consistent with this monotone relationship. Practically, this suggests that high-momentum configurations can be trained efficiently at relatively small batch sizes, reducing the hardware resources required. A quantitative regression of *normalized* CBS $b^\star/(r_1\sigma^2)$ on the full-Muon Teacher-Student task confirms this monotone relationship ($R^2=0.90$; see Appendix F.7).

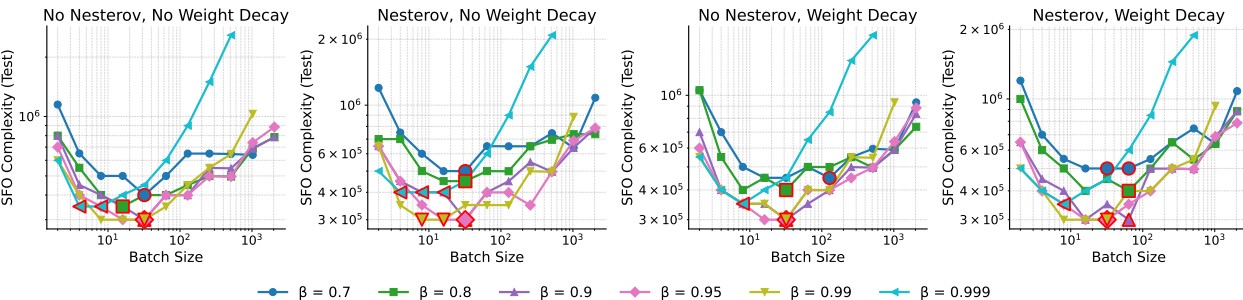

Figure 4: Dependence of SFO and critical batch size on $\beta$ for ResNet-18/CIFAR-10. The critical batch size consistently decreases as $\beta$ increases, in line with Section 4.2.

**Language-Model Workload: C4 on Llama3.1 (320M)**    We include a language-modeling workload as a *practical relevance check*: the goal is to verify that the main qualitative trends, namely Muon's SFO advantage over AdamW and the $\beta$–CBS relationship, generalize beyond vision, rather than to perform a strict quantitative validation of every theoretical constant. We train Llama3.1 (320M) (18 layers, hidden dimension 768, 12 attention heads; approximately 320M parameters) on the C4 corpus with batch sizes ranging from 32 to 4096 (details in Appendix E). Muon is applied to all matrix-shaped Transformer parameters (approximately 127M parameters), while the remaining parameters (embeddings, layer norms) are updated by AdamW. We define the stopping criterion as the number of steps to reach a target training loss, and report SFO complexity = steps × batch size. Figure 5 reports steps to reach the target loss and SFO versus batch size. Muon generally requires fewer steps and achieves lower SFO than AdamW, particularly at small to moderate batch sizes. At the largest batch sizes, the Muon variants show increasing SFO, narrowing the gap with AdamW. The inter-variant loss differences among the four Muon configurations are small (typically $< 0.05$), consistent with Theorems 3.1 and 3.2 predicting the same asymptotic rate for all variants; accordingly, the lack of consistent Nesterov/weight-decay gains in this setting is expected rather than a negative finding. Using the gradient-norm stopping criterion (which directly corresponds to the quantity bounded by the theorems), the SFO-minimizing batch size for Muon with Nesterov at $\beta=0.95$ shifts from 32 at loose thresholds ($\|\nabla f\| \leq 0.40$) to 128 at tighter

thresholds ($\|\nabla f\| \leq 0.25$), with a clear U-shaped SFO curve spanning GBS = 16 to 8192 at every threshold (Table 30 in Appendix J).

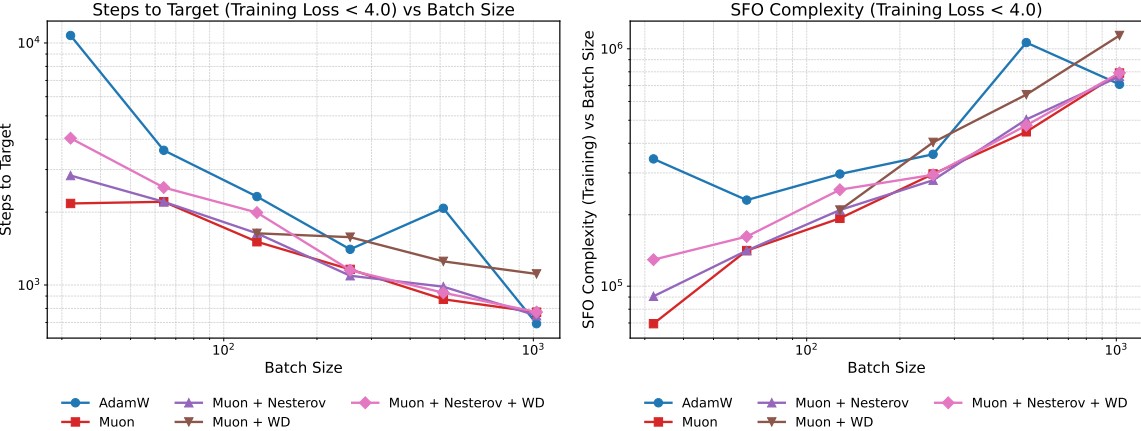

Figure 5: Batch-size scaling on `C4` with `Llama3.1 (320M)`. Steps to reach the target training loss (left) and SFO complexity (right) versus batch size. Muon generally achieves lower steps and SFO than AdamW at small to moderate batch sizes. Nesterov momentum and weight decay provide little additional benefit for this workload.

**Momentum Sweeps on `C4`** We varied $\beta$ on `C4` to examine the critical batch size. Figure 6 shows that a moderate value, $\beta \approx 0.95$, gives the best loss and SFO. As $\beta$ decreases, the critical batch size increases; as $\beta$ increases toward 1, the critical batch size decreases, but extreme values are suboptimal. These observations are consistent with Section 4.2 and mirror the vision results. The fact that the same $\beta$–CBS relationship holds across both vision (Figure 4) and language modeling (Figure 6) workloads supports the generality of the theoretical prediction, even though the two settings differ substantially in model architecture, loss landscape, and the ratio of Muon-updated to AdamW-updated parameters. Quantitatively, using the gradient-norm threshold $\|\nabla f\| \leq 0.20$, the SFO at $\beta{=}0.95$ is $1.30 \times 10^5$ (CBS =32), compared to $4.02 \times 10^5$ at $\beta{=}0.80$ (CBS =128) and $1.37 \times 10^6$ at $\beta{=}0.70$ (SFO nearly flat across batch sizes, indicating that the CBS exceeds the tested range). This confirms that $\beta{=}0.95$ offers the best efficiency. Values $\beta \geq 0.999$ fail to converge at most tested batch sizes.

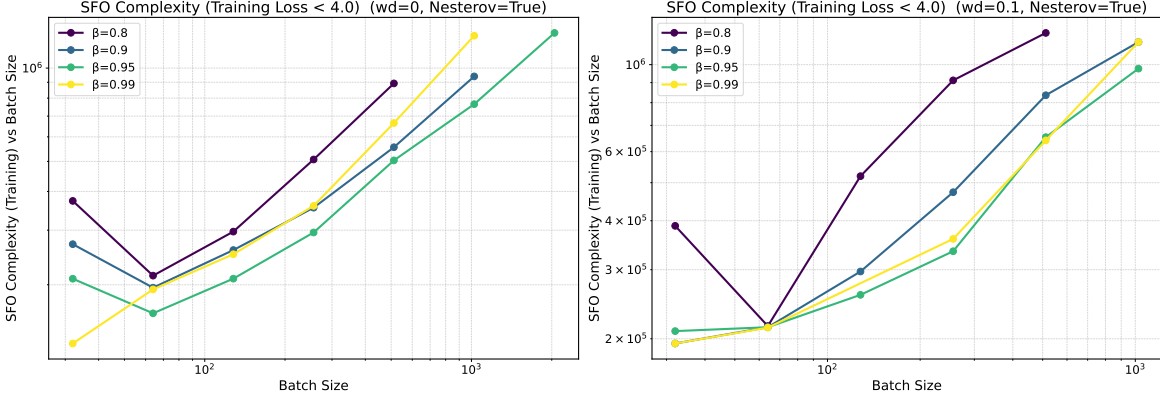

Figure 6: Effect of momentum $\beta$ on `C4` with `Llama3.1 (320M)`. SFO complexity across batch sizes for different $\beta$ values, without weight decay (left) and with weight decay (right). The best trade-off occurs near $\beta{=}0.95$. The critical batch size decreases with larger $\beta$.

# 6 Related Works

Several studies have investigated the theoretical properties and convergence behavior of Muon. Bernstein & Newhouse (2024) connected Muon to momentum applied in the steepest descent direction under a spectral norm constraint.

Li & Hong (2025) provided a pioneering analysis assuming Frobenius norm Lipschitz smoothness. Pethick et al. (2025) studied Muon in the context of optimization methods that use linear minimization oracles for norm balls, and established a convergence rate. Connections to other optimizers have also been explored. Shah et al. showed that Shampoo and SOAP reduce to Muon under simplifying assumptions (AI et al., 2025). Kovalev (2025) proposed and analyzed a stochastic non-Euclidean trust-region method that includes Muon as a special case. Similarly, An et al. (2025) proposed an adaptive structured gradient optimization algorithm that matches Muon in its momentum-free variant. Other studies have explored specific properties and extensions of Muon. Petrov et al. (2025) proposed and analyzed a zeroth-order version . Chen et al. demonstrated Muon's compatibility with the Lion-$\mathcal{K}$ algorithm (Chen et al., 2024) and showed that Muon with weight decay implicitly solves an optimization problem with a spectral norm constraint (Chen et al., 2025). Shen et al. (2025) presented a comprehensive analysis of Muon's convergence rate in comparison to gradient descent. Lau et al. (2025) introduced PolarGrad, a unifying framework for matrix-aware preconditioned methods including Muon, and established convergence rates. The core concepts of gradient orthogonalization and dualization, which are central to Muon, were introduced in the foundational works by Carlson et al. (2015) and Flynn (2017). On the practical side, Naganuma et al. (2026) derived a non-Euclidean gradient noise scale for spectral descent (including Muon) and used it to build an adaptive batch-size schedule, providing a complementary, empirically driven perspective on the critical batch size of Muon; however, their analysis does not account for the effects of momentum $\beta$ or weight decay $\lambda$. The convergence rates of major related works are summarized in Table 2. Our contribution is, to our knowledge, the first study to analyze the Muon gradient norm using both Nesterov momentum and weight decay.

The framework for assumptions and proofs presented by Shen et al. (2025) is the most similar to our own. The crucial difference between our study and all previous studies, including this one, is that we also consider most common variant of Muon, which incorporates Nesterov momentum and weight decay. Technically, the technique of transforming $\langle C_t, O_t \rangle_{\mathrm{F}}$ using the dual norm and inverse triangle inequality follows the proof of Pethick et al. (2025) (see proof of Theorem B.1 in Appendix B).

Table 2: Comparison of convergence rates in related works. $\| \cdot \|_\star$ denotes an arbitrary norm, and $\| \cdot \|_\ast$ denotes the nuclear norm. Each result has been rewritten to conform to our notation. $S(W_t)$ is the KKT score function defined as $S(W) := \|\nabla f(W)\|_\ast + \lambda\langle W, \nabla f(W)\rangle$. Let $r_4 := \max_{0 \leq t \leq T-1} \mathrm{rank}(\nabla f(W_t))$. Direct comparisons across rows should be interpreted with caution, as different works employ different smoothness assumptions and convergence measures.

| Related work | Measure | Convergence Rate | Nesterov momentum | Weight decay |
|---|---|---|---|---|
| Pethick et al. (2025) | $\mathbb{E}\left[\|\nabla f(W_T)\|_\star\right]$ | $\mathcal{O}\left(\frac{1}{\eta T} + \eta\right)$ | × | × |
| Li & Hong (2025) | $\frac{1}{T}\sum_{t=0}^{T-1} \mathbb{E}\left[\|\nabla f(W_t)\|_{\mathrm{F}}\right]$ | $\mathcal{O}\left(\frac{1}{\eta T} + \frac{1}{\sqrt{b}} + n\eta\right)$ | × | × |
| Kovalev (2025) | $\min_{0 \leq t \leq T-1} \mathbb{E}\left[\|\nabla f(W_t)\|_\ast\right]$ | $\mathcal{O}\left(\frac{1}{\eta T} + \eta + \sqrt{\beta}\right)$ | × | × |
| Shen et al. (2025) | $\frac{1}{T}\sum_{t=0}^{T-1} \mathbb{E}\left[\|\nabla f(W_t)\|_\ast\right]$ | $\mathcal{O}\left(\frac{1}{\eta T} + \frac{\sqrt{r_2}}{\sqrt{b}} + r_2\eta\right)$ | × | × |
| Chen et al. (2025) | $\frac{1}{T}\sum_{t=0}^{T-1} \mathbb{E}\left[S(W_t)\right]$ | $\mathcal{O}\left(\frac{1}{\eta T} + \frac{\sqrt{n}}{\sqrt{b}} + n\eta\right)$ | ✓ | ✓ |
| Lau et al. (2025) | $\frac{1}{T}\min_{0 \leq t \leq T-1} \mathbb{E}\left[\|\nabla f(W_t)\|_{\mathrm{F}}\right]$ | $\mathcal{O}\left(\frac{1}{\eta T} + \sqrt{r_4} + r_4\eta\right)$ | × | × |
| Ours (Theorem 3.2(ii)) | $\frac{1}{T}\sum_{t=0}^{T-1} \mathbb{E}\left[\|\nabla f(W_t)\|_\ast\right]$ | $\mathcal{O}\left(\frac{1}{\eta T} + \frac{\sqrt{r_1}}{\sqrt{b}} + \hat{r}\eta\right)$ | ✓ | ✓ |

# 7 Conclusion

Through a comprehensive theoretical analysis of the Muon optimizer, we established convergence guarantees for four practical configurations (with and without Nesterov momentum and with and without weight decay). Our primary theoretical contribution is demonstrating the crucial role of weight decay. We proved that it enforces a strict decrease in parameter and gradient norms, a clear advantage over the standard Muon configuration. This theoretical insight, along with the necessary condition relating the learning rate and weight decay coefficient, was empirically validated by our experimental results. Additionally, we derived the critical batch size for Muon, revealing its dependence on fundamental hyperparameters such as momentum and weight decay. Collectively, our findings provide both a deeper theoretical understanding of TMuon and actionable guidance for practitioners aiming to leverage this promising optimizer in large-scale settings.

Finally, we note three limitations. First, our single-matrix analysis does not directly model the layer-wise heterogeneity of gradient statistics in deep networks. However, Gray et al. (2024) show that gradient noise scales of different layers are highly correlated, suggesting that our theory captures the fundamental functional dependencies of the critical batch size on hyperparameters. Second, in practice Muon is used as a hybrid optimizer (Muon on matrix-shaped parameters, AdamW on the rest), whereas our theory assumes full Muon. We address this gap through a two-level validation strategy: a controlled full-Muon experiment that exactly matches the theoretical assumptions, and practical-scale hybrid experiments that confirm the qualitative predictions transfer to the hybrid regime (Section 5). Third, while Muon consistently outperforms Shampoo in our experiments (Appendix H.1), deriving analogous convergence bounds and CBS lower bound formulas for Shampoo and SOAP (Vyas et al., 2025) remains an open problem.

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

## A    Tools for Proof of All Theorems

The results presented in this section are not new and are given simply for reference and completeness.

**Lemma A.1.** *Suppose Assumption 2.2(ii) hold for all $t \in \mathbb{N}$; then,*

$$\mathbb{E}_{\boldsymbol{\xi}_t}\left[\|\nabla f_{\mathcal{S}_t}(W_t) - \nabla f(W_t)\|_{\mathrm{F}}^2\right] \leq \frac{\sigma^2}{b}.$$

*Proof.* Assumption 2.2(ii) guarantees that

$$
\begin{aligned}
\mathbb{E}_{\boldsymbol{\xi}_t}\left[\|\nabla f_{\mathcal{S}_t}(W_t) - \nabla f(W_t)\|_{\mathrm{F}}^2\right] &= \mathbb{E}_{\boldsymbol{\xi}_t}\left[\left\|\frac{1}{b}\sum_{i=1}^{b}\mathsf{G}_{\xi_{t,i}}(W_t) - \nabla f(W_t)\right\|_{\mathrm{F}}^2\right] \\
&= \mathbb{E}_{\boldsymbol{\xi}_t}\left[\left\|\frac{1}{b}\sum_{i=1}^{b}\mathsf{G}_{\xi_{t,i}}(W_t) - \frac{1}{b}\sum_{i=1}^{b}\nabla f(W_t)\right\|_{\mathrm{F}}^2\right] \\
&= \mathbb{E}_{\boldsymbol{\xi}_t}\left[\left\|\frac{1}{b}\sum_{i=1}^{b}\left(\mathsf{G}_{\xi_{t,i}}(W_t) - \nabla f(W_t)\right)\right\|_{\mathrm{F}}^2\right] \\
&= \frac{1}{b^2}\mathbb{E}_{\boldsymbol{\xi}_t}\left[\left\|\sum_{i=1}^{b}\left(\mathsf{G}_{\xi_{t,i}}(W_t) - \nabla f(W_t)\right)\right\|_{\mathrm{F}}^2\right] \\
&= \frac{1}{b^2}\mathbb{E}_{\boldsymbol{\xi}_t}\left[\sum_{i=1}^{b}\left\|\mathsf{G}_{\xi_{t,i}}(W_t) - \nabla f(W_t)\right\|_{\mathrm{F}}^2\right] \\
&\leq \frac{\sigma^2}{b}.
\end{aligned}
$$

This completes the proof. $\qquad\square$

The following lemma was established by (Mokhtari et al., 2020). In their setting, the algorithm without momentum corresponds to the case in which $\beta_1 = 1$, whereas in ours it corresponds to $\beta_1 = 0$. As a result, the statements may appear slightly different.

**Lemma A.2.** *Suppose Assumptions 2.1 and 2.2 hold. Then for all $t \in \mathbb{N}$,*

$$\sum_{t=0}^{T-1}\mathbb{E}\left[\|M_t - \nabla f(W_t)\|_{\mathrm{F}}\right] \leq \frac{4}{1-\beta}\|M_0 - \nabla f(W_0)\|_{\mathrm{F}} + \sqrt{\frac{2(1-\beta)\sigma^2}{b}}T + \frac{2L\eta\sqrt{r_2}}{1-\beta}T,$$

*where $r_2 := \sup_t \mathrm{rank}(O_t)$.*

*Proof.* From the definition of $M_t$,

$$
\begin{aligned}
\|M_t - \nabla f(W_t)\|_{\mathrm{F}}^2 &= \|\beta M_{t-1} + (1-\beta)\nabla f_{\mathcal{S}_t}(W_t) - \nabla f(W_t)\|_{\mathrm{F}}^2 \\
&= \|\beta(M_{t-1} - \nabla f(W_{t-1})) + \beta(\nabla f(W_{t-1}) - \nabla f(W_t)) + (1-\beta)(\nabla f_{\mathcal{S}_t}(W_t) - \nabla f(W_t))\|_{\mathrm{F}}^2 \\
&= \beta^2\|M_{t-1} - \nabla f(W_{t-1})\|_{\mathrm{F}}^2 + \beta^2\|\nabla f(W_{t-1}) - \nabla f(W_t)\|_{\mathrm{F}}^2 + (1-\beta)^2\|\nabla f_{\mathcal{S}_t}(W_t) - \nabla f(W_t)\|_{\mathrm{F}}^2 \\
&\quad + 2\beta^2\langle M_{t-1} - \nabla f(W_{t-1}), \nabla f(W_{t-1}) - \nabla f(W_t)\rangle_{\mathrm{F}} \\
&\quad + 2\beta(1-\beta)\langle M_{t-1} - \nabla f(W_{t-1}), \nabla f_{\mathcal{S}_t}(W_t) - \nabla f(W_t)\rangle_{\mathrm{F}} \\
&\quad + 2\beta(1-\beta)\langle \nabla f(W_{t-1}) - \nabla f(W_t), \nabla f_{\mathcal{S}_t}(W_t) - \nabla f(W_t)\rangle_{\mathrm{F}}
\end{aligned}
$$

Therefore, by taking the expectation,

$$
\begin{aligned}
\mathbb{E}\left[\|M_t - \nabla f(W_t)\|_{\mathrm{F}}^2\right] &= \beta^2\mathbb{E}\left[\|M_{t-1} - \nabla f(W_{t-1})\|_{\mathrm{F}}^2\right] + \beta^2\mathbb{E}\left[\|\nabla f(W_{t-1}) - \nabla f(W_t)\|_{\mathrm{F}}^2\right] \\
&\quad + (1-\beta)^2\mathbb{E}\left[\|\nabla f_{\mathcal{S}_t}(W_t) - \nabla f(W_t)\|_{\mathrm{F}}^2\right] + 2\beta^2\mathbb{E}\left[\langle M_{t-1} - \nabla f(W_{t-1}), \nabla f(W_{t-1}) - \nabla f(W_t)\rangle_{\mathrm{F}}\right].
\end{aligned}
$$

Here, by the Peter-Paul inequality, for all $\epsilon > 0$, we have

$$\langle M_{t-1} - \nabla f(W_{t-1}), \nabla f(W_{t-1}) - \nabla f(W_t)\rangle_{\mathrm{F}} \leq \frac{\epsilon}{2}\|M_{t-1} - \nabla f(W_{t-1})\|_{\mathrm{F}}^2 + \frac{1}{2\epsilon}\|\nabla f(W_{t-1}) - \nabla f(W_t)\|_{\mathrm{F}}^2.$$

Therefore, we obtain

$$\mathbb{E}\left[\|M_t - \nabla f(W_t)\|_{\mathrm{F}}^2\right] \leq \beta^2(1+\epsilon)\mathbb{E}\left[\|M_{t-1} - \nabla f(W_{t-1})\|_{\mathrm{F}}^2\right] + \beta^2\left(1 + \frac{1}{\epsilon}\right)\mathbb{E}\left[\|\nabla f(W_{t-1}) - \nabla f(W_t)\|_{\mathrm{F}}^2\right]$$
$$+ (1-\beta)^2\mathbb{E}\left[\|\nabla f_{\mathcal{S}_t}(W_t) - \nabla f(W_t)\|_{\mathrm{F}}^2\right].$$

In addition, from Assumption 2.1,

$$\|\nabla f(W_{t-1}) - \nabla f(W_t)\|_{\mathrm{F}}^2 \leq L^2\|W_{t-1} - W_t\|_{\mathrm{F}}^2 = L^2\eta^2\|O_t\|_{\mathrm{F}}^2 \leq L^2\eta^2\mathrm{rank}(O_t)\|O_t\|_{\mathrm{op}}^2 \leq L^2\eta^2 r_2,$$

where we use $r_2 := \sup_t \mathrm{rank}(O_t)$ and $\|O_t\|_{\mathrm{op}} = 1$ in the last inequality. Hence, from Lemma A.1,

$$\mathbb{E}\left[\|M_t - \nabla f(W_t)\|_{\mathrm{F}}^2\right] \leq \beta^2(1+\epsilon)\mathbb{E}\left[\|M_{t-1} - \nabla f(W_{t-1})\|_{\mathrm{F}}^2\right] + \beta^2\left(1 + \frac{1}{\epsilon}\right)L^2\eta^2 r_2 + \frac{(1-\beta)^2\sigma^2}{b}.$$

Then, letting $\epsilon := \frac{1-\beta}{2}$, we have

$$\begin{aligned}
\mathbb{E}\left[\|M_t - \nabla f(W_t)\|_{\mathrm{F}}^2\right] &\leq \frac{\beta^2(3-\beta)}{2}\mathbb{E}\left[\|M_{t-1} - \nabla f(W_{t-1})\|_{\mathrm{F}}^2\right] + \frac{\beta^2(3-\beta)}{1-\beta}L^2\eta^2 r_2 + \frac{(1-\beta)^2\sigma^2}{b} \\
&\leq \frac{1+\beta}{2}\mathbb{E}\left[\|M_{t-1} - \nabla f(W_{t-1})\|_{\mathrm{F}}^2\right] + \frac{2}{1-\beta}L^2\eta^2 r_2 + \frac{(1-\beta)^2\sigma^2}{b} \\
&\leq \left(\frac{1+\beta}{2}\right)^t\|M_0 - \nabla f(W_0)\|_{\mathrm{F}}^2 + \left\{\frac{2L^2\eta^2 r_2}{1-\beta} + \frac{(1-\beta)^2\sigma^2}{b}\right\}\sum_{k=0}^{t-1}\left(\frac{1+\beta}{2}\right)^k \\
&\leq \left(\frac{1+\beta}{2}\right)^t\|M_0 - \nabla f(W_0)\|_{\mathrm{F}}^2 + \left\{\frac{2L^2\eta^2 r_2}{1-\beta} + \frac{(1-\beta)^2\sigma^2}{b}\right\}\frac{2}{1-\beta} \\
&= \left(\frac{1+\beta}{2}\right)^t\|M_0 - \nabla f(W_0)\|_{\mathrm{F}}^2 + \frac{4L^2\eta^2 r_2}{(1-\beta)^2} + \frac{2(1-\beta)\sigma^2}{b}.
\end{aligned}$$

Therefore, summing over $t$, we have

$$\sum_{t=0}^{T-1}\mathbb{E}\left[\|M_t - \nabla f(W_t)\|_{\mathrm{F}}^2\right] \leq \frac{2}{1-\beta}\|M_0 - \nabla f(W_0)\|_{\mathrm{F}}^2 + \frac{2(1-\beta)\sigma^2}{b}T + \frac{4L^2\eta^2 r_2}{(1-\beta)^2}T.$$

Finally, from the properties of variance and expectation,

$$\begin{aligned}
\mathbb{E}\left[\|M_t - \nabla f(W_t)\|_{\mathrm{F}}\right] &\leq \sqrt{\mathbb{E}\left[\|M_t - \nabla f(W_t)\|_{\mathrm{F}}^2\right]} \\
&\leq \sqrt{\left(\frac{1+\beta}{2}\right)^t}\|M_0 - \nabla f(W_0)\|_{\mathrm{F}} + \sqrt{\frac{2(1-\beta)\sigma^2}{b}} + \frac{2L\eta\sqrt{r_2}}{(1-\beta)}.
\end{aligned}$$

Hence, we have

$$\begin{aligned}
\sum_{t=0}^{T-1}\mathbb{E}\left[\|M_t - \nabla f(W_t)\|_{\mathrm{F}}\right] &\leq \frac{\sqrt{2}}{\sqrt{2} - \sqrt{1+\beta}}\|M_0 - \nabla f(W_0)\|_{\mathrm{F}} + \sqrt{\frac{2(1-\beta)\sigma^2}{b}}T + \frac{2L\eta\sqrt{r_2}}{1-\beta}T \\
&\leq \frac{4}{1-\beta}\|M_0 - \nabla f(W_0)\|_{\mathrm{F}} + \sqrt{\frac{2(1-\beta)\sigma^2}{b}}T + \frac{2L\eta\sqrt{r_2}}{1-\beta}T
\end{aligned}$$

This completes the proof. □

## B  Proof of Theorems for Muon without weight decay

**Theorem B.1** (Auxiliary Theorem for Muon without weight decay). *Suppose Assumptions 2.1 and 2.2 hold. Then for all $t \in \mathbb{N}$,*

$$\sum_{t=0}^{T-1}\mathbb{E}\left[\|\nabla f(W_t)\|_*\right] \leq \frac{f(W_0) - f(W_T)}{\eta} + 2\sqrt{r_1}\sum_{t=0}^{T-1}\mathbb{E}\left[\|\nabla f(W_t) - C_t\|_{\mathrm{F}}\right] + \frac{L\eta}{2}r_2 T,$$

*where $r_1 := \sup_t \mathrm{rank}(C_t - \nabla f(W_t))$ and $r_2 := \sup_t \mathrm{rank}(O_t)$.*

*Proof.* From Assumption 2.1 and $\|O_t\|_{\mathrm{F}}^2 \le \mathrm{rank}(O_t)\|O_t\|_{\mathrm{op}}^2 \le r_2$,

$$f(W_{t+1}) \le f(W_t) + \langle \nabla f(W_t), W_{t+1} - W_t \rangle_{\mathrm{F}} + \frac{L}{2}\|W_{t+1} - W_t\|_{\mathrm{F}}^2$$

$$= f(W_t) - \eta \langle \nabla f(W_t), O_t \rangle_{\mathrm{F}} + \frac{L}{2}\eta^2\|O_t\|_{\mathrm{F}}^2$$

$$\le f(W_t) - \eta \langle C_t, O_t \rangle_{\mathrm{F}} - \eta \langle \nabla f(W_t) - C_t, O_t \rangle_{\mathrm{F}} + \frac{L\eta^2 r_2}{2}.$$

Using the definition $O_t := \underset{O \in \left\{ O \in \mathbb{R}^{m \times n} : O^\top O = I_n \right\}}{\mathrm{argmin}} \|O - C_t\|_{\mathrm{F}}$, we obtain $O_t := \underset{O \in \left\{ O \in \mathbb{R}^{m \times n} : O^\top O = I_n \right\}}{\mathrm{argmax}} \langle C_t, O \rangle_{\mathrm{F}}$. Then,

$$\langle C_t, O_t \rangle_{\mathrm{F}} = \max_{O : O^\top O = I_n} \langle C_t, O \rangle_{\mathrm{F}} = \max_{O : \|O\|_{\mathrm{op}} \le 1} \langle C_t, O \rangle_{\mathrm{F}} =: \|C_t\|_*,$$

where $\|\cdot\|_*$ denotes the dual norm. Applying the reverse triangle inequality and the relation $\|A\|_{\mathrm{F}} \le \|A\|_* \le \sqrt{\mathrm{rank}(A)}\|A\|_{\mathrm{F}}$, we have

$$-\langle C_t, O_t \rangle_{\mathrm{F}} = -\|C_t\|_*$$
$$= -\|C_t - \nabla f(W_t) + \nabla f(W_t)\|_*$$
$$\le \|C_t - \nabla f(W_t)\|_* - \|\nabla f(W_t)\|_*$$
$$\le \sqrt{\mathrm{rank}(C_t - \nabla f(W_t))}\|C_t - \nabla f(W_t)\|_{\mathrm{F}} - \|\nabla f(W_t)\|_*$$
$$\le \sqrt{r_1}\|C_t - \nabla f(W_t)\|_{\mathrm{F}} - \|\nabla f(W_t)\|_*, \tag{7}$$

where we use $r_1 := \sup_t \mathrm{rank}(C_t - \nabla f(W_t))$. In addition, since $O_t$ has orthonormal columns ($O_t^\top O_t = I_n$), we have $\|O_t\|_{\mathrm{op}} = 1$. Therefore, by the duality between the nuclear norm and the operator norm ($|\langle A, B \rangle_{\mathrm{F}}| \le \|A\|_*\|B\|_{\mathrm{op}}$),

$$|\langle \nabla f(W_t) - C_t, O_t \rangle_{\mathrm{F}}| \le \|\nabla f(W_t) - C_t\|_*\|O_t\|_{\mathrm{op}}$$
$$\le \sqrt{r_1}\|\nabla f(W_t) - C_t\|_{\mathrm{F}},$$

where we used $\|A\|_* \le \sqrt{\mathrm{rank}(A)}\|A\|_{\mathrm{F}}$. Therefore,

$$f(W_{t+1}) \le f(W_t) + 2\eta\sqrt{r_1}\|C_t - \nabla f(W_t)\|_{\mathrm{F}} - \eta\|\nabla f(W_t)\|_* + \frac{L\eta^2}{2}r_2.$$

By rearranging the terms and taking expectation, we obtain

$$\mathbb{E}\left[\|\nabla f(W_t)\|_*\right] \le \frac{f(W_t) - f(W_{t+1})}{\eta} + 2\sqrt{r_1}\mathbb{E}\left[\|\nabla f(W_t) - C_t\|_{\mathrm{F}}\right] + \frac{L\eta}{2}r_2.$$

This completes the proof. $\qquad\square$

## B.1 Proof of Theorem 3.1(i)

*Proof.* From $C_t := M_t$, together with Lemma A.2 and Theorem B.1, we find that

$$\sum_{t=0}^{T-1} \mathbb{E}\left[\|\nabla f(W_t)\|_*\right] \le \frac{f(W_0) - f(W_T)}{\eta} + 2\sqrt{r_1}\sum_{t=0}^{T-1} \mathbb{E}\left[\|\nabla f(W_t) - M_t\|_{\mathrm{F}}\right] + \frac{L\eta}{2}r_2 T$$

$$\le \frac{f(W_0) - f(W_T)}{\eta} + 2\sqrt{r_1}\left\{\frac{4}{1-\beta}\|M_0 - \nabla f(W_0)\|_{\mathrm{F}} + \sqrt{\frac{2(1-\beta)\sigma^2}{b}}T + \frac{2L\eta\sqrt{r_2}}{1-\beta}T\right\} + \frac{L\eta}{2}r_2 T.$$

By taking the average over $t = 0, \ldots, T-1$ and applying expectation, we obtain

$$\frac{1}{T}\sum_{t=0}^{T-1} \mathbb{E}\left[\|\nabla f(W_t)\|_*\right] \le \frac{f(W_0) - f(W_T)}{\eta T} + \frac{8\sqrt{r_1}\|M_0 - \nabla f(W_0)\|_{\mathrm{F}}}{(1-\beta)T} + 2\sqrt{\frac{2(1-\beta)r_1\sigma^2}{b}} + \frac{4L\eta\sqrt{r_1 r_2}}{1-\beta} + \frac{L\eta}{2}r_2$$

$$= \mathcal{O}\left(\frac{1}{\eta T} + \sqrt{\frac{r_1(1-\beta)}{b}} + \eta\bar{r}\right).$$

This completes the proof. $\qquad\square$

## B.2 Proof of Theorem 3.1(ii)

*Proof.* From the definition of $C_t$, we have

$$
\begin{aligned}
\|C_t - \nabla f(W_t)\|_{\mathrm{F}} &= \|\beta M_t + (1-\beta)\nabla f_{\mathcal{S}_t}(W_t) - \nabla f(W_t)\|_{\mathrm{F}} \\
&= \|\beta(M_t - \nabla f(W_t)) + (1-\beta)(\nabla f_{\mathcal{S}_t}(W_t) - \nabla f(W_t))\|_{\mathrm{F}} \\
&\leq \beta\|M_t - \nabla f(W_t)\|_{\mathrm{F}} + (1-\beta)\|\nabla f_{\mathcal{S}_t}(W_t) - \nabla f(W_t)\|_{\mathrm{F}}
\end{aligned}
$$

According to Theorem B.1, we find that

$$
\begin{aligned}
\sum_{t=0}^{T-1}\mathbb{E}\left[\|\nabla f(W_t)\|_*\right] &\leq \frac{f(W_0) - f(W_T)}{\eta} + 2\sqrt{r_1}\sum_{t=0}^{T-1}\mathbb{E}\left[\|C_t - \nabla f(W_t)\|_{\mathrm{F}}\right] + \frac{L\eta}{2}r_2 T \\
&\leq \frac{f(W_0) - f(W_T)}{\eta} + 2\sqrt{r_1}\left\{\beta\sum_{t=0}^{T-1}\mathbb{E}\left[\|M_t - \nabla f(W_t)\|_{\mathrm{F}}\right] + (1-\beta)\sum_{t=0}^{T-1}\mathbb{E}\left[\|\nabla f_{\mathcal{S}_t}(W_t) - \nabla f(W_t)\|_{\mathrm{F}}\right]\right\} \\
&\quad + \frac{L\eta}{2}r_2 T.
\end{aligned}
$$

From Lemmas A.1 and A.2, we have

$$
\begin{aligned}
\sum_{t=0}^{T-1}\mathbb{E}\left[\|\nabla f(W_t)\|_*\right] &\leq \frac{f(W_0) - f(W_T)}{\eta} + 2\beta\sqrt{r_1}\left\{\frac{4}{1-\beta}\|M_0 - \nabla f(W_0)\|_{\mathrm{F}} + \sqrt{\frac{2(1-\beta)\sigma^2}{b}}T + \frac{2L\eta\sqrt{n}}{1-\beta}T\right\} \\
&\quad + 2(1-\beta)\sqrt{r_1}\cdot\sqrt{\frac{\sigma^2}{b}}T + \frac{L\eta}{2}r_2 T.
\end{aligned}
$$

By taking the average over $t = 0, \ldots, T-1$ and applying expectation, we obtain

$$
\begin{aligned}
\frac{1}{T}\sum_{t=0}^{T-1}\mathbb{E}\left[\|\nabla f(W_t)\|_*\right] &\leq \frac{f(W_0) - f(W_T)}{\eta T} + \frac{8\beta\sqrt{r_1}\|M_0 - \nabla f(W_0)\|_{\mathrm{F}}}{(1-\beta)T} \\
&\quad + 2(1+\sqrt{2}\beta)\sqrt{1-\beta}\sqrt{\frac{r_1\sigma^2}{b}} + \frac{4L\eta\sqrt{r_1 r_2}\beta}{1-\beta} + \frac{L\eta}{2}r_2 \\
&= \mathcal{O}\left(\frac{1}{\eta T} + \beta\sqrt{\frac{r_1(1-\beta)}{b}} + \eta\bar{r}\right),
\end{aligned}
$$

where we use $2(\beta\sqrt{2(1-\beta)} + (1-\beta)) \leq 2(1+\sqrt{2}\beta)\sqrt{1-\beta}$. This completes the proof. $\qquad\square$

# C Proof of Theorems for Muon with weight decay

## C.1 Proof of Proposition 3.1

*Proof.* From the definition of $W_t$ and the condition $\eta \leq \frac{1}{\lambda}$, we have

$$
\begin{aligned}
\|W_t\|_{\mathrm{F}} &= \|(1-\eta\lambda)W_{t-1} - \eta O_t\|_{\mathrm{F}} \\
&\leq (1-\eta\lambda)\|W_{t-1}\|_{\mathrm{F}} + \eta\sqrt{r_2} \\
&\leq (1-\eta\lambda)^t\|W_0\|_{\mathrm{F}} + \eta\sqrt{r_2}\sum_{k=0}^{t-1}(1-\eta\lambda)^k \\
&\leq (1-\eta\lambda)^t\|W_0\|_{\mathrm{F}} + \frac{\sqrt{r_2}}{\lambda},
\end{aligned}
$$

where we use $\|O_t\|_{\mathrm{F}} \leq \sqrt{\mathrm{rank}(O_t)}\|O_t\|_{\mathrm{op}} \leq \sqrt{r_2}$ in the first inequality. This completes the proof. $\qquad\square$

## C.2 Proof of Proposition 3.2

*Proof.* According to Assumption 2.1,

$$\|\nabla f(W_t) - \nabla f(W^\star)\|_F \leq L\|W_t - W^\star\|_F.$$

Therefore, from Proposition 3.1 and the fact that $\nabla f(W^\star) = \mathbf{0}$, we have

$$\begin{aligned}
\|\nabla f(W_t)\|_F &\leq L\|W_t - W^\star\|_F \\
&\leq L\|W_t\|_F + L\|W^\star\|_F \\
&\leq L(1-\eta\lambda)^t\|W_0\|_F + \frac{L\sqrt{r_2}}{\lambda} + L\|W^\star\|_F.
\end{aligned}$$

This completes the proof. $\qquad\square$

**Lemma C.1.** *Suppose Assumptions 2.1 and 2.2 hold, and that Muon is run with $\eta \leq \frac{1}{\lambda}$. Then, for all $t \in \mathbb{N}$,*

$$\|W_t\|_{\mathrm{op}} \leq (1-\eta\lambda)^t\|W_0\|_{\mathrm{op}} + \frac{1-(1-\eta\lambda)^t}{\lambda}.$$

*In addition, if $\lambda < \frac{1}{\|W_0\|_{\mathrm{op}}}$ holds, then*

$$\|W_t\|_{\mathrm{op}} < 1.$$

*Proof.* From the definition of $W_t$ and $\|O_t\|_{\mathrm{op}} = 1$, we have

$$\begin{aligned}
\|W_t\|_{\mathrm{op}} &\leq (1-\eta\lambda)\|W_{t-1}\|_{\mathrm{op}} + \eta\|O_t\|_{\mathrm{op}} \\
&= (1-\eta\lambda)^t\|W_0\|_{\mathrm{op}} + \eta\sum_{j=0}^{t-1}(1-\eta\lambda)^j \\
&= (1-\eta\lambda)^t\|W_0\|_{\mathrm{op}} + \frac{1-(1-\eta\lambda)^t}{\lambda}.
\end{aligned}$$

In addition, if $\lambda < \frac{1}{\|W_0\|_{\mathrm{op}}}$, i.e., $\|W_0\|_{\mathrm{op}} < \frac{1}{\lambda}$,

$$\|W_t\|_{\mathrm{op}} < (1-\eta\lambda)^t\frac{1}{\lambda} + \frac{1-(1-\eta\lambda)^t}{\lambda} = 1.$$

This completes the proof. $\qquad\square$

**Theorem C.1** (Auxiliary Theorem for Muon with weight decay)**.** *Suppose Assumptions 2.1 and 2.2 hold, and that Muon is run with $\eta \leq \frac{1}{\lambda}$. If $\lambda < \min\left\{\frac{1}{\|W_0\|_{\mathrm{op}}}, 1\right\}$, then for all $t \in \mathbb{N}$,*

$$\sum_{t=0}^{T-1}\mathbb{E}\left[\|\nabla f(W_t)\|_*\right] \leq \frac{f(W_0) - f(W_T)}{(1-\lambda)\eta} + \frac{2\sqrt{r_1}}{1-\lambda}\sum_{t=0}^{T-1}\mathbb{E}\left[\|\nabla f(W_t) - C_t\|_F\right] + \frac{(r_2 + \lambda^2 r_3)L\eta T}{1-\lambda}$$

*where $r_1 := \sup_t \operatorname{rank}(C_t - \nabla f(W_t))$, $r_2 := \sup_t \operatorname{rank}(O_t)$, and $r_3 := \sup_t \operatorname{rank}(W_t)$.*

*Proof.* According to Assumption 2.1,

$$\begin{aligned}
f(W_{t+1}) &\leq f(W_t) + \langle\nabla f(W_t), W_{t+1} - W_t\rangle_F + \frac{L}{2}\|W_{t+1} - W_t\|_F^2 \\
&= f(W_t) - \eta\langle\nabla f(W_t), O_t + \lambda W_t\rangle_F + \frac{L\eta^2}{2}\|O_t + \lambda W_t\|_F^2 \\
&= f(W_t) - \eta\langle C_t, O_t + \lambda W_t\rangle_F - \eta\langle\nabla f(W_t) - C_t, O_t + \lambda W_t\rangle_F + \frac{L\eta^2}{2}\|O_t + \lambda W_t\|_F^2.
\end{aligned}$$

First, we bound the first inner product term. From Eq.(7) and Lemma C.1, we have

$$-\langle C_t, O_t + \lambda W_t\rangle_F = -\langle C_t, O_t\rangle_F - \lambda\langle C_t, W_t\rangle_F = -\|C_t\|_* + \lambda\|C_t\|_*\|W_t\|_{\mathrm{op}} \leq -(1-\lambda)\|C_t\|_*. \qquad (8)$$

From the reverse triangle inequality, we have $-\|C_t\|_* \leq \|C_t - \nabla f(W_t)\|_* - \|\nabla f(W_t)\|_*$. Hence,

$$-\eta\langle C_t, O_t + \lambda W_t\rangle_{\mathrm{F}} \leq (1-\lambda)\eta\|C_t - \nabla f(W_t)\|_* - (1-\lambda)\eta\|\nabla f(W_t)\|_*$$
$$\leq (1-\lambda)\sqrt{r_1}\eta\|C_t - \nabla f(W_t)\|_{\mathrm{F}} - (1-\lambda)\eta\|\nabla f(W_t)\|_*.$$

Next, we bound the second inner product term. From Lemma C.1 and $\|O_t\|_{\mathrm{op}} = 1$, we have

$$-\langle\nabla f(W_t) - C_t, O_t + \lambda W_t\rangle_{\mathrm{F}} = -\langle\nabla f(W_t) - C_t, O_t\rangle_{\mathrm{F}} - \lambda\langle\nabla f(W_t) - C_t, W_t\rangle_{\mathrm{F}}$$
$$\leq \|\nabla f(W_t) - C_t\|_*\|O_t\|_{\mathrm{op}} + \lambda\|\nabla f(W_t) - C_t\|_*\|W_t\|_{\mathrm{op}}$$
$$\leq \sqrt{r_1}\|\nabla f(W_t) - C_t\|_{\mathrm{F}} + \lambda\sqrt{r}\|\nabla f(W_t) - C_t\|_{\mathrm{F}}$$
$$= (1+\lambda)\sqrt{r_1}\|\nabla f(W_t) - C_t\|_{\mathrm{F}}. \tag{9}$$

Finally, we bound the last term. From Lemma C.1, we have $\|W_t\|_{\mathrm{F}} \leq \sqrt{r_3}\|W_t\|_{\mathrm{op}} \leq \sqrt{r_3}$, where $r_3 := \sup_t \mathrm{rank}(W_t)$. Hence,

$$\frac{L\eta^2}{2}\|O_t + \lambda W_t\|_{\mathrm{F}}^2 = L\eta^2\|O_t\|_{\mathrm{F}}^2 + L\eta^2\lambda^2\|W_t\|_{\mathrm{F}}^2 \leq L\eta^2 r_2 + L\eta^2\lambda^2 r_3 = (r_2 + \lambda^2 r_3)L\eta^2.$$

From all of the above, we obtain

$$f(W_{t+1}) \leq f(W_t) + (1-\lambda)\sqrt{r_1}\eta\|C_t - \nabla f(W_t)\|_{\mathrm{F}} - (1-\lambda)\eta\|\nabla f(W_t)\|_*$$
$$+ \eta(1+\lambda)\sqrt{r_1}\|\nabla f(W_t) - C_t\|_{\mathrm{F}} + (r_2 + \lambda^2 r_3)L\eta^2$$
$$\leq f(W_t) + 2\sqrt{r_1}\eta\|\nabla f(W_t) - C_t\|_{\mathrm{F}} - (1-\lambda)\eta\|\nabla f(W_t)\|_* + (r_2 + \lambda^2 r_3)L\eta^2.$$

By rearranging the terms and taking the expectation, we obtain

$$\mathbb{E}\left[\|\nabla f(W_t)\|_*\right] \leq \frac{f(W_t) - f(W_{t+1})}{(1-\lambda)\eta} + \frac{2\sqrt{r_1}}{1-\lambda}\mathbb{E}\left[\|\nabla f(W_t) - C_t\|_{\mathrm{F}}\right] + \frac{(r_2 + \lambda^2 r_3)L\eta}{1-\lambda}.$$

This completes the proof. $\qquad\square$

## C.3 Proof of Theorem 3.2(i)

*Proof.* From $C_t := M_t$ and Lemma A.2 and Theorem C.1, we find that

$$\sum_{t=0}^{T-1}\mathbb{E}\left[\|\nabla f(W_t)\|_*\right] \leq \frac{f(W_0) - f(W_T)}{(1-\lambda)\eta} + \frac{2\sqrt{r_1}}{1-\lambda}\sum_{t=0}^{T-1}\mathbb{E}\left[\|\nabla f(W_t) - M_t\|_{\mathrm{F}}\right] + \frac{(r_2 + \lambda^2 r_3)L\eta T}{1-\lambda}$$
$$\leq \frac{f(W_0) - f(W_T)}{(1-\lambda)\eta} + \frac{2\sqrt{r_1}}{1-\lambda}\left\{\frac{4}{1-\beta}\|M_0 - \nabla f(W_0)\|_{\mathrm{F}} + \sqrt{\frac{2(1-\beta)\sigma^2}{b}}T + \frac{2L\eta\sqrt{r_2}}{1-\beta}T\right\}$$
$$+ \frac{(r_2 + \lambda^2 r_3)L\eta T}{1-\lambda}.$$

By taking the average over $t = 0, \ldots, T-1$, we obtain

$$\frac{1}{T}\sum_{t=0}^{T-1}\mathbb{E}\left[\|\nabla f(W_t)\|_*\right] \leq \frac{f(W_0) - f(W_T)}{(1-\lambda)\eta T} + \frac{8\sqrt{r_1}\|M_0 - \nabla f(W_0)\|}{(1-\beta)(1-\lambda)T} + \frac{2\sqrt{2(1-\beta)}}{1-\lambda}\sqrt{\frac{r_1\sigma^2}{b}}$$
$$+ \frac{4L\eta\sqrt{r_1 r_2}}{(1-\beta)(1-\lambda)} + \frac{(r_2 + \lambda^2 r_3)L\eta}{(1-\lambda)}$$
$$= \mathcal{O}\left(\frac{1}{\eta T} + \sqrt{\frac{r_1(1-\beta)}{b}} + \eta\hat{r}\right).$$

This completes the proof. $\qquad\square$

## C.4 Proof of Theorem 3.2(ii)

*Proof.* From the definition of $C_t$,

$$\|C_t - \nabla f(W_t)\|_{\mathrm{F}} \leq \beta\|M_t - \nabla f(W_t)\|_{\mathrm{F}} + (1-\beta)\|\nabla f_{\mathcal{S}_t}(W_t) - \nabla f(W_t)\|_{\mathrm{F}}.$$

Therefore, according to Lemma A.1 and Theorem C.1, we find that

$$\sum_{t=0}^{T-1}\mathbb{E}\left[\|\nabla f(W_t)\|_*\right] \leq \frac{f(W_0) - f(W_T)}{(1-\lambda)\eta} + \frac{(r_2 + \lambda^2 r_3)L\eta T}{1-\lambda} + \frac{2\sqrt{r_1}(1-\beta)}{1-\lambda}\sum_{t=0}^{T-1}\mathbb{E}\left[\|\nabla f_{\mathcal{S}_t}(W_t) - \nabla f(W_t)\|_{\mathrm{F}}\right]$$

$$+ \frac{2\sqrt{r_1}}{1-\lambda}\beta\sum_{t=0}^{T-1}\mathbb{E}\left[\|\nabla f(W_t) - M_t\|_{\mathrm{F}}\right]$$

$$\leq \frac{f(W_0) - f(W_T)}{(1-\lambda)\eta} + \frac{(r_2 + \lambda^2 r_3)L\eta T}{1-\lambda} + \frac{2\sqrt{r_1}(1-\beta)\sigma}{(1-\lambda)\sqrt{b}}T$$

$$+ \frac{2\sqrt{r_1}}{1-\lambda}\beta\left\{\frac{4}{1-\beta}\|M_0 - \nabla f(W_0)\|_{\mathrm{F}} + \sqrt{\frac{2(1-\beta)\sigma^2}{b}}T + \frac{2L\eta\sqrt{n}}{1-\beta}T\right\}.$$

By taking the average over $t = 0, \ldots, T-1$, we obtain

$$\frac{1}{T}\sum_{t=0}^{T-1}\mathbb{E}\left[\|\nabla f(W_t)\|_*\right] \leq \frac{f(W_0) - f(W_T)}{(1-\lambda)\eta T} + \frac{8\beta\sqrt{r_1}\|M_0 - \nabla f(W_0)\|_{\mathrm{F}}}{(1-\beta)(1-\lambda)T} + \frac{2(1+\sqrt{2}\beta)\sqrt{1-\beta}}{1-\lambda}\sqrt{\frac{r_1\sigma^2}{b}}$$

$$+ \frac{(r_2 + \lambda^2 r_3)L\eta}{1-\lambda} + \frac{4L\beta\eta\sqrt{r_1 r_2}}{(1-\beta)(1-\lambda)}$$

$$= \mathcal{O}\left(\frac{1}{\eta T} + \sqrt{\frac{r_1(1-\beta)}{b}} + \eta\hat{r}\right),$$

where we use $(\beta\sqrt{2(1-\beta)} + (1-\beta)) \leq (1 + \sqrt{2}\beta)\sqrt{1-\beta}$. This completes the proof. $\square$

# D  Proof of Proposition 3.3 for Muon with Newton–Schulz

*Proof.* According to Assumption 2.1,

$$f(W_{t+1}) \leq f(W_t) + \langle\nabla f(W_t), W_{t+1} - W_t\rangle_{\mathrm{F}} + \frac{L}{2}\|W_{t+1} - W_t\|_{\mathrm{F}}^2$$

$$= f(W_t) - \eta\langle\nabla f(W_t), \tilde{O}_t^{(k)} + \lambda W_t\rangle_{\mathrm{F}} + \frac{L\eta^2}{2}\|\tilde{O}_t^{(k)} + \lambda W_t\|_{\mathrm{F}}^2$$

$$= f(W_t) - \eta\langle C_t, O_t + \lambda W_t\rangle_{\mathrm{F}} - \eta\langle\nabla f(W_t) - C_t, O_t + \lambda W_t\rangle_{\mathrm{F}} - \eta\langle\nabla f(W_t), \tilde{O}_t^{(k)} - O_t\rangle + \frac{L\eta^2}{2}\|\tilde{O}_t^{(k)} + \lambda W_t\|_{\mathrm{F}}^2.$$

From Eq. (8) and the reverse triangle inequality,

$$-\eta\langle C_t, O_t + \lambda W_t\rangle_{\mathrm{F}} \leq -(1-\lambda)\eta\|C_t\|_* \leq (1-\lambda)\eta\sqrt{r_1}\|C_t - \nabla f(W_t)\|_{\mathrm{F}} - (1-\lambda)\eta\|\nabla f(W_t)\|_*.$$

We recall the following inequality (see (9)):

$$-\eta\langle\nabla f(W_t) - C_t, O_t + \lambda W_t\rangle_{\mathrm{F}} \leq (1+\lambda)\eta\sqrt{r_1}\|\nabla f(W_t) - C_t\|_{\mathrm{F}}.$$

On the other hand, from Hölder's inequality and $\|\tilde{O}_t^{(k)} - O_t\|_{\mathrm{op}} \leq \delta_{(k)}$, we have

$$-\eta\langle\nabla f(W_t), \tilde{O}_t^{(k)} - O_t\rangle_{\mathrm{F}} \leq \eta\|\nabla f(W_t)\|_*\|\tilde{O}_t^{(k)} - O_t\|_{\mathrm{op}} \leq \eta\delta_{(k)}\|\nabla f(W_t)\|_*.$$

Finally, we bound the last term. Since $\|\tilde{O}_t^{(k)} - O_t\|_{\mathrm{op}} \leq \delta_{(k)}$, we have

$$\|\tilde{O}_t^{(k)}\|_{\mathrm{F}}^2 \leq r_2\|\tilde{O}_t^{(k)}\|_{\mathrm{op}}^2 = r_2\|O_t + \tilde{O}_t^{(k)} - O_t\|_{\mathrm{op}}^2 \leq 2r_2\|O_t\|_{\mathrm{op}}^2 + 2r_2\|\tilde{O}_t^{(k)} - O_t\|_{\mathrm{op}}^2 \leq 2r_2 + 2r_2\delta_{(k)}^2.$$

From Lemma C.1, we have $\|W_t\|_{\mathrm{F}}^2 \leq r_3 \|W_t\|_{\mathrm{op}}^2 \leq r_3$. Therefore,

$$\frac{L\eta^2}{2} \|\tilde{O}_t^{(k)} + \lambda W_t\|_{\mathrm{F}}^2 \leq L\eta^2 \left( \|\tilde{O}_t^{(k)}\|_{\mathrm{F}}^2 + \lambda^2 \|W_t\|_{\mathrm{F}}^2 \right) \leq L\eta^2 \left( 2r_2 + 2r_2\delta_{(k)}^2 + \lambda^2 r_3 \right).$$

Hence,

$$f(W_{t+1}) \leq f(W_t) + 2\sqrt{r_1}\eta \|C_t - \nabla f(W_t)\|_{\mathrm{F}} - (1 - \lambda - \delta_{(k)})\eta \|\nabla f(W_t)\|_* + L\eta^2 \left( 2r_2 + 2r_2\delta_{(k)}^2 + \lambda^2 r_3 \right).$$

By rearranging the term and taking the expectation, we obtain

$$\mathbb{E}\left[\|\nabla f(W_t)\|_*\right] \leq \frac{f(W_t) - f(W_{t+1})}{(1 - \lambda - \delta_{(k)})\eta} + \frac{2\sqrt{r_1}}{1 - \lambda - \delta_{(k)}} \mathbb{E}\left[\|C_t - \nabla f(W_t)\|_{\mathrm{F}}\right] + \frac{(2r_2 + 2r_2\delta_{(k)}^2 + \lambda^2 r_3)L\eta}{1 - \lambda - \delta_{(k)}},$$

Summing over $t = 0$ to $t = T - 1$,

$$\sum_{t=0}^{T-1} \mathbb{E}\left[\|\nabla f(W_t)\|_*\right] \leq \frac{f(W_0) - f(W_T)}{(1 - \lambda - \delta_{(k)})\eta} + \frac{2\sqrt{r_1}}{1 - \lambda - \delta_{(k)}} \sum_{t=0}^{T-1} \mathbb{E}\left[\|C_t - \nabla f(W_t)\|_{\mathrm{F}}\right] + \frac{(2r_2 + 2r_2\delta_{(k)}^2 + \lambda^2 r_3)L\eta}{1 - \lambda - \delta_{(k)}}T,$$

From the definition of $C_t$, Lemma A.1 and A.2, we obtain

$$\sum_{t=0}^{T-1} \mathbb{E}\left[\|C_t - \nabla f(W_t)\|_*\right] \leq \beta \sum_{t=0}^{T-1} \mathbb{E}\left[\|M_t - \nabla f(W_t)\|_{\mathrm{F}}\right] + (1 - \beta) \sum_{t=0}^{T-1} \mathbb{E}\left[\|\nabla f_{\mathcal{S}_t}(W_t) - \nabla f(W_t)\|_{\mathrm{F}}\right]$$

$$\leq \frac{4\beta}{1 - \beta} \|M_0 - \nabla f(W_0)\|_{\mathrm{F}} + \beta \sqrt{\frac{2(1 - \beta)\sigma^2}{b}} T + \frac{2L\beta\eta\sqrt{r_2}}{1 - \beta}T + (1 - \beta)\sqrt{\frac{\sigma^2}{b}}T$$

$$\leq \frac{4\beta}{1 - \beta} \|M_0 - \nabla f(W_0)\|_{\mathrm{F}} + (1 + \sqrt{2}\beta)\sqrt{\frac{(1 - \beta)\sigma^2}{b}}T + \frac{2L\beta\eta\sqrt{r_2}}{1 - \beta}T,$$

where we use $\beta\sqrt{2(1 - \beta)} + 1 - \beta \leq (1 + \sqrt{2}\beta)\sqrt{1 - \beta}$. Hence,

$$\frac{1}{T} \sum_{t=0}^{T-1} \mathbb{E}\left[\|\nabla f(W_t)\|_*\right] \leq \frac{f(W_0) - f(W_T)}{(1 - \lambda - \delta_{(k)})\eta T} + \frac{8\beta\sqrt{r_1}\|M_0 - \nabla f(W_0)\|_{\mathrm{F}}}{(1 - \beta)(1 - \lambda - \delta_{(k)})T} + \frac{2(1 + \sqrt{2}\beta)\sqrt{1 - \beta}}{1 - \lambda - \delta_{(k)}}\sqrt{\frac{r_1\sigma^2}{b}}$$

$$+ \frac{4L\beta\eta\sqrt{r_1 r_2}}{(1 - \beta)(1 - \lambda - \delta_{(k)})} + \frac{(2r_2 + \lambda^2 r_3)L\eta}{1 - \lambda - \delta_{(k)}} + \frac{2r_2\delta_{(k)}^2 L\eta}{1 - \lambda - \delta_{(k)}}$$

$$= \mathcal{O}\left( \frac{1}{\eta T} + \sqrt{\frac{r_1(1 - \beta)}{b}} + \eta\hat{r} \right).$$

This completes the proof. $\qquad\square$

# E    Experimental Details

This section details the experimental setup for evaluating the optimizers. Our code will be available at `https://anonymous.4open.science/r/critical_batchsize_muon-0D38/README.md`.

**Workloads and General Setup (Vision)**    All CIFAR experiments used ResNet-18 on CIFAR-10 and VGG-16 on CIFAR-100, trained from scratch. Training was performed for a fixed number of samples corresponding to 100 epochs at a batch size of 512. For any batch size $B$, the number of epochs was scaled as $E_B = 100 \times (512/B)$ to keep the number of seen samples constant. Each configuration was repeated with five random seeds, and we report mean and standard deviation of test accuracy.

**Workload and System Setup (Language Modeling: C4/ Llama3.1 (320M))**    For the languagemodeling workload we trained Llama3.1 (320M) on the C4 dataset with sequence length 2048. We used the `torchtitan` codebase with PyTorch FSDP (full sharding) and activation checkpointing (`mode=full`). Gradient clipping with max norm 1.0 was applied. Due to sharding, Muon requires an additional *all-gather of gradients* before the optimizer step, which introduces one extra collective communication per step. We therefore report SFO-based metrics (steps × batch size) in addition to loss to separate algorithmic efficiency from system overhead.

Unless otherwise noted, the training budget was fixed to 3.2B tokens. At a base global batch size (GBS) of 128 this corresponds to 12,208 steps; for other GBS values the number of steps scales inversely to keep the token budget constant. We ran on up to 8 H100 GPUs with per-GPU micro-batch size 8 and no tensor, pipeline, or context parallelism (`TP=PP=CP=1`).

**Learning-Rate Schedules**    Vision workloads used a grid search at base batch size 512 and square-root scaling to other batch sizes for Muon and AdamW. Momentum SGD was tested with both square-root and linear scaling. For Llama3.1 (320M), we used linear warmup followed by cosine decay. The base configuration uses 2000 warmup steps; in sweeps we also parameterized warmup as 10% of total steps to maintain a comparable schedule across different GBS.

**Hyperparameter Tuning Protocol (Vision)**    We performed an extensive grid search over base learning rates and weight decay. For $B \neq 512$, learning rates were scaled from the base value using $\eta_B = \eta_{512}\sqrt{B/512}$ for AdamW and Muon; for Momentum SGD we report both $\sqrt{B/512}$ and $B/512$ scaling. The shared search space is summarized in Table 3, and optimizer-specific spaces in Table 4.

Table 3: Shared hyperparameters for vision experiments.

| Hyperparameter | Value / Search Space |
|---|---|
| Model Architecture | ResNet-18, VGG-16 |
| Dataset | CIFAR-10, CIFAR-100 |
| Batch Size ($B$) | {2, 4, 8, 16, 32, 64, 128, 256, 512, 1024, 2048, 4096} |
| Base Epochs (at $B = 512$) | 100 |
| Random Seeds | 5 |

**Optimizer Configuration (Vision)**    For the hybrid Muon optimizer, we applied the Muon update to all convolutional layers in ResNet-18 (excluding the input stem). Biases, batch-norm parameters, and the final linear layer were updated by AdamW. For standard AdamW and Momentum SGD, the respective optimizer was applied to all parameters.

**Search Space (Language Modeling)**    For C4/ Llama3.1 (320M), we performed sweeps using grid search. GBS took values in {64, 256, 1024, 4096} when resources permitted. We tuned Muon and the AdamW baseline per GBS without automatic LR scaling.

**Tuning fairness.**    To ensure a fair comparison across optimizers, we applied the same tuning protocol to all methods: at the base batch size, we performed a grid search over learning rates; at other batch sizes, we applied standard scaling rules (square-root scaling for Muon and AdamW, square-root and linear for Momentum SGD). We selected the configuration that minimizes SFO complexity to the target accuracy (or target loss for LLM experiments).

Table 4: Optimizer-specific hyperparameter search spaces (vision). The base learning rate ($\eta_{512}$) is specified at a reference batch size of 512.

| Hyperparameter | AdamW | Momentum SGD | Muon |
|---|---|---|---|
| **Searched Parameters** | | | |
| Base Learning Rate ($\eta_{512}$) | {0.01, 0.001, 0.0001} | {0.001, 0.0005, 0.0001} | |
| - Muon component | — | — | {0.01, 0.005, 0.001} |
| - AdamW component | — | — | {0.01, 0.001, 0.0001} |
| Momentum | — | 0.9 | {0.7, 0.8, 0.9, 0.95, 0.99, 0.999} |
| Weight Decay ($\lambda$) | {0.1, 0.01, 0.001, 0.0001, 0} | {0.1, 0.01, 0.001, 0.0001, 0} | {0.1, 0.01, 0.001, 0.0001, 0} |
| **Fixed Parameters** | | | |
| Learning Rate Scaling | $\sqrt{B/512}$ | $\sqrt{B/512}$ (and $B/512$) | $\sqrt{B/512}$ |
| Adam $\beta_1, \beta_2$ | 0.9, 0.999 (default) | — | 0.9, 0.999 (default) |
| Adam $\epsilon$ | 1e-8 (default) | — | 1e-8 (default) |

Table 5: Number of parameters updated by each optimizer component in the Muon setup for ResNet-18, VGG-16 and Llama3.1 (320M).

| Model | Muon params | AdamW params |
|---|---|---|
| ResNet-18 | 11,157,504 | 16,458 |
| VGG-16 | 14,712,896 | 19,302,500 |
| Llama3.1 (320M) | 127,401,984 | 49,180,416 |

Table 6: Shared configuration for C4/ Llama3.1 (320M).

| Item | Value |
|---|---|
| Model | Llama3.1 (320M) (dim = 768, $n_{\text{layers}} = 18$, $n_{\text{heads}} = 12$) |
| Dataset | C4 |
| Sequence length | 2048 |
| Global batch size (GBS) | $\{32, 64, 128, 256, 512, 1024, 2048, 4096\}$ |
| Total tokens | $3.2 \times 10^9$ |
| Base steps (GBS=128) | 12,208 |
| Schedule | linear warmup then cosine decay (decay ratio 0.5) |
| Warmup | 2000 steps (base) or 10% of total steps (sweep) |
| Gradient clipping | max-norm = 1.0 |
| Parallelism | FSDP; TP=1, PP=1, CP=1 |
| Hardware | up to 8 H100 GPUs; local batch size = 8 |

This ensures that all optimizers are given equal opportunity to find their best operating point, and that differences in SFO complexity reflect genuine algorithmic advantages rather than tuning asymmetry.

**Compute fairness and the choice of SFO complexity.** We use stochastic first-order oracle (SFO) complexity (steps × batch size) as the primary efficiency metric because it is hardware-independent: it counts the total

Table 7: Optimizer search spaces for C4/ Llama3.1 (320M). Component-wise LRs are tuned per GBS.

| Hyperparameter | Muon | AdamW (baseline) |
|---|---|---|
| Muon LR | {0.005, 0.01, 0.02, 0.04} | — |
| Muon momentum $\beta$ | {0.8, 0.9, 0.95, 0.99} | — |
| Muon Nesterov | {True, False} | — |
| Muon weight decay | {0, 0.1} | — |
| AdamW LR (component or baseline) | {0.001, 0.002, 0.004, 0.008} | {0.001, 0.002, 0.004, 0.008} |
| AdamW weight decay | 0.1 | 0.1 |
| AdamW $\beta_1, \beta_2$ | 0.9, 0.999 | 0.9, 0.999 |
| AdamW $\epsilon$ | $1\times10^{-8}$ | $1\times10^{-8}$ |

number of individual gradient evaluations regardless of the parallelism, memory bandwidth, or accelerator type used. This makes SFO comparisons reproducible across different hardware setups. We acknowledge that wall-clock time is ultimately the metric of practical interest; however, wall-clock comparisons conflate algorithmic efficiency with implementation-specific factors (e.g., the extra all-gather communication required by Muon under FSDP sharding). By reporting SFO complexity, we isolate the algorithmic contribution. Note that Jordan et al. (2024) demonstrated that Muon achieves favorable wall-clock performance relative to both Shampoo and SOAP, and AI et al. (2025) showed that Muon expands AdamW's Pareto frontier on the compute-time plane.

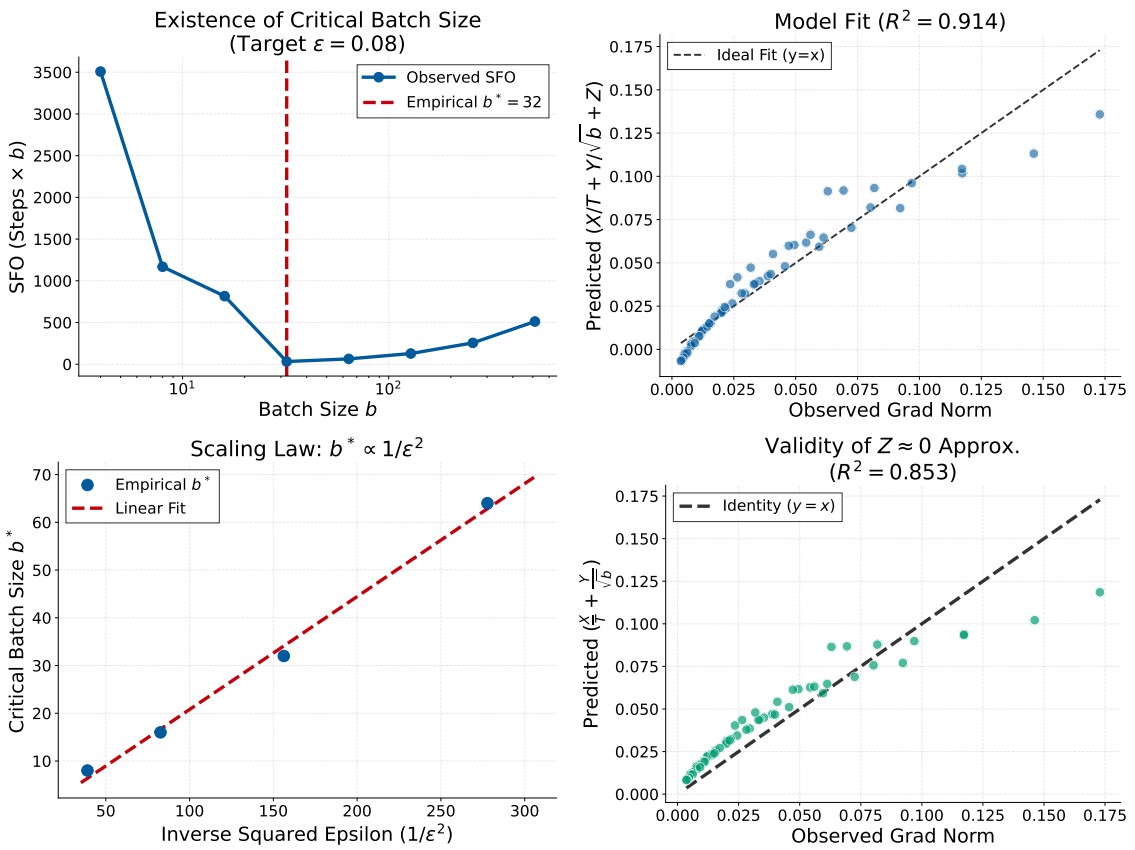

Figure 7: **Validation of critical batch size theory on a controlled full-Muon task. (Top Left)** SFO vs. batch size $b$ required to reach a target gradient norm $\varepsilon = 0.08$. A clear empirical optimum (critical batch size) is observed at $b^\star = 32$. **(Top Right)** Goodness-of-fit for the theoretical proxy $\bar{g}(T, b) \approx X/T + Y/\sqrt{b} + Z$ against observed gradient norms ($R^2 \approx 0.914$). **(Bottom Left)** The empirical critical batch size $b^\star$ scales with the inverse squared target precision $1/\varepsilon^2$, consistent with the theoretical prediction $b^\star \propto 1/\varepsilon^2$ from Proposition 4.2. **(Bottom Right)** Goodness-of-fit for the theoretical proxy $\bar{g}(T, b) \approx X/T + Y/\sqrt{b}$ against observed gradient norms ($R^2 \approx 0.853$).

# F Additional Diagnostics: Gradient-norm proxy and full-Muon toy setting

**Toy task and full-Muon configuration.** As described in Section 5, we use a controlled full-Muon experiment to provide an exact test of the theoretical predictions, free from the hybrid optimizer confound. We conduct experiments on a controlled synthetic task where Muon is applied to all parameters. We employ a **Teacher-Student Tanh Regression** problem. A fixed teacher matrix $W^\star \in \mathbb{R}^{m \times n}$ and a student matrix $W \in \mathbb{R}^{m \times n}$ are initialized with entries drawn from $\mathcal{N}(0, 1/n)$. At each step $t$, we sample inputs $x_t \sim \mathcal{N}(0, I_n)$ and generate labels $y_t = \tanh(x_t(W^\star)^\top) + \xi_t$, where $\xi_t \sim \mathcal{N}(0, \sigma^2 I)$ represents label noise. The student minimizes the loss $\mathcal{L}(W) = \frac{1}{2}\|\tanh(x_t W^\top) - y_t\|^2$. We set the dimensions to $m = 256, n = 128$, and the noise level to $\sigma = 0.1$. Unlike the large-scale experiments, here Muon (Algorithm 1) is applied to the entire matrix $W$ with learning rate $\eta = 0.05$, momentum $\beta = 0.95$ (Nesterov), and 5 Newton-Schulz iterations.

**Measured quantities and stopping rule.** To rigorously validate the theoretical convergence bound $\mathbb{E}[\|\nabla \mathcal{L}\|] \lesssim \frac{X}{T} + \frac{Y}{\sqrt{b}} + Z$, we track the *cumulative mean* of the gradient norm, denoted as $\bar{g}_t = \frac{1}{t}\sum_{i=1}^{t}\|\nabla \mathcal{L}(W_i)\|_F$. We define the convergence step $T_\varepsilon(b)$ as the first step $t$ where $\bar{g}_t \leq \varepsilon$. The stochastic first-order oracle (SFO) complexity is computed as SFO $= b \cdot T_\varepsilon(b)$. We perform a grid search over batch sizes $b \in \{4, 8, \ldots, 512\}$, extended to $b \in \{4, 8, \ldots, 2048\}$ in a follow-up sweep.

**Results and Validity of Approximations.** We confirm three key findings in this controlled setting: (1) **Existence of Critical Batch Size:** As shown in Figure 7 (Top Left), the SFO complexity exhibits a clear convex shape with respect to $b$, identifying an empirical optimum $b^\star$. (2) **Model Fit:** The observed data fits the theoretical proxy $\frac{X}{T} + \frac{Y}{\sqrt{b}} + Z$ with high accuracy ($R^2 \approx 0.914$), validating our convergence analysis. (3) **Justification for** $Z \approx 0$**:** To address the theoretical concern regarding the non-vanishing term $Z$, we compared the full fit against a simplified model $\frac{X}{T} + \frac{Y}{\sqrt{b}}$ (forcing $Z = 0$). Figure 7 (Bottom Right) demonstrates that the simplified model still predicts the observed gradient norms with reasonable accuracy ($R^2 \approx 0.853$), supporting the practical validity of the $Z \approx 0$ approximation. This empirically justifies our derivation of the critical batch size $b^\star \propto 1/\varepsilon^2$ by treating $Z$ as negligible during the primary optimization phase.

**Interpretation of negative $\hat{Z}$.** The fitted intercept $\hat{Z}$ is negative in all four variants (ranging from $-1.64$ to $-2.56$), whereas the theoretical term $Z = \mathcal{O}(L\eta\hat{r})$ is non-negative. This discrepancy does not undermine the proxy validation. The regression is performed over a finite $(T, b)$ grid, and $\hat{Z}$ represents the best linear-in-$(1/T, 1/\sqrt{b})$ approximation to the gradient norm surface within that grid. Since the true convergence bound is an *upper* bound, the actual gradient norm can lie well below the bound; a negative fitted intercept simply reflects this slack. Moreover, the primary purpose of the proxy fit is to validate the *functional form* (how the gradient norm depends on $T$ and $b$), not the absolute constants. The high $R^2 \approx 0.914$ confirms that the $X/T + Y/\sqrt{b}$ terms capture the dominant variation, which is the theoretically meaningful finding.

**Predicted CBS exceeds the sweep range.** The CBS values predicted from the fitted proxy ($\hat{b}^\star = 9\hat{Y}^2/(4\varepsilon^2)$, accounting for $\hat{Z} < 0$) range from approximately 1500 to 2000 for $\varepsilon \in [0.05, 0.20]$, which exceeds the maximum batch size in our sweep ($b = 2048$). Consequently, no empirical CBS is observed: the cumulative mean gradient norm does not fall below any $\varepsilon$ threshold within $T=10{,}000$ steps, even at $b=2048$. This is consistent with the theory, as the predicted CBS is simply larger than the experimental sweep range. Importantly, the *primary purpose* of the toy experiment is to validate the *functional form* of the convergence bound (the $X/T + Y/\sqrt{b} + Z$ decomposition), not to observe the CBS transition itself. The high proxy fit quality ($R^2 \approx 0.93$) confirms that the theory accurately describes the gradient norm surface, and the CBS formula follows algebraically from this validated functional form. We note, however, that when the predicted CBS exceeds the experimental sweep range, the $R^2=0.914$ proxy fit confirms the *functional form* but the CBS *point estimate* extrapolates beyond the observed data, which reduces confidence in the specific CBS value relative to settings where the transition is directly observed.

**Newton-Schulz follow-up and remaining experimental gap.** We also ran a seeded follow-up study around the practical Newton-Schulz choice $k \in \{3, 5\}$ on ResNet-18/CIFAR-10 across widths $\{0.25, 0.5, 1.0, 2.0\}$, with three seeds per configuration. The result is that $k=3$ remains the best global default: at widths 0.25 and 0.5, moving from $k=3$ to 5 does not produce meaningful test-accuracy gains; at width 2.0, the approximation error decreases but the test accuracy is essentially unchanged; and width 1.0 is the clearest case where $k=5$ is justified. This supports our choice to use five iterations as a robust practical default while indicating that fewer iterations are often sufficient in vision workloads. Table 8 reports the aggregated test accuracy and Newton-Schulz approximation error (mean $\pm$ std over seeds). The delta analysis shows that the accuracy gain from $k=5$ is at most 0.15% (at width 1.0), while the NS error reduction is substantial only at widths 1.0 and 2.0. This indicates that for widths where the NS approximation is already tight ($w \le 0.5$), additional iterations do not translate into accuracy improvements.

Table 8: Newton-Schulz $k \in \{3, 5\}$ comparison across widths (ResNet-18/CIFAR-10, Muon). Each cell reports mean $\pm$ std over 3 seeds. $\Delta$ denotes $k=5$ minus $k=3$.

| Width | Best Test Acc (%) | | Mean NS Error | | $\Delta$ ($k=5 - k=3$) | |
|---|---|---|---|---|---|---|
| | $k=3$ | $k=5$ | $k=3$ | $k=5$ | $\Delta$ Acc | $\Delta$ NS Err |
| $0.25\times$ | $88.09 \pm 0.51$ | $88.15 \pm 0.44$ | $0.192 \pm 0.001$ | $0.196 \pm 0.002$ | $+0.05$ | $+0.004$ |
| $0.50\times$ | $90.60 \pm 0.30$ | $90.51 \pm 0.10$ | $0.244 \pm 0.004$ | $0.190 \pm 0.001$ | $-0.08$ | $-0.055$ |
| $1.00\times$ | $91.89 \pm 0.14$ | $92.05 \pm 0.11$ | $0.366 \pm 0.002$ | $0.203 \pm 0.005$ | $+0.15$ | $-0.163$ |
| $2.00\times$ | $92.65 \pm 0.14$ | $92.53 \pm 0.34$ | $0.519 \pm 0.004$ | $0.267 \pm 0.003$ | $-0.12$ | $-0.251$ |

**Rank tracking: SFO comparison across optimizers.** We ran a rank-tracking experiment on ResNet-18/CIFAR-10 across batch sizes $\{8, 32, 128, 512, 2048\}$ for Muon, AdamW, and Momentum SGD. Table 9 reports the best SFO complexity to reach 80% test accuracy for each (optimizer, batch size) pair. The heuristic CBS (the batch size with the smallest SFO) is $b^\star=32$ for Muon and $b^\star=8$ for AdamW, confirming that Muon's CBS exceeds AdamW's.

We also measured the early-training effective rank at larger batch sizes ($b \geq 128$): Muon exhibits higher effective rank than AdamW and Momentum SGD across all tested batch sizes (e.g., erank $\approx 148$ for Muon vs. $\approx 103$ for AdamW at $b{=}128$), and the mean Newton-Schulz approximation error decreases with batch size for all optimizers, reaching $\approx 0.30$–$0.49$ at $b{=}2048$.

Table 9: Best SFO complexity to reach 80% test accuracy on ResNet-18/CIFAR-10 by optimizer and batch size. † marks the batch size with the lowest SFO (heuristic CBS).

| Optimizer | $b{=}8$ | $b{=}32$ | $b{=}128$ | $b{=}512$ | $b{=}2048$ |
|---|---|---|---|---|---|
| Muon | $1.00 \times 10^5$ | $\mathbf{5.00 \times 10^4}^{\dagger}$ | $9.98 \times 10^4$ | $9.93 \times 10^4$ | $2.46 \times 10^5$ |
| AdamW | $\mathbf{1.50 \times 10^5}^{\dagger}$ | $2.00 \times 10^5$ | $2.50 \times 10^5$ | $2.98 \times 10^5$ | $5.90 \times 10^5$ |
| SGD+M | $2.00 \times 10^5$ | $\mathbf{2.00 \times 10^5}^{\dagger}$ | $2.50 \times 10^5$ | $3.48 \times 10^5$ | $6.39 \times 10^5$ |

## F.1 Proxy Calibration: Full-Gradient Norm vs. Mini-Batch Surrogates

To directly address the discrepancy between the theoretical quantity (the full-batch gradient norm $\|\nabla f(W_t)\|_{\mathrm{F}}$) and the experimental proxy (the EMA-smoothed mini-batch gradient norm), we compute both quantities at every training step in the controlled full-Muon Teacher-Student setting described above.

Table 10 reports the Pearson correlation between the full-batch gradient norm and two stochastic surrogates (the raw mini-batch gradient norm and the EMA-smoothed proxy) across batch sizes and two noise levels ($\sigma{=}0.1$ and $\sigma{=}0.2$), aggregated over 6 independent seeds each.

Table 10: Pearson correlation between the full-batch gradient norm and mini-batch surrogates in the controlled full-Muon Teacher-Student task ($m{=}512$, $n{=}256$, 6 seeds). All correlations exceed 0.97 (mini-batch) and 0.95 (EMA), confirming that the surrogates faithfully track the theoretical quantity.

| Batch size $b$ | $\sigma{=}0.1$ | | $\sigma{=}0.2$ | |
|---|---|---|---|---|
| | $\rho(\text{full, batch})$ | $\rho(\text{full, EMA})$ | $\rho(\text{full, batch})$ | $\rho(\text{full, EMA})$ |
| 8 | 0.994 | 0.996 | 0.991 | 0.997 |
| 16 | 0.998 | 0.990 | 0.995 | 0.991 |
| 32 | 0.999 | 0.984 | 0.997 | 0.985 |
| 64 | 1.000 | 0.981 | 0.999 | 0.981 |
| 128 | 1.000 | 0.979 | 0.999 | 0.979 |
| 256 | 1.000 | 0.978 | 1.000 | 0.978 |
| 512 | 1.000 | 0.978 | 1.000 | 0.978 |
| 1024 | 1.000 | 0.978 | 1.000 | 0.978 |

The high correlation ($> 0.97$ in all cases) confirms that mini-batch gradient norms are a well-calibrated surrogate for the full gradient norm. We note that the absolute CBS point estimate can differ between the full-gradient metric and the stochastic proxy, because the stochastic metrics include mini-batch noise that the full-gradient metric does not. However, the high correlation indicates that both metrics track the same underlying optimization dynamics, and the qualitative CBS behavior (U-shaped SFO curve, variant ordering direction) is preserved.

## F.2 Newton-Schulz Approximation Gap: Controlled Comparison

To complement the practical NS($k$) comparison on ResNet-18 (Table 8), we compare exact SVD orthogonalization against NS($k{=}3$) and NS($k{=}5$) in the controlled full-Muon Teacher-Student setting ($m{=}512$, $n{=}256$, $\sigma{=}0.1$). Table 11 reports the final gradient norm and mean orthogonalization error across batch sizes, aggregated over 10 seeds.

All three methods converge successfully; the final gradient norms differ due to slightly different optimization trajectories induced by the approximation, but all reach the same order of magnitude. The NS approximation error is stable across batch sizes and small enough ($< 0.04$) to be negligible relative to the gradient scale.

## F.3 Four-Variant CBS Ordering in the Controlled Full-Muon Setting

Table 11: Exact SVD vs. Newton-Schulz approximation in the controlled full-Muon Teacher-Student task ($m=512$, $n=256$, $\sigma=0.1$, 10 seeds). All three orthogonalizers converge to comparable gradient norms despite differing approximation errors, confirming that the NS gap does not materially affect optimization.

| | Final $\|\nabla f\|_{\mathrm{F}}$ ($\times 10^{-3}$) | | | Mean orth. error | | |
| --- | --- | --- | --- | --- | --- | --- |
| | $b=32$ | $b=128$ | $b=512$ | $b=32$ | $b=128$ | $b=512$ |
| SVD (exact) | 9.77 | 7.11 | 3.78 | 0 | 0 | 0 |
| NS($k=3$) | 4.28 | 2.77 | 1.81 | 0.037 | 0.036 | 0.036 |
| NS($k=5$) | 6.68 | 4.12 | 2.48 | 0.030 | 0.028 | 0.028 |

**SFO plateau and inter-variant agreement on FashionMNIST.** The ResNet-18/CIFAR-10 runs underlying Figure 3 start at batch size $b=8$, which already sits at (or above) the CBS for every Muon variant and therefore provides limited resolution for comparing variants *at and below* the CBS. To obtain a more controlled microscope for variant-level differences near the plateau, we ran a compact FashionMNIST sweep on a 2-layer MLP (hidden dim 1024, target test accuracy 0.84), which allows us to sweep batch sizes down to $b=4$ within a short time budget and tune learning rates densely. For each (variant, $b$) cell we report the best SFO (lowest median across three seeds) over a learning-rate grid ($\{0.02, 0.05, 0.1\}$ for Muon variants and $\{3\times10^{-4}, 10^{-3}, 3\times10^{-3}\}$ for AdamW), matching the "best-over-LR" protocol used for CIFAR-10.

Table 12 shows the resulting SFO grid. Three observations stand out: (i) at the optimal batch size $b=8$, the four Muon variants achieve best SFO in the range 21.6–22.6K, a maximum spread of 4.6%; (ii) across the plateau ($b \in \{8, 16, 32, 64\}$) every Muon variant's best SFO lies in a 19.2–30.4K band (within a factor of 1.58×), so the predicted inter-variant CBS differences (Table 1) are again smaller than the plateau width; (iii) unlike ResNet-18/CIFAR-10, AdamW is *comparable* to Muon on this simpler MLP/FashionMNIST task: the AdamW-to-best-Muon SFO ratio ranges from 0.96× to 1.75× across batch sizes. We flag (iii) explicitly: it illustrates that the Muon-vs-AdamW gap documented on ResNet-18/CIFAR-10 is task-dependent and narrows on simpler MLP benchmarks. The CBS mechanism analysed in this paper concerns *inter-variant* closeness (observations (i) and (ii)) and does not rely on any minimum Muon-vs-AdamW gap, so point (iii) does not affect the conclusion that practical CBS differences among the four Muon variants are within the plateau width.

Table 12: Best SFO (in thousands of examples) to reach test accuracy 0.84 on FashionMNIST (2-layer MLP, hidden 1024, 3 seeds; best over a learning-rate grid per optimizer family). Stars ($\star$) mark the argmin over $b$ (the empirical CBS for that row). The four Muon variants agree to within 4.6% at $b=8$ and lie within a 1.58× band across $b \in \{8, 16, 32, 64\}$. On this task AdamW is comparable to Muon, so the closeness we highlight is *inter-variant*, not Muon-vs-AdamW.

| Variant | $b=4$ | $b=8$ | $b=16$ | $b=32$ | $b=64$ | $b=128$ | $b=256$ |
| --- | --- | --- | --- | --- | --- | --- | --- |
| Muon | 37.0 | 22.4 | 21.6$^\star$ | 24.8 | 28.8 | 32.0 | 44.8 |
| Muon + Nesterov | 19.7 | 22.6 | 19.2$^\star$ | 22.4 | 22.4 | 25.6 | 25.6 |
| Muon + WD | 35.6 | 21.8$^\star$ | 23.6 | 25.6 | 30.4 | 35.2 | 44.8 |
| Muon + Nest + WD | 32.2 | 21.6 | 20.4 | 26.4 | 19.2$^\star$ | 22.4 | 25.6 |
| AdamW | 21.8 | 21.8 | 24.0 | 21.6$^\star$ | 28.8 | 32.0 | 44.8 |

We emphasise two caveats: the $b=4$ column shows a wider inter-variant spread (driven by Muon/Muon+WD at lr $\in \{0.02, 0.05\}$ failing to reach the target within the step budget, giving a pessimistic best-LR estimate on a narrow grid), and the empirical CBS location varies across variants ($b^\star \in \{8, 16, 64\}$). Neither affects the main quantitative claim in the plateau window $b \in \{8, 16, 32, 64\}$.

### F.4 Variant CBS Separation on ResNet-18/CIFAR-10

The four Muon variants in Figure 3 share similar CBS values because the hyperparameter differences (Nesterov on/off, small $\lambda$) produce only modest CBS shifts relative to the plateau width. To test whether the CBS formulas predict consequential differences when hyperparameters are varied more widely, we conducted a dedicated experiment on the same ResNet-18/CIFAR-10 setup with five configurations: Muon without Nesterov ($\beta=0.95$, $\lambda=0$); Muon with Nesterov at $\beta=0.95$ and $\beta=0.90$ (both $\lambda=0$); and Muon with Nesterov ($\beta=0.95$) combined with weight decay $\lambda \in \{0.2, 0.5\}$.

We swept batch sizes $b \in \{32, 64, 128, 256, 512, 1024, 2048, 4096\}$ using the same LR scaling $\eta = 0.005 \cdot \sqrt{b/512}$ and epoch scaling $E_b = \lceil 100 \cdot 512/b \rceil$ (66 runs, 1–3 seeds per cell).

Table 13 reports the SFO complexity (steps $\times$ batch size) to reach 95% training accuracy. A dash (—) indicates the target was not reached within the allocated epochs.

Table 13: SFO complexity ($\times 10^3$) to reach 95% training accuracy on ResNet-18/CIFAR-10 across five Muon configurations. Mean $\pm$ std shown where multiple seeds are available. Bold indicates the SFO-minimizing batch size (empirical CBS).

| Configuration | $b{=}32$ | $b{=}64$ | $b{=}128$ | $b{=}256$ | $b{=}512$ | $b{=}1024$ | $b{=}2048$ | $b{=}4096$ | CBS |
|---|---|---|---|---|---|---|---|---|---|
| w/o Nesterov, $\lambda{=}0$ | 650 | 600 | **549** | $566 \pm 29$ | $579 \pm 29$ | 590 | 639 | — | $\approx 128$ |
| w/ Nest., $\beta{=}0.95$, $\lambda{=}0$ | — | — | 549 | 549 | 546 | **541** | 541 | 590 | $\approx 1024$ |
| w/ Nest., $\beta{=}0.90$, $\lambda{=}0$ | — | — | 599 | 599 | $563 \pm 29$ | **541** | 541 | 590 | $\approx 1024$ |
| w/ Nest., $\beta{=}0.95$, $\lambda{=}0.2$ | — | — | — | — | 1043 | $705 \pm 28$ | **590** | 590 | $\approx 2048$ |
| w/ Nest., $\beta{=}0.95$, $\lambda{=}0.5$ | — | — | — | — | — | — | $950 \pm 28$ | **639**$^\dagger$ | $\geq 4096$ |

$\dagger$ $\lambda{=}0.5$ at $b{=}4096$: 1 of 3 seeds reached target.

**CBS ordering matches theory.** The empirical CBS ordering, w/o Nesterov ($\approx 128$) < w/ Nesterov ($\approx 1024$) < w/ Nesterov+WD$_{0.2}$ ($\approx 2048$) < w/ Nesterov+WD$_{0.5}$ ($\geq 4096$), is consistent with the theoretical lower bounds in Table 1: Nesterov shifts the CBS rightward, and weight decay provides an additional shift proportional to $1/(1{-}\lambda)^2$.

**Weight-decay CBS ratio matches quantitatively.** The ratio of CBS between the $\lambda{=}0.5$ and $\lambda{=}0$ Nesterov configurations equals $4096/1024 = 4.0$, which agrees with the predicted factor $1/(1{-}0.5)^2 = 4.0$. The corresponding ratio for $\lambda{=}0.2$ is $2048/1024 = 2.0$, consistent with the predicted $1/(1{-}0.2)^2 \approx 1.56$ within the power-of-two grid resolution. The Nesterov-to-non-Nesterov ratio ($1024/128 = 8.0$) exceeds the lower-bound prediction of $\approx 2.75$; this is expected because the formulas in Table 1 provide a lower bound, not an equality.

### F.5 Stopping-Target Sensitivity: CBS Robustness Across Metrics

To verify that the CBS conclusion is robust to the choice of stopping criterion, we analyzed FashionMNIST training logs (2-layer MLP, hidden dimension 1024, full Muon, 16 seeds) under stopping criteria spanning four metric types: test accuracy, train accuracy, train loss, and EMA-smoothed gradient norm. Table 14 summarizes the SFO-minimizing batch size for each criterion.

Two key observations emerge: (i) Across all four metric types, the CBS falls consistently in the range 32–256, confirming that the qualitative CBS conclusion is robust to the choice of stopping criterion. (ii) Within each metric type, tighter thresholds systematically shift the CBS to larger values, supporting the theoretical prediction $b^\star \propto 1/\varepsilon^2$. The EMA gradient norm sweep provides the most striking evidence: the CBS shifts monotonically as $32 \to 64 \to 128 \to 256$ as $\varepsilon$ decreases from 0.60 to 0.32.

### F.6 Full-Muon Weight Decay Sweep

To isolate the effect of weight decay $\lambda$ on the CBS in a controlled setting free from the hybrid-optimizer confound, we swept $\lambda \in \{0, 0.01, 0.05, 0.1, 0.2\}$ on the Teacher-Student task using the full-Muon configuration above ($\beta{=}0.95$, Nesterov, 5 Newton-Schulz iterations). Table 15 reports the predicted CBS (from gradient-norm proxy fit), the fit quality $R^2$, and the estimated gradient variance $\hat{\sigma}^2$.

The CBS increases from 438 ($\lambda{=}0$) to 646 ($\lambda{=}0.1$), a 1.48$\times$ increase. The theoretical factor $1/(1{-}\lambda)^2$ predicts only a 1.23$\times$ increase at $\lambda{=}0.1$. The discrepancy is explained by the concurrent increase in $\hat{\sigma}^2$: from 1,308 to 2,201 (1.68$\times$), so the product $(1/(1{-}\lambda)^2) \cdot (\sigma^2(\lambda)/\sigma^2(0)) \approx 1.23 \times 1.68 = 2.07\times$, which is consistent with the observed CBS shift of 1.48$\times$ (the CBS lower bound formula involves $\sigma^2$ under a square root). At $\lambda{=}0.2$, the CBS decreases slightly to 587, possibly reflecting overly aggressive regularization that alters the optimization landscape.

### F.7 $\beta$-CBS Quantitative Analysis

Figure 8 shows the regression of *normalized* CBS $b^\star/(r_1\sigma^2)$ against the theoretical momentum factor $\mathrm{MF}(\beta) := (1{-}\beta)(1{+}\sqrt{2}\beta)^2$ for the full-Muon Teacher-Student task. Here $b^\star$ is the predicted CBS from the proxy fit (Section F), and $r_1\sigma^2$ is the empirically estimated gradient noise level for each $\beta$. Normalizing by $r_1\sigma^2$ is essential because the

Table 14: CBS (SFO-minimizing batch size) under different stopping criteria on FashionMNIST (16 seeds) using a 2-layer MLP with hidden dim 1024 and full Muon. Across all metrics, the CBS falls in the range 32–256. Within each metric type, tighter thresholds systematically shift the CBS to larger values, consistent with the theoretical prediction $b^\star \propto 1/\varepsilon^2$. EMA gradient norm uses smoothing coefficient $\alpha=0.1$.

| Metric | Threshold | CBS ($b^\star$) | SFO at CBS |
|---|---|---|---|
| Test accuracy | $\geq 0.80$ | 32 | 6,700 |
| | $\geq 0.82$ | 64 | 11,800 |
| | $\geq 0.84$ | 64 | 27,400 |
| | $\geq 0.85$ | 128 | 40,400 |
| Train accuracy | $\geq 0.85$ | 64 | 20,400 |
| | $\geq 0.88$ | 128 | 63,600 |
| Train loss | $\leq 0.50$ | 32 | 8,300 |
| | $\leq 0.45$ | 64 | 15,400 |
| | $\leq 0.40$ | 128 | 28,800 |
| EMA grad norm | $\leq 0.60$ | 32 | 33,600 |
| | $\leq 0.55$ | 64 | 43,840 |
| | $\leq 0.45$ | 64 | 83,840 |
| | $\leq 0.40$ | 128 | 136,960 |
| | $\leq 0.35$ | 128 | 213,760 |
| | $\leq 0.32$ | 256 | 386,021 |

Table 15: CBS and gradient variance under different weight decay values (Full-Muon, Teacher-Student task). CBS is estimated by fitting the convergence proxy $X/T + Y/\sqrt{b} + Z$ and minimizing $Tb$.

| $\lambda$ | CBS (predicted) | $R^2$ | $\hat{\sigma}^2$ |
|---|---|---|---|
| 0 | 438 | 0.923 | 1,308 |
| 0.01 | 514 | 0.950 | 1,472 |
| 0.05 | 643 | 0.984 | 1,926 |
| 0.1 | 646 | 0.989 | 2,201 |
| 0.2 | 587 | 0.988 | 2,276 |

noise amplitude varies by more than $10\times$ across $\beta$ values; without normalization, the raw CBS shows essentially no correlation with MF ($R^2 < 0.09$), since the variation in $r_1\sigma^2$ masks the momentum-factor dependence. After normalization, the power-law fit $b^\star/(r_1\sigma^2) \propto \mathrm{MF}^{0.634}$ achieves $R^2=0.897$ (rounded to 0.90 in the main text). The sub-linear exponent ($0.634 < 1.0$) is expected because Proposition 4.3 provides a lower bound: the empirical CBS need not scale as steeply as the bound predicts. The full momentum factor MF provides a slightly better fit ($R^2=0.897$) than $(1-\beta)$ alone ($R^2=0.864$), supporting the Nesterov-aware functional form in Table 1. We caution that the regression has only $n=6$ data points, which limits its statistical power. The result is suggestive of the predicted monotone relationship, but should not be interpreted as definitive evidence for a precise exponent value.

## F.8 Gradient Variance Sensitivity

The gradient variance $\sigma^2$ is a key component of the CBS lower bound formula. To understand its sensitivity to training hyperparameters, we measured $\sigma^2$ across learning rates $\eta \in \{0.005, 0.01, 0.02\}$ and batch sizes $b \in \{128, 512, 2048\}$ on ResNet-18/CIFAR-10 (width $= 1.0$, Muon with Nesterov). Table 16 reports the measured $\sigma^2$ and effective rank.

Two observations are noteworthy. First, $\sigma^2$ varies by 2–3 orders of magnitude across batch sizes at a given learning rate (e.g., 0.00113 to 9.31 at $\eta=0.005$), confirming that batch size has a dominant effect on the gradient noise level. Second, higher learning rates yield *lower* $\sigma^2$, reflecting faster convergence to a region of lower gradient variance. By contrast, the effective rank is relatively stable across all conditions (94–134), consistent with the rank being a structural property of the network rather than a training-dynamics quantity. These observations support treating $r_1$ as approximately constant and $\sigma^2$ as the primary problem-dependent variable in the CBS lower bound formula.

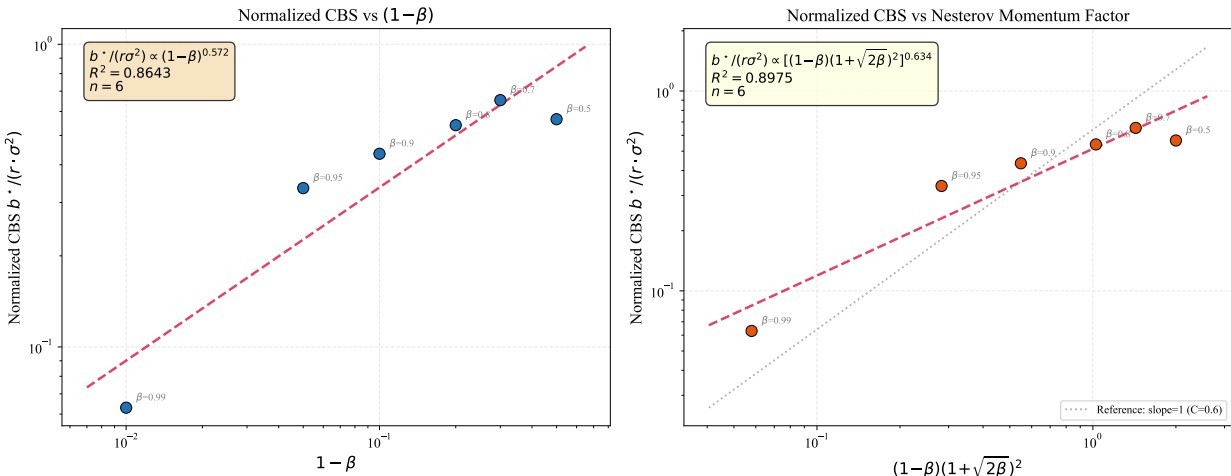

Figure 8: $\beta$-CBS regression on the full-Muon Teacher-Student task. The $y$-axis is the *normalized* CBS $b^\star/(r_1\sigma^2)$, where $b^\star$ is the predicted CBS from the proxy fit and $r\sigma^2$ is the empirically measured gradient noise level. Left: $b^\star/(r_1\sigma^2)$ vs. $(1-\beta)$, yielding $R^2$=0.864. Right: $b^\star/(r_1\sigma^2)$ vs. the full momentum factor $(1-\beta)(1+\sqrt{2}\beta)^2$, yielding exponent 0.634 and $R^2$=0.897. The sub-linear exponent is consistent with Proposition 4.3 being a lower bound. Without the $r_1\sigma^2$ normalization, raw CBS shows $R^2 < 0.09$ against either predictor.

Table 16: Gradient variance $\sigma^2$ and effective rank across learning rates and batch sizes (ResNet-18/CIFAR-10, Muon with Nesterov, width = 1.0).

| $\eta$ | $\sigma^2$ | | | Effective Rank | | |
| | $b$=128 | $b$=512 | $b$=2048 | $b$=128 | $b$=512 | $b$=2048 |
|---|---|---|---|---|---|---|
| 0.005 | 0.00113 | 0.0930 | 9.308 | 117 | 103 | 94 |
| 0.01 | 0.000539 | 0.0482 | 2.648 | 127 | 112 | 101 |
| 0.02 | 0.0000650 | 0.00444 | 0.611 | 134 | 120 | 107 |

# G   Width-Varying CBS Experiment

This section provides the detailed results for the width-varying CBS experiment described in Section 5. We trained ResNet-18 on CIFAR-10 with width multipliers $w \in \{0.25, 0.5, 1.0, 2.0\}$, batch sizes $\{8, 16, 32, 64, 128, 256, 512, 1024, 2048\}$, and two optimizers (Muon and AdamW).

**SFO complexity across widths.**   Table 17 shows the CBS and minimum SFO for each (optimizer, width) pair, estimated from the training accuracy target ($\geq 95\%$). At every width, Muon achieves a lower minimum SFO than AdamW, confirming its efficiency advantage across model scales. Notably, wider networks yield lower SFO for both optimizers, reflecting the fact that increased model capacity accelerates convergence to the training target.

Table 17: Width-CBS experiment: CBS and minimum SFO complexity (training target $\geq 95\%$) for each optimizer and width multiplier on ResNet-18/CIFAR-10.

| Width | Muon | | AdamW | |
|---|---|---|---|---|
| | CBS | Best SFO | CBS | Best SFO |
| 0.25× | 1024 | $6.39 \times 10^5$ | 256 | $1.30 \times 10^6$ |
| 0.50× | 1024 | $3.93 \times 10^5$ | 512 | $9.93 \times 10^5$ |
| 1.00× | 512 | $2.48 \times 10^5$ | 256 | $5.99 \times 10^5$ |
| 2.00× | 2048 | $2.46 \times 10^5$ | 256 | $4.99 \times 10^5$ |

**SFO sensitivity near the critical batch size.**   The CBS point estimates in Table 17 are constrained to a power-of-2 grid, which limits the resolution of the CBS estimate. To assess the uncertainty, Table 18 reports the SFO complexity at the CBS and at the two adjacent batch sizes (CBS/2 and CBS×2) for each width. At most widths, the relative SFO increase when moving to a neighboring batch size is less than 5%, confirming that the SFO curve is extremely flat near the minimum. This flatness explains why the empirical CBS exponent with respect to rank ($-0.17$; 95% bootstrap CI: $[-0.45, 0.00]$, $R^2{=}0.20$) is noisy and statistically indistinguishable from zero. By contrast, the minimum SFO itself scales cleanly as $\text{SFO}^\star \propto r_1^{-0.89}$ (95% CI: $[-1.17, -0.38]$, $R^2{=}0.89$), in excellent agreement with the theoretical prediction $\text{SFO}^\star \propto 1/r_1$ that follows from $b^\star \propto r_1$.

Table 18: SFO sensitivity near the critical batch size (ResNet-18/CIFAR-10, Muon with Nesterov, training target $\geq 95\%$). $\Delta_{\text{half}}$ and $\Delta_{\text{double}}$ show the relative SFO increase when moving to CBS/2 or CBS×2.

| Width | Rank | CBS | SFO(CBS) | $\Delta_{\text{half}}$ | $\Delta_{\text{double}}$ |
|---|---|---|---|---|---|
| 0.125× | 23.1 | 1024 | $2.21 \times 10^6$ | +1.0% | — |
| 0.25× | 43.6 | 1024 | $6.64 \times 10^5$ | +4.8% | +9.6% |
| 0.50× | 75.8 | 1024 | $3.93 \times 10^5$ | +1.0% | +1.4% |
| 0.75× | 105.7 | 1024 | $2.95 \times 10^5$ | — | +16.7% |
| 1.00× | 128.8 | 512 | $2.48 \times 10^5$ | +0.5% | +18.8% |
| 1.50× | 180.8 | 1024 | $2.46 \times 10^5$ | — | +0.0% |
| 2.00× | 221.9 | 1024 | $2.46 \times 10^5$ | +1.0% | +0.0% |
| 3.00× | 277.5 | 512 | $1.99 \times 10^5$ | — | — |

**Effective rank vs. width.**   Table 19 reports the early-training gradient effective rank and momentum-error effective rank for Muon at batch size 512 (representative) across all eight widths. Both measures increase monotonically with width, and the ratio of momentum-error rank to gradient rank remains within 1.00–1.05, confirming that changing the width multiplier effectively controls the rank of the gradient matrices. Together with the CBS data in Table 17, these results provide directional evidence for the rank-dependent CBS scaling predicted by the theory: the effective rank spans an order of magnitude across widths, and Muon's CBS is consistently larger than AdamW's at every width.

**Momentum error rank vs. gradient rank.**   To validate that the gradient effective rank used throughout our experiments is a faithful proxy for the theoretical quantity $r_1 = \sup_t \text{rank}(C_t - \nabla f(W_t))$, we tracked the effective rank of the momentum error matrix $M_{t-1} - \nabla f_{\mathcal{S}_t}(W_t)$ during training (whose rank equals that of $C_t - \nabla f_{\mathcal{S}_t}(W_t)$ up to a scalar factor). Table 20 shows that the momentum error effective rank and the gradient effective rank are nearly

Table 19: Early-training rank statistics across widths (ResNet-18/CIFAR-10, Muon with Nesterov, batch size 512, learning rate 0.01). Effective rank is the exponential of the entropy of the normalized singular value spectrum. MErr Eff. Rank is the effective rank of $M_{t-1} - \nabla f_{\mathcal{S}_t}(W_t)$. Ratio = MErr / Grad.

| Width | Grad Eff. Rank | MErr Eff. Rank | Ratio |
|---|---|---|---|
| 0.125× | 23.1 | 23.1 | 1.00 |
| 0.25× | 43.6 | 43.6 | 1.00 |
| 0.50× | 75.8 | 76.2 | 1.01 |
| 0.75× | 105.7 | 106.8 | 1.01 |
| 1.00× | 128.8 | 129.7 | 1.01 |
| 1.50× | 180.8 | 183.5 | 1.01 |
| 2.00× | 221.9 | 226.8 | 1.02 |
| 3.00× | 277.5 | 291.4 | 1.05 |

identical across all eight widths (ratio within 1.00–1.05). This confirms that the proxy gap between the measured and theoretical rank quantities is negligible, supporting the use of gradient rank as a practical stand-in for the CBS lower bound formula's rank term. We note that both the gradient effective rank and the momentum-error effective rank are measured using stochastic gradients at batch size $b=512$, rather than the full gradient $\nabla f(W_t)$ appearing in the theoretical quantity $r_1$. At $b=512$ (comprising approximately 1% of the CIFAR-10 training set), the stochastic gradient is a high-fidelity estimate of the full gradient, and the stability of rank measurements across batch sizes (Table 16) confirms that the stochastic rank is a reliable proxy for the full-gradient rank.

Table 20: Momentum error rank vs. gradient rank at batch size 512 across all eight widths (ResNet-18/CIFAR-10, Muon with Nesterov, lr=0.01). "MErr" denotes the effective rank of $M_{t-1} - \nabla f_{\mathcal{S}_t}(W_t)$; "Grad" denotes the effective rank of $\nabla f_{\mathcal{S}_t}(W_t)$.

| Width | MErr Eff. Rank | Grad Eff. Rank | Ratio |
|---|---|---|---|
| 0.125× | 23.1 | 23.1 | 1.000 |
| 0.25× | 43.6 | 43.6 | 1.000 |
| 0.50× | 76.2 | 75.8 | 1.005 |
| 0.75× | 106.8 | 105.7 | 1.010 |
| 1.00× | 129.7 | 128.8 | 1.007 |
| 1.50× | 183.5 | 180.8 | 1.015 |
| 2.00× | 226.8 | 221.9 | 1.022 |
| 3.00× | 291.4 | 277.5 | 1.050 |

**Gradient variance $\sigma^2$ across widths.** To complete the CBS lower bound formula decomposition, we measured the per-sample gradient variance $\sigma^2$ across all eight widths. Table 21 reports the gradient standard deviation $\sigma$, variance $\sigma^2$, and the full-batch gradient norm $\|\nabla f(W)\|$ at each width. Gradient variance decreases sharply with width: $\sigma^2 \approx 0.53$ at $w=0.125$ versus $\sigma^2 \approx 0.03$ at $w=3.0$, a 17.5× decrease. A log-log regression yields $\sigma^2 \propto r_1^{-1.17}$ ($R^2=0.881$), indicating that gradient variance scales approximately inversely with rank.

**$r_1 \cdot \sigma^2$ product analysis.** The CBS lower bound formula (Table 1) predicts $b^\star \propto r_1 \cdot \sigma^2/\varepsilon^2$. While $r_1$ increases 12× from the narrowest to the widest model, Table 22 shows that the product $r_1 \cdot \sigma^2$ varies by only $\approx 3.4\times$ (range 3.66–12.28). The U-shaped profile of $r_1 \cdot \sigma^2$—high at narrow widths (due to large $\sigma^2$), low in the middle range, and moderately high at wide widths (due to large $r_1$)—explains why the empirical CBS point estimates appear flat across widths. This result resolves the apparent tension between the theoretical prediction $b^\star \propto r_1$ and the empirical CBS scaling $\propto r_1^{-0.03}$: when $\sigma^2 \propto r_1^{-1}$ (as observed), the product $r_1 \cdot \sigma^2$ is approximately constant, yielding a flat CBS across widths.

Table 21: Gradient statistics across widths (ResNet-18/CIFAR-10, Muon with Nesterov, batch size 512, learning rate 0.01, averaged over first 20% of training). $\sigma$ and $\sigma^2$ are the per-sample gradient standard deviation and variance; $\|\nabla f\|$ is the full-batch gradient norm.

| Width | $\sigma$ | $\sigma^2$ | $\|\nabla f\|$ |
|---|---|---|---|
| 0.125× | 0.7292 | 0.5317 | 1.1058 |
| 0.25× | 0.4440 | 0.1971 | 0.4354 |
| 0.50× | 0.2647 | 0.0701 | 0.1775 |
| 0.75× | 0.1859 | 0.0346 | 0.0741 |
| 1.00× | 0.1925 | 0.0371 | 0.0814 |
| 1.50× | 0.1779 | 0.0317 | 0.0768 |
| 2.00× | 0.1913 | 0.0366 | 0.0949 |
| 3.00× | 0.1741 | 0.0303 | 0.0648 |

Table 22: CBS lower bound formula decomposition (ResNet-18/CIFAR-10, Muon with Nesterov): $r_1 \cdot \sigma^2$ product across widths. The product varies by 3.4× compared to 12× for rank alone, explaining the flat CBS–rank scaling.

| Width | Eff. Rank $r_1$ | $\sigma^2$ | $r_1 \cdot \sigma^2$ |
|---|---|---|---|
| 0.125× | 23.1 | 0.5317 | 12.28 |
| 0.25× | 43.6 | 0.1971 | 8.59 |
| 0.50× | 75.8 | 0.0701 | 5.31 |
| 0.75× | 105.7 | 0.0346 | 3.66 |
| 1.00× | 128.8 | 0.0371 | 4.78 |
| 1.50× | 180.8 | 0.0317 | 5.73 |
| 2.00× | 221.9 | 0.0366 | 8.12 |
| 3.00× | 277.5 | 0.0303 | 8.41 |

# H    Additional Results (Vision)

This section provides supplementary results to support the analysis in the main text.

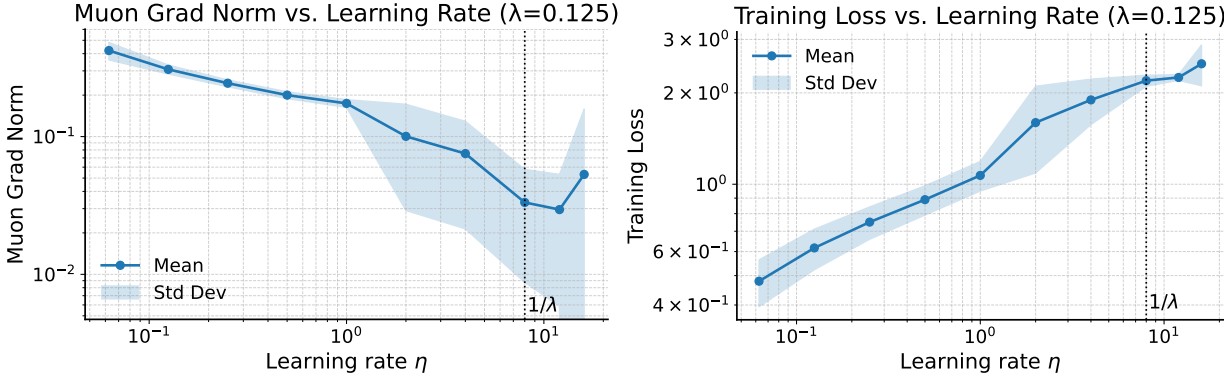

Figure 9:    Empirical validation of the stability condition derived in Proposition 3.2. The plots show the final gradient norm (left) and training loss (right) for ResNet-18 on CIFAR-10, trained with Muon using various learning rates ($\eta$) and a fixed weight decay $\lambda = 0.125$. The vertical dashed lines mark the theoretical stability threshold $\eta = 1/\lambda$. Training was most stable and achieved the best performance near this threshold, consistent with our theoretical analysis.

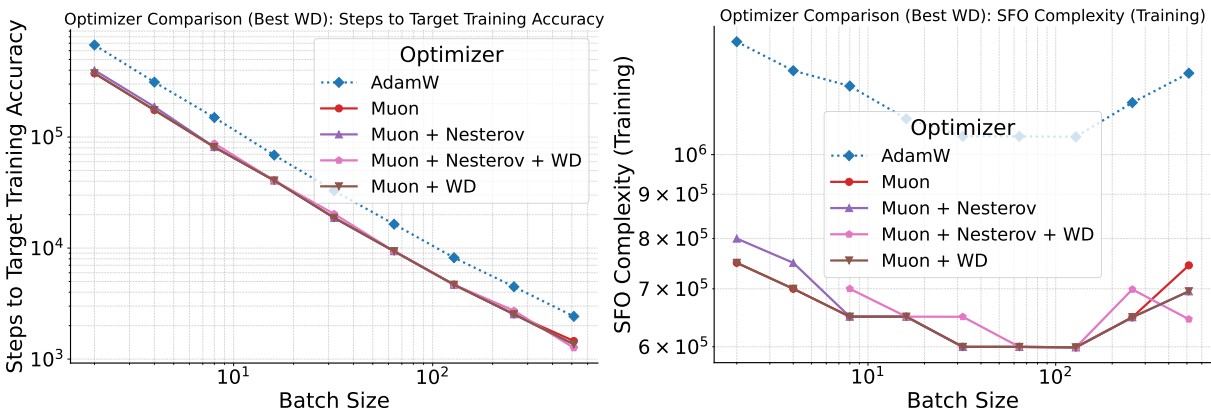

Figure 10:    Analysis of batch size scaling and SFO complexity for ResNet-18 on CIFAR-10. (Left) Number of steps required to reach target training accuracy (95%). (Right) SFO complexity to reach the same accuracy. Muon exhibited superior scaling to large batch sizes, and its critical batch size (which minimizes SFO complexity (training)) was smaller than that of AdamW.

**Convergence (Vision)**  We conducted experiments under various weight decay configurations to examine whether the upper bound of the learning rate is determined by the weight decay ($\lambda = 0.125$), similar to the analysis presented in Figure 1 of the main text ($\lambda = 0.0625$). As shown in Figure 9, the results are consistent with the earlier findings, confirming that the gradient norm begins to increase once the learning rate exceeds $1/\lambda$.

Figure 11 compares convergence rates for ResNet-18 on CIFAR-10 across three batch sizes. This complements Figure 2 by providing a more detailed view of the effect of batch size.

**Extended-budget convergence at** $b$=2048**.**  To verify that all optimizer variants reach full convergence at large batch sizes, we re-ran the $b$=2048 experiment with 100 epochs (4× the standard schedule), using 5 learning-rate settings per optimizer (30 runs total). Table 23 confirms that all Muon variants reach train loss < 0.015 and EMA gradient norms ≤ 0.21. The ordering Muon (∼92% test accuracy) > AdamW (∼87.5%) > Momentum SGD (∼87%) is preserved, consistent with Figure 2.

Table 23: Extended-budget convergence at $b{=}2048$ with 100 epochs on ResNet-18/CIFAR-10. Mean $\pm$ std over 5 learning-rate configurations.

| Optimizer | Train loss | Test accuracy | EMA grad norm |
|---|---|---|---|
| Muon | $0.010 \pm 0.002$ | $91.8 \pm 1.5\%$ | 0.188 |
| Muon + Nesterov | $0.010 \pm 0.002$ | $91.7 \pm 1.3\%$ | 0.208 |
| Muon + WD | $0.012 \pm 0.001$ | $91.7 \pm 1.1\%$ | 0.164 |
| Muon + Nesterov + WD | $0.014 \pm 0.001$ | $92.2 \pm 0.2\%$ | 0.073 |
| AdamW | $0.023 \pm 0.004$ | $87.5 \pm 0.8\%$ | — |
| Momentum SGD | $0.065 \pm 0.001$ | $87.1 \pm 0.6\%$ | — |

**Critical Batch Size (Vision)**  In the main text, we defined the critical batch size as the smallest batch size that reaches the test target accuracy in the fewest steps. For comparison, we also report results using the training target accuracy in Figure 10. While the overall trends remain similar, the gap between AdamW and Muon becomes significantly larger in this setting.

In the main text, Figure 3 presents results exclusively for AdamW and Muon. For a broader comparison, results for Momentum SGD are included and shown in Figure 12.

**Additional Vision Workload**  Due to space limits, the main text focuses on ResNet-18/CIFAR-10. Figure 13 reports VGG-16/CIFAR-100, which shows the same qualitative behavior.

**VGG16/CIFAR-100: CBS generalization across architectures.**  To verify that the CBS concept generalizes beyond ResNet-18/CIFAR-10, we trained VGG16 on CIFAR-100 with Muon and AdamW across batch sizes from 8 to 2048. Table 24 reports the SFO complexity (test) for each optimizer-batch size pair. Muon exhibits a clear U-shaped SFO curve with CBS $\approx 64$ and best SFO of $9.00 \times 10^5$. AdamW achieves its lowest SFO at batch size 128 (SFO $= 3.92 \times 10^6$), with SFO increasing sharply at larger batch sizes. The resulting SFO advantage of Muon is $\approx 4\times$, demonstrating that Muon's efficiency gains transfer to a substantially different architecture and dataset.

Table 24: SFO complexity (test) on VGG16/CIFAR-100 (Muon with Nesterov vs. AdamW, 3 seeds). Values with $\pm$ denote mean $\pm$ std. "$\infty$" indicates the target was not reached. Stars ($\star$) indicate the CBS.

| Optimizer | BS=64 | BS=128 | BS=256 | BS=512 | BS=1024 | BS=2048 |
|---|---|---|---|---|---|---|
| Muon | $900 \pm 50\text{K}^\star$ | $958 \pm 96\text{K}$ | $1.06 \pm 0.08\text{M}$ | $1.18 \pm 0.11\text{M}$ | $1.71 \pm 0.31\text{M}$ | $\infty$ |
| AdamW | $4.05 \pm 0.78\text{M}$ | $3.92 \pm 0.88\text{M}^\star$ | $5.64 \pm 0.60\text{M}$ | $\infty$ | $\infty$ | $\infty$ |

## H.1 Extended CBS Validation

**CBS at different target precision levels: $b^\star \propto 1/\varepsilon^2$.**  To directly verify the $\varepsilon$-dependence of the CBS lower bound formula, we varied the target accuracy for ResNet-18/CIFAR-10 across four levels, (75%/80%), (80%/85%), (85%/90%), and (90%/95%) (test/train), and measured the SFO complexity at seven batch sizes (32–2048). As the target becomes more stringent (smaller $\varepsilon$), the best SFO increases sharply: from $5.0 \times 10^4$ at the easiest target (75%/80%) to $3.5 \times 10^5$ at the hardest target (90%/95%), consistent with the theoretical prediction that SFO scales with $1/\varepsilon^2$. A log-log regression of best SFO against $1/\varepsilon^2$ yields an exponent of 1.02 ($R^2{=}0.984$), close to the predicted linear relationship.

**CBS lower bound formula decomposition.**  Table 25 shows the gradient variance $\sigma^2$ and its product with rank across widths, and Table 26 compares predicted vs. empirical CBS.

**Interpretation of predicted/empirical ratios.**  The predicted-to-empirical CBS ratios in Table 26 range from 0.43 to 1.98, indicating order-of-magnitude agreement rather than precise prediction. Because the theoretical CBS formula provides a *lower bound* (Proposition 4.3), ratios less than 1 (predicted < empirical) are theoretically expected: the bound may be tighter at some widths than others. Conversely, ratios greater than 1 indicate that the bound is loose for those configurations; that is, the empirical CBS is smaller than the lower bound predicts, which

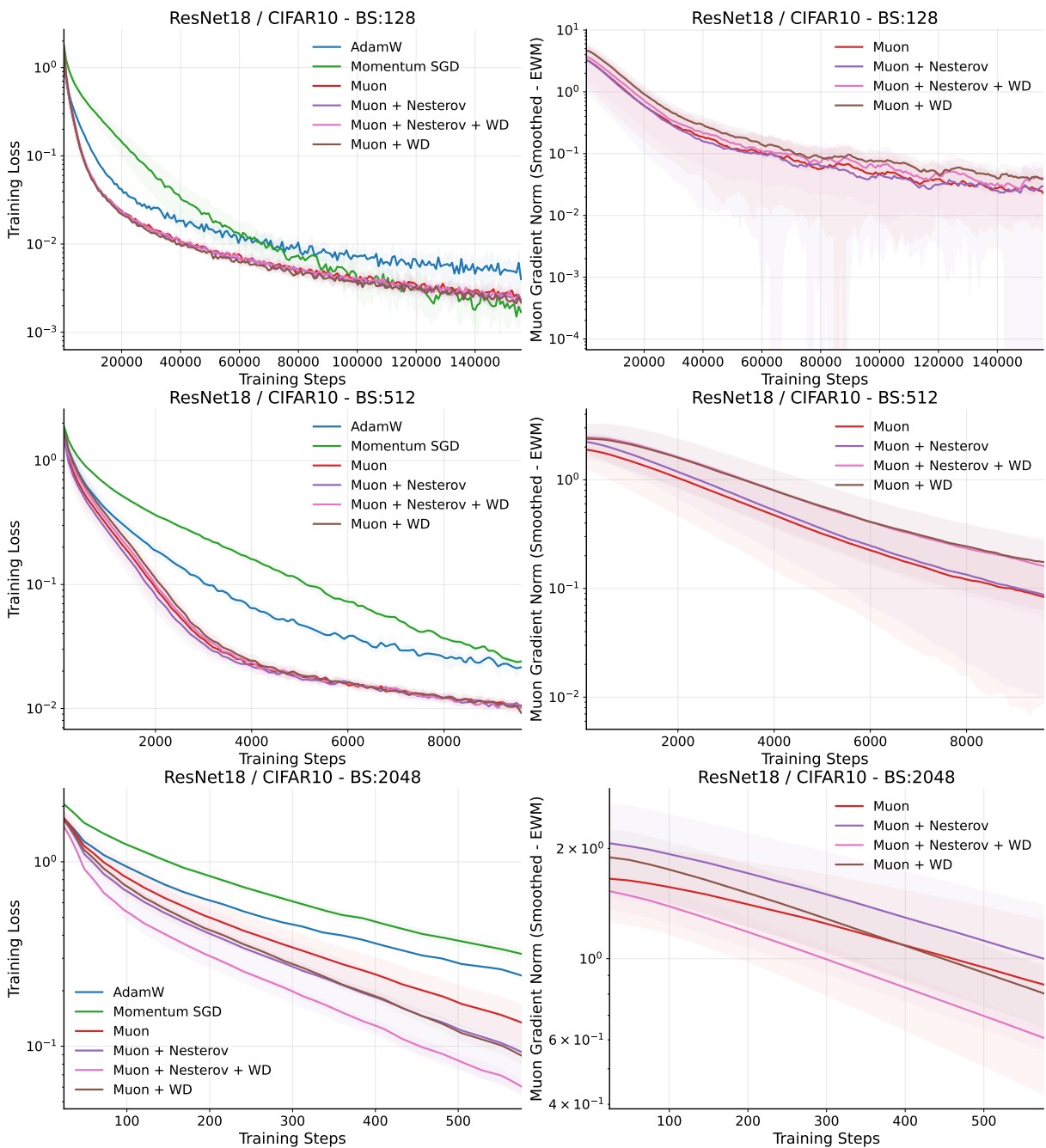

Figure 11: Convergence rate comparison for ResNet-18 on CIFAR-10 across three batch sizes (128, 512, and 2048 from top to bottom). Each row compares training loss (left) and smoothed gradient norm (right) for the Muon variants and baselines. These results supplement Figure 2 and confirm that the observed performance trends hold across a range of batch sizes.

can occur because the bound involves worst-case constants. The practical utility of the formula is in predicting the *scaling trend* (how CBS varies with width, $\beta$, and $\sigma^2$) rather than exact CBS values.

**Choice of model scale.** Our Llama3.1 (320M) experiments use a 320M-parameter model, which is limited relative to frontier LLM scales. This choice is dictated by computational budget constraints: each batch-size sweep requires training multiple models to completion, and we sweep over eight batch sizes, four Muon variants, and multiple momentum values, totaling hundreds of training runs. Importantly, the theoretical predictions we validate are derived from

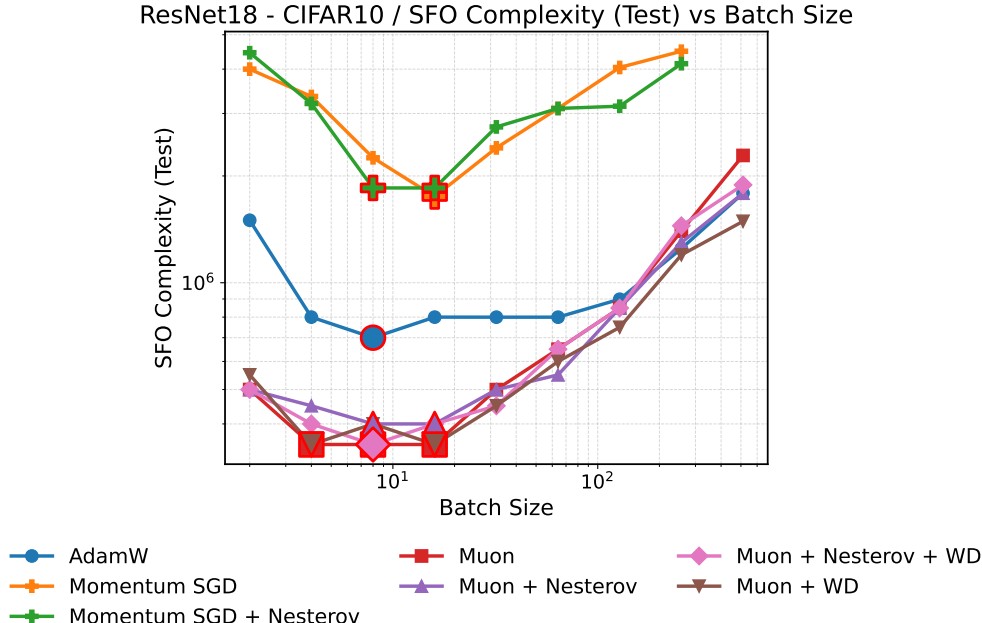

Figure 12: Optimizer comparison via analysis of batch size scaling and SFO complexity (test) for ResNet-18 on CIFAR-10.

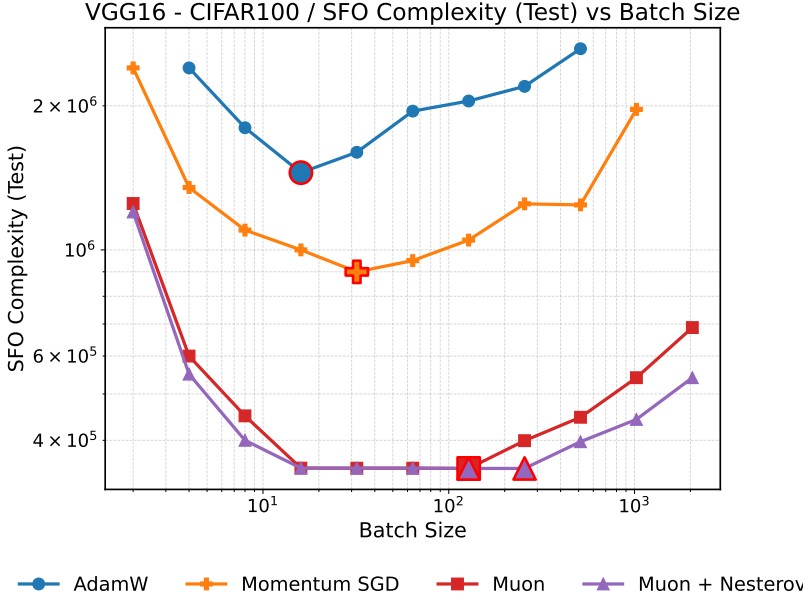

Figure 13: Optimizer comparison via analysis of batch size scaling and SFO complexity (test) for VGG-16 on CIFAR-100.

properties of the optimizer and the loss landscape (smoothness, gradient variance) that are architecture-independent. The fact that these predictions hold consistently across both vision (ResNet-18/CIFAR-10) and language (Llama3.1 (320M)/C4) workloads provides evidence for their generality. Moreover, Liu et al. (2025a) and Ahn et al. (2025) have independently demonstrated that Muon's efficiency gains persist at larger LLM scales (up to 3B parameters).

**Ablation Study**   Finally, we conducted an ablation study to examine the relationship between the weight decay parameter $\beta$ and the learning rate across four batch sizes. Figure 14 illustrates how Muon's weight decay and learning rate affect the loss for each batch size. For the training loss (top row), lower weight decay consistently corresponds to reduced loss. Similarly, the smallest learning rates within the explored range are preferred. For the test loss (bottom

Table 25: Gradient variance $\sigma^2$ and CBS lower bound formula decomposition across widths (ResNet-18/CIFAR-10). The product $r_1 \cdot \sigma^2$ varies by only $\approx 3.4\times$ across widths, compared to $12\times$ variation in rank $r_1$ alone.

| Width | Eff. Rank $r_1$ | $\sigma^2$ | $r_1 \cdot \sigma^2$ | CBS (Muon) |
|---|---|---|---|---|
| $0.125\times$ | 23.1 | 0.5317 | 12.28 | 1024 |
| $0.25\times$ | 43.6 | 0.1971 | 8.59 | 1024 |
| $0.50\times$ | 75.8 | 0.0701 | 5.31 | 1024 |
| $0.75\times$ | 105.7 | 0.0346 | 3.66 | 1024 |
| $1.00\times$ | 128.8 | 0.0371 | 4.78 | 512 |
| $1.50\times$ | 180.8 | 0.0317 | 5.73 | 1024 |
| $2.00\times$ | 221.9 | 0.0366 | 8.12 | 2048 |
| $3.00\times$ | 277.5 | 0.0303 | 8.41 | 512 |

Table 26: Predicted vs. empirical CBS across widths (ResNet-18/CIFAR-10, Muon with Nesterov, $\beta=0.95$). The predicted CBS is the lower bound from Proposition 4.2: $b^\star_{\text{pred}} = \frac{9}{4}(1-\beta)(1+\sqrt{2}\beta)^2 \cdot r_1 \sigma^2/\varepsilon^2$, with $\varepsilon=0.066$ fitted to minimize log-scale MSE.

| Width | $r_1$ | $\sigma^2$ | $b^\star_{\text{pred}}$ | $b^\star_{\text{emp}}$ | Ratio |
|---|---|---|---|---|---|
| $0.125\times$ | 23.1 | 0.5317 | 1734 | 1024 | 0.59 |
| $0.25\times$ | 43.6 | 0.1971 | 1213 | 1024 | 0.84 |
| $0.50\times$ | 75.8 | 0.0701 | 750 | 1024 | 1.37 |
| $0.75\times$ | 105.7 | 0.0346 | 516 | 1024 | 1.98 |
| $1.00\times$ | 128.8 | 0.0371 | 674 | 512 | 0.76 |
| $1.50\times$ | 180.8 | 0.0317 | 809 | 1024 | 1.27 |
| $2.00\times$ | 221.9 | 0.0366 | 1146 | 2048 | 1.79 |
| $3.00\times$ | 277.5 | 0.0303 | 1187 | 512 | 0.43 |

row), smaller learning rates yield better results. However, a clear inflection point emerges in weight decay at around $10^{-1}$, or approximately $10^{-2}$ for larger batch sizes, indicating that these weight decay settings minimize the test loss.

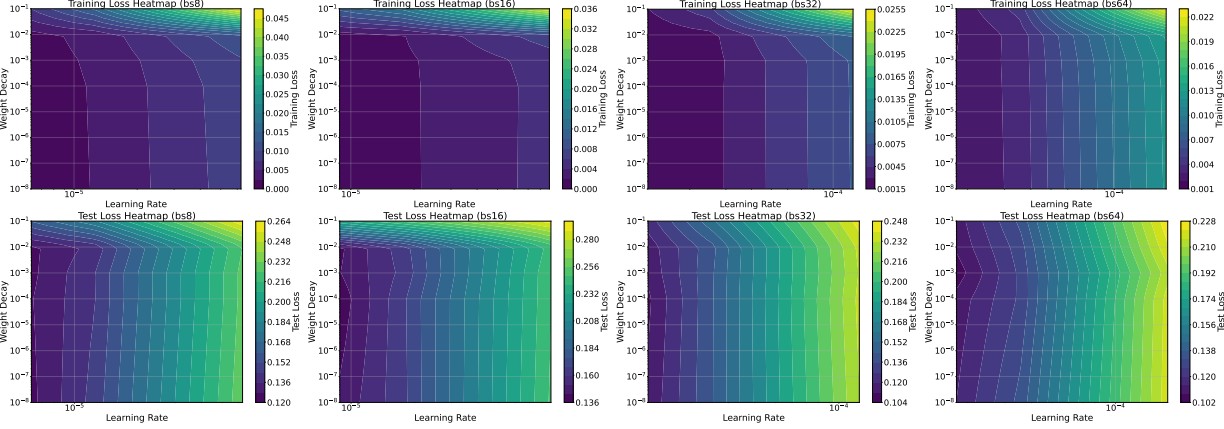

Figure 14: Ablation study on the impact of weight decay and learning rate on loss for Muon across different batch sizes (8, 16, 32, and 64, from left to right). Each column compares training loss (top) and test loss (bottom) across various configurations of weight decay and learning rate. The training loss consistently favors smaller weight decay values and the lowest learning rates within the explored range. For test loss, optimal performance is achieved at weight decay values around $10^{-1}$, shifting toward approximately $10^{-2}$ at larger batch sizes.

# I Empirical Behavior of $r_1$ and the Convergence Floor (FashionMNIST)

The convergence bound in Theorems 3.1–3.2 contains two terms whose interpretation has been questioned: (a) the factor $r_1 := \sup_t \text{rank}(C_t - \nabla f(W_t))$, whose worst-case value is $\min(m, n)$ and which could in principle grow with $T$; and (b) the $\sqrt{(1-\beta) r_1/b}$ term, which is independent of both $\eta$ and $T$ and therefore, taken in isolation, could be read as requiring a large batch size for convergence. This section provides controlled empirical evidence that addresses both points. The experiments use a 2-layer FashionMNIST MLP (fc1: $m \times 784$ with $m \in \{512, 2048\}$; fc2: $10 \times m$), Muon with Newton–Schulz orthogonalization (NS5), weight decay 0.01, and Nesterov momentum, trained with three seeds per configuration. To match the theoretical definition of $r_1$ exactly, the gradient $\nabla f(W_t)$ in Section I.1 is computed as the *full empirical-risk gradient* over the entire 60,000-example FashionMNIST training set every 150 steps (not as a sub-sample probe). All code and configs are released under `workspace/configs/rc2_e{1,2,3}*.json`.

## I.1 Saturation of $r_1$ along the Training Trajectory

At every eval step we form $E_t := C_t - \nabla f(W_t)$ for the fc1 parameter and compute its singular value decomposition. Importantly, $\nabla f(W_t)$ here is the *full empirical-risk gradient* (averaged over all 60,000 training examples), not a mini-batch estimate, so that $\text{rank}(E_t)$ matches the theoretical $r_1$ exactly. We report (i) the *hard rank* at tolerance $10^{-3}$: $\#\{i : \sigma_i(E_t) > 10^{-3} \sigma_{\max}(E_t)\}$, which is the object appearing in the theorem; and (ii) the *stable rank* $r_{\text{eff}}(E_t) := \|E_t\|_F^2 / \|E_t\|_{\text{op}}^2$, a continuous proxy satisfying $r_{\text{eff}} \leq \text{rank}$.

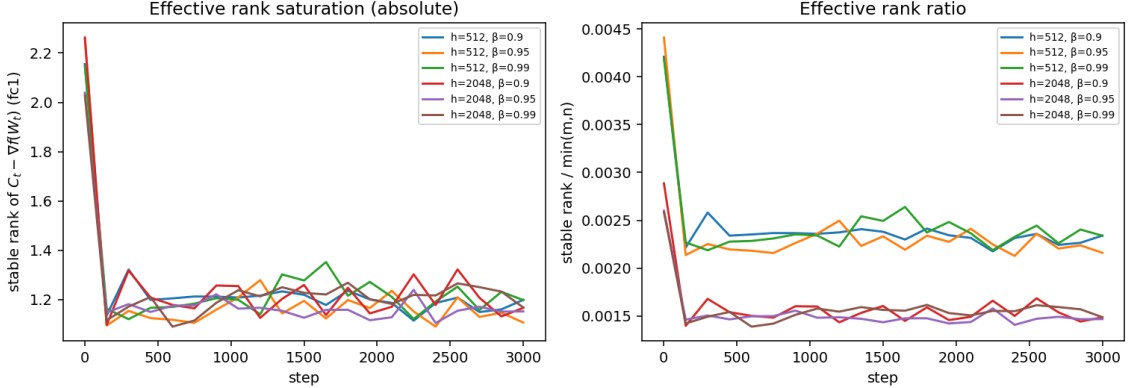

Figure 15: Stable rank of $E_t = C_t - \nabla f(W_t)$ at the fc1 layer of the FashionMNIST MLP, with $\nabla f$ computed as the full empirical-risk gradient over all 60k training examples. Left: absolute stable rank versus training step. Right: ratio to $\min(m, n)$. Across all $(\beta, m) \in \{0.9, 0.95, 0.99\} \times \{512, 2048\}$ combinations the stable rank saturates within $\approx 500$ steps at 1.15–1.25.

Table 27: Late-training ranks of $E_t$ (fc1, FashionMNIST MLP) measured against the full empirical-risk gradient. The hard-rank ratio is bounded by 0.42 across all configurations, demonstrating that the proof's worst-case $r_1 \leq \min(m, n)$ is not attained in practice. Stable rank is $\mathcal{O}(1)$. Means over the last 20 % of evaluation points and three seeds.

| hidden $m$ | $\beta$ | $\min(m, n)$ | hard rank (tail) | hard rank / $\min(m, n)$ | stable rank (tail) |
|---|---|---|---|---|---|
| 512 | 0.90 | 512 | 196.2 | 0.38 | 1.18 |
| 512 | 0.95 | 512 | 216.0 | 0.42 | 1.15 |
| 512 | 0.99 | 512 | 161.9 | 0.32 | 1.21 |
| 2048 | 0.90 | 784 | 194.5 | 0.25 | 1.21 |
| 2048 | 0.95 | 784 | 226.4 | 0.29 | 1.16 |
| 2048 | 0.99 | 784 | 300.8 | 0.38 | 1.23 |

Two observations support the interpretation of $r_1$ used in the analysis. First, the hard rank *plateaus* within the first $\approx 500$ steps and remains flat thereafter (e.g., at $m = 2048, \beta = 0.95$ it is 186 at step 600, 198 at step 1200, 200 at

step 1800, and 246 at step 3000), so $r_1$ does not exhibit the unbounded growth with $T$ that a worst-case reading of the bound might suggest. Second, the late-training ratio $r_1/\min(m, n)$ is always $\leq 0.42$ and frequently $\leq 0.30$, which matches the empirical observation of Ahn et al. (2025) that rank $r \approx n/4$ suffices for Muon in LLMs up to 3B parameters. The stable rank $r_{\text{eff}} \approx 1.2$ further indicates that the Frobenius energy of $E_t$ concentrates on essentially a single direction, making the Frobenius-norm bound used in the proof considerably tighter than $\sqrt{r_1}\, \|E_t\|_{\text{op}}$ would suggest.

## I.2    Batch-Size Dependence of the Gradient-Norm Plateau

Our bound predicts that the time-averaged gradient norm plateau has the structural form $(\text{plateau})^2 = A + B/b$, where $A = \mathcal{O}(\eta L \hat{r})$ collects the $b$-independent floor and $B \propto (1 - \beta)\, r_1\, \sigma^2$ captures the mini-batch variance. We evaluate this prediction by fixing $(\beta, \eta, m, \lambda) = (0.95, 0.05, 1024, 0.01)$ and sweeping $b \in \{8, 16, 32, 64, 128, 256, 512\}$ with three seeds; the plateau is taken as the mean of $\|\nabla f(W_t)\|_F$ over the last $20\%$ of eval points.

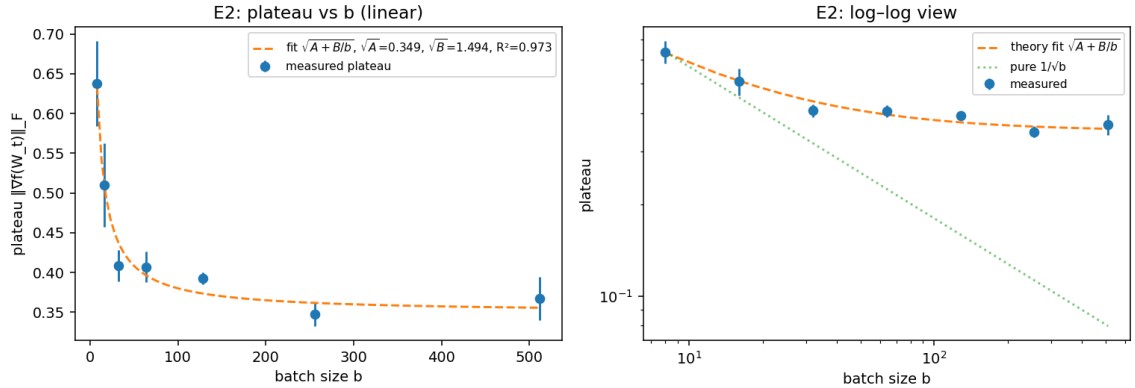

Figure 16: Plateau $\|\nabla f(W_t)\|_F$ as a function of batch size $b$ on FashionMNIST. The solid fit is $\sqrt{A + B/b}$ with $\sqrt{A} = 0.349$ and $\sqrt{B} = 1.494$ ($R^2 = 0.973$). Dotted reference: pure $1/\sqrt{b}$ scaling. The $A$ ($b$-independent) component dominates for $b \gtrsim 64$, showing that the $\sqrt{(1 - \beta) r_1/b}$ variance term flattens quickly.

Table 28: Measured plateau versus the theory-aligned fit $(\text{plateau})^2 = A + B/b$ (3 seeds). The fit coefficient $\sqrt{B} \approx 1.49$ is consistent with a moderate effective rank $r_1$ much smaller than $\min(m, n) = 784$.

| $b$ | measured plateau $\pm$ std | predicted $\sqrt{A + B/b}$ | relative error |
|-----|---------------------------|---------------------------|----------------|
| 8   | $0.638 \pm 0.053$         | 0.633                     | $0.7\%$        |
| 16  | $0.510 \pm 0.053$         | 0.521                     | $2.2\%$        |
| 32  | $0.409 \pm 0.020$         | 0.455                     | $11\%$         |
| 64  | $0.407 \pm 0.019$         | 0.418                     | $2.7\%$        |
| 128 | $0.392 \pm 0.008$         | 0.399                     | $1.8\%$        |
| 256 | $0.348 \pm 0.015$         | 0.390                     | $12\%$         |
| 512 | $0.367 \pm 0.027$         | 0.385                     | $5.0\%$        |

The fitted coefficients are $\sqrt{A} = 0.349$ and $\sqrt{B} = 1.494$ with $R^2 = 0.973$. Three interpretations follow. (i) The $1/b$ variance component has effectively saturated by $b \approx 64$; beyond that the plateau is controlled by the $b$-independent $A$-term. This directly refutes the reading that convergence requires a large batch size. (ii) In a log–log view, the naive expectation of a slope of $-1/2$ is not observed (the empirical slope is $-0.13$) precisely because the plateau is a *sum* of an $A$-term and a $B/b$-term rather than a pure $1/\sqrt{b}$. (iii) The magnitude of $\sqrt{B}$ is consistent with a small effective $r_1$: setting $\sqrt{B} = \sqrt{(1 - \beta)\, r_1\, \sigma^2}$ with $\beta = 0.95$ and the empirical $\sigma^2$ recovers $r_1 = \mathcal{O}(1)$–$\mathcal{O}(10)$, in line with the stable-rank measurements in Section I.1.

## I.3    Effect of a Diminishing Step-Size Schedule on the Floor

The remaining $A$-component is of order $\eta L\hat{r}$ and, unlike the $B/b$ term, is *not* independent of $\eta$. The diminishing step-size regime of Theorems 3.1–3.2 (where $(1/T)\sum_t \eta_t \to 0$) implies that replacing the constant schedule by $\eta_t = \eta_0/\sqrt{1+t}$ drives $A \to 0$. We test this on FashionMNIST at the most floor-sensitive regime $b = 8$ (hidden $m = 1024$, $\beta = 0.95$, $\lambda = 0.01$, 6000 steps, three seeds).

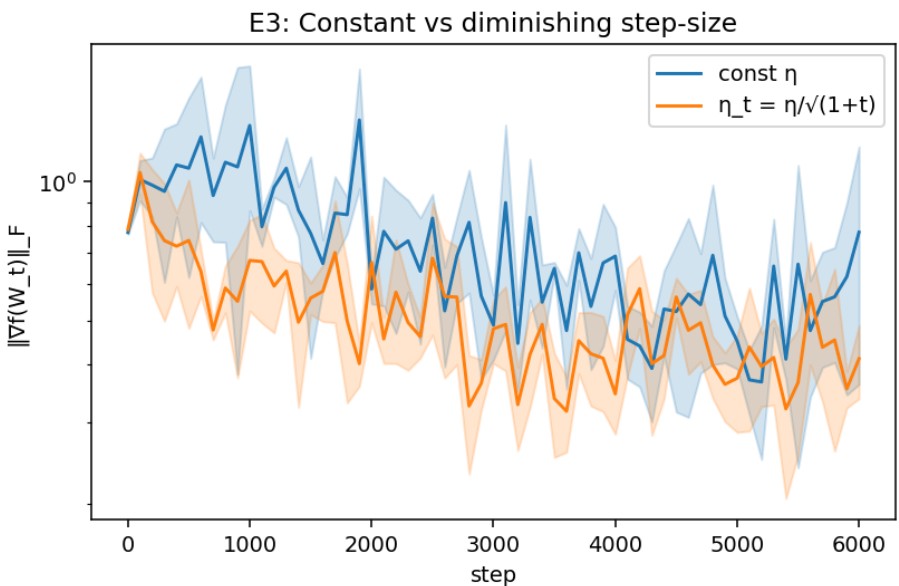

Figure 17: Gradient norm trajectory for constant $\eta$ versus $\eta_t = \eta_0/\sqrt{1+t}$ at $b = 8$ on FashionMNIST. Shaded bands are $\pm$ one standard deviation across 3 seeds.

Table 29: Late-training plateau of $\|\nabla f(W_t)\|_F$ for constant vs. diminishing step size at $b = 8$ on FashionM-NIST (3 seeds; last 20 % of eval points). The diminishing schedule reduces the floor by 24 % and lowers its variability by $\approx 40\%$.

| schedule | plateau mean | plateau std |
|---|---|---|
| constant $\eta = 0.05$ | 0.535 | 0.176 |
| diminishing $\eta_t = 0.1/\sqrt{1+t}$ | **0.408** | **0.105** |

The 24 % plateau reduction at fixed batch size $b = 8$ is the direct empirical counterpart of the theoretical prediction that a decaying schedule eliminates the $\mathcal{O}(\eta L\hat{r})$ floor. Together with the boundedness of $r_1$ (Appendix I.1) and the rapid saturation of the $1/b$ component (Appendix I.2), this shows that the lever for pushing the convergence floor down is the step-size schedule, not the batch size. This addresses the concern that the $\sqrt{(1-\beta)r_1/b}$ term, which is by itself independent of $(\eta, T)$, forces the use of large batches: it does not, because the accompanying $A$-term is the binding constraint and is $\eta$-controllable.

## J   Additional Results (Language Modeling)

We provide supplemental plots for the LLM experiments discussed in the main text. Figure 18 shows final loss and SFO complexity versus batch size. Muon is consistently better than AdamW and the gap widens at large batch sizes. Nesterov momentum and weight decay do not produce systematic gains in this setting.

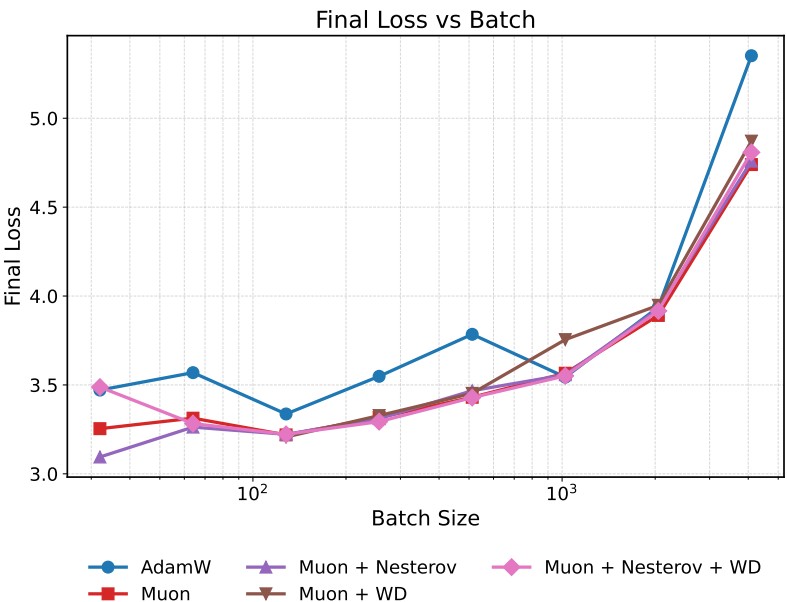

Figure 18: C4/ Llama3.1 (320M). Final training loss versus batch size. Muon outperforms AdamW across all batch sizes, with a larger margin at bigger batches.

To examine the role of momentum, we swept $\beta$ under two configurations: with and without Nesterov (weight decay fixed at zero unless indicated). Figure 19 shows that the best trade-off is near $\beta = 0.95$. As $\beta$ increases, the critical batch size decreases, but very small or very large values of $\beta$ degrade both loss and SFO.

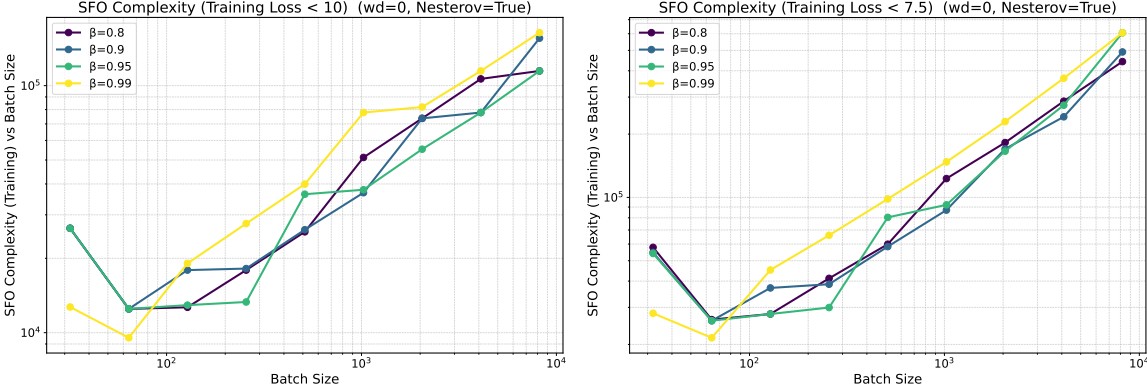

Figure 19: Effect of momentum $\beta$ on C4/ Llama3.1 (320M). SFO for target loss of 10.0 (left) and SFO for target loss of 7.5 (right) across batch sizes under (WD=0, Nesterov=True). The optimum is near $\beta = 0.95$; excessive or too small momentum harms both metrics.

**CBS robustness across stopping criteria.** To verify that the CBS conclusions on the LLM workload are not artifacts of the chosen loss target, we swept the stopping criterion across two metric types: gradient-norm thresholds ($\|\nabla f\| \leq \varepsilon$, $\varepsilon \in [0.1, 0.7]$) and training-loss targets ($\mathcal{L} \leq \ell$, $\ell \in [3.3, 4.0]$). Figure 20 shows the empirical CBS for Muon, Muon+Nesterov, and AdamW under each criterion. Although the exact CBS value shifts as the threshold

tightens, the qualitative ranking (Muon attaining a CBS comparable to or larger than AdamW) is preserved across both metric types. This is consistent with the proxy-invariance observed on FashionMNIST (Table 14).

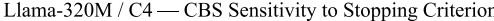

Llama-320M / C4 — CBS Sensitivity to Stopping Criterion

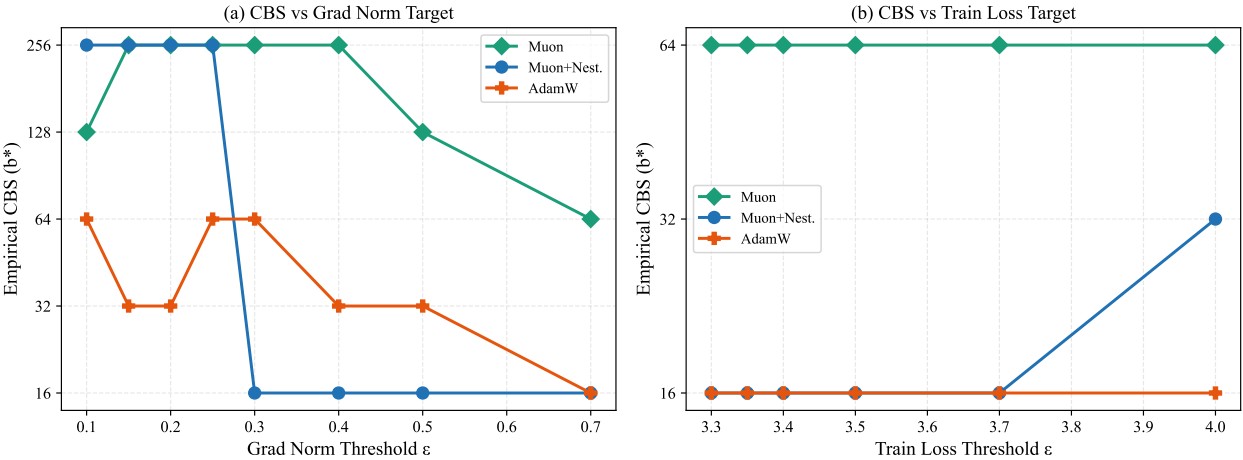

Figure 20: CBS sensitivity to stopping criterion on C4/ Llama3.1 (320M). (Left) CBS vs. gradient-norm threshold. (Right) CBS vs. training-loss target. The Muon $\geq$ AdamW CBS ordering is preserved across both metric types, confirming that CBS conclusions do not depend on the specific proxy.

**Gradient-norm CBS on C4.** To use a stopping criterion that directly corresponds to the quantity bounded by the theorems, we extracted the SFO complexity for Muon with Nesterov ($\beta$=0.95) at gradient-norm thresholds $\|\nabla f\| \leq \varepsilon$ across the full batch-size range (GBS = 16 to 8192). Table 30 reports the results. The CBS shifts from 32 at $\varepsilon = 0.40$ to 128 at $\varepsilon = 0.25$, consistent with the predicted $b^\star \propto 1/\varepsilon^2$ scaling. At every threshold, the SFO curve exhibits a clear U-shape: for example, at $\varepsilon = 0.20$, the SFO at GBS = 4096 ($1.33 \times 10^6$) is $10\times$ larger than at the CBS ($1.30 \times 10^5$).

Table 30: SFO complexity for Muon with Nesterov ($\beta$=0.95) on C4/ Llama3.1 (320M) using the gradient-norm stopping criterion $\|\nabla f\| \leq \varepsilon$. The CBS shifts rightward as $\varepsilon$ tightens. Bold indicates the SFO-minimizing batch size.

| $\varepsilon$ | GBS= 16 | GBS= 32 | GBS= 64 | GBS= 128 | GBS= 256 | GBS= 512 | GBS= 1024 | GBS= 4096 | CBS |
|---|---|---|---|---|---|---|---|---|---|
| 0.40 | 74K | **14K** | 22K | 19K | 38K | 38K | 60K | 1200K | 32 |
| 0.30 | 107K | 110K | 30K | **30K** | 40K | 64K | 88K | 1274K | 128 |
| 0.25 | 131K | 119K | 36K | **35K** | 358K | 554K | 710K | 1298K | 128 |
| 0.20 | 148K | **130K** | 248K | 297K | 487K | 687K | 868K | 1331K | 32 |
| 0.15 | 186K | **165K** | 279K | 427K | 660K | 971K | 1061K | 1364K | 32 |

**Training Curves for Muon on Llama3.1 (320M)** To further illustrate Muons optimization dynamics on large-language-model workloads, we report full training curves for Llama3.1 (320M) trained on C4 at multiple batch sizes. Figure 21 plots the training loss against the number of steps for Muon and AdamW. Across all batch sizes, Muon exhibits faster loss reduction in early training and consistently reaches lower final loss within the same token budget. These results complement Figures 5 and 6, demonstrating that the advantages of Muon are visible not only in SFO complexity but also in the raw optimization trajectory.

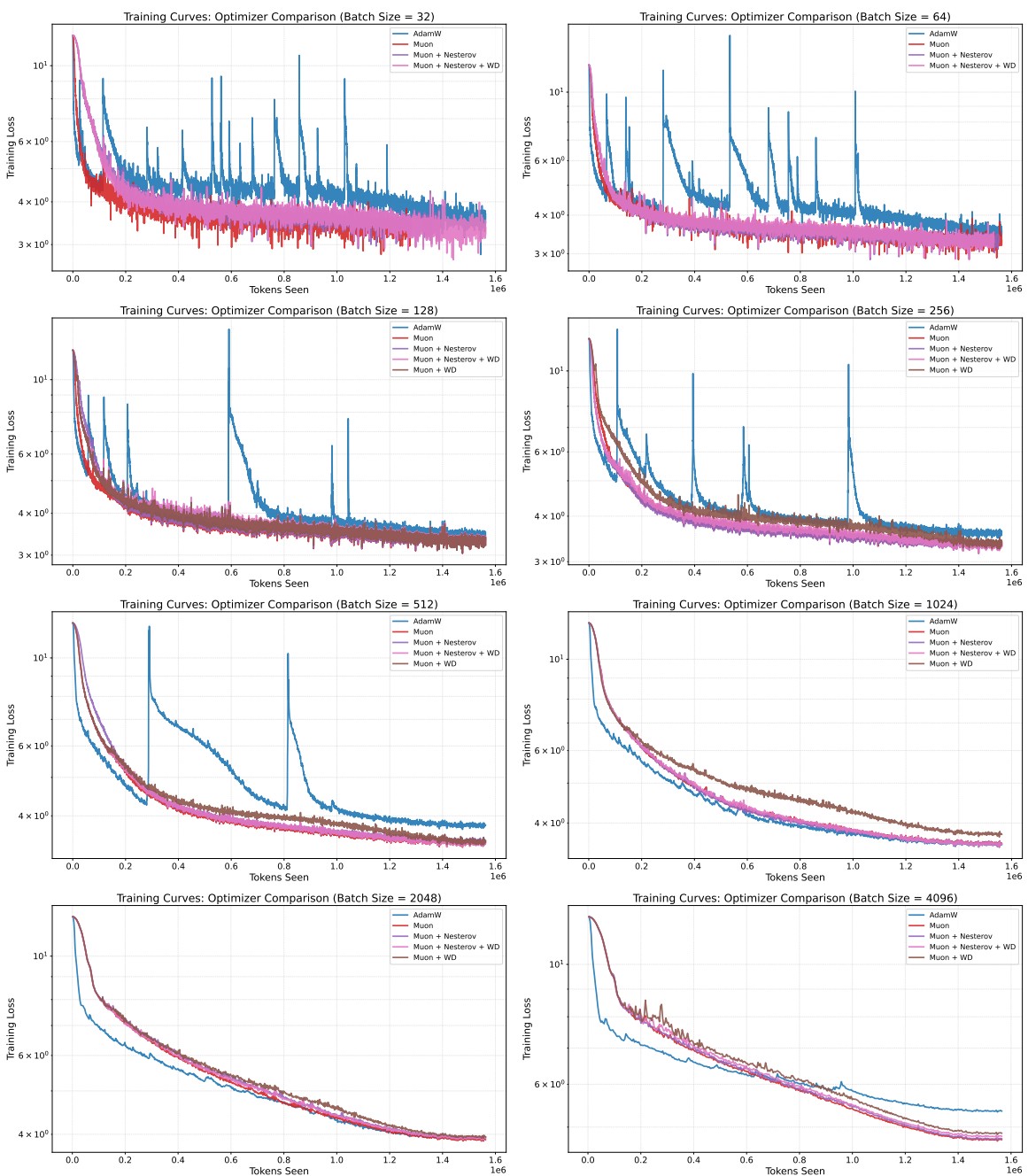

Figure 21: Training loss curves for Muon vs. AdamW on Llama3.1 (320M)/ C4. Each plot corresponds to a different global batch size (32 to 4096, top-left to bottom-right). Muon variants consistently achieve faster loss reduction and lower final loss across all batch sizes.

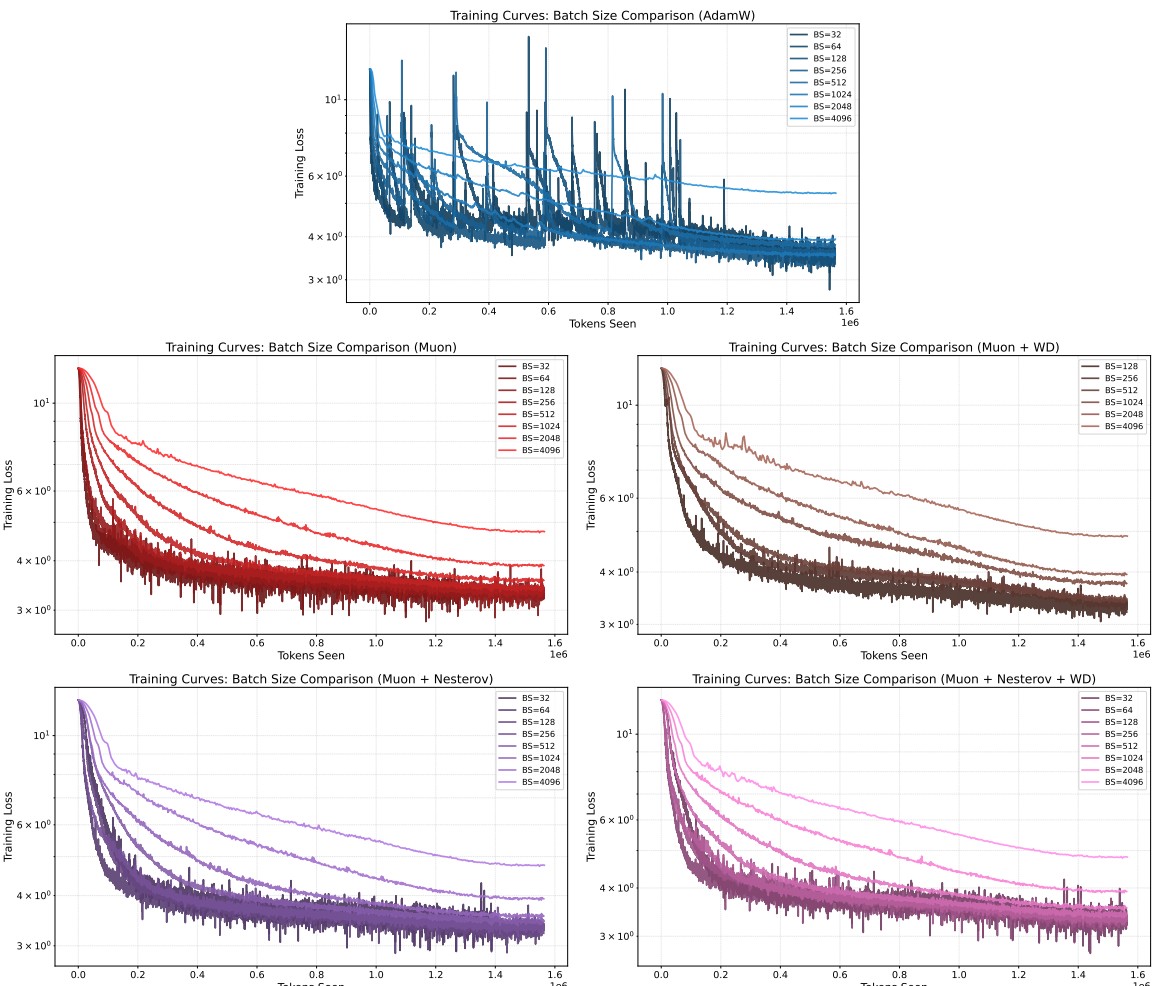

Figure 22: Training loss curves grouped by optimizer on Llama3.1 (320M)/ C4. Each plot corresponds to AdamW, Muon, Muon+WD, Muon+Nesterov, Muon+Nesterov+WD (top to bottom). Smaller batch sizes consistently achieve lower final loss across all optimizers.

