# OpenReview forum: "Convergence Bound and Critical Batch Size of Muon Optimizer"
_TMLR — Rejected by TMLR_

### Review · Reviewer_hsB9 · 2026-03-20

**Summary Of Contributions:**

This paper studies the convergence of muon optimizers. The paper considers several variants: the variants with or without the momentum, and the one with or without weight decay. For all these variants, the paper derives convergence rates in expectation in terms of the step size, the iteration number, the batch size and the rank of the gradients. The paper also analyzes the critical batch size for Muon and indicates the existence of a critical batch size to minimize the computational complexity. Experimental results are also included to illustrate the behavior of muon optimizers.

**Strength**

- The paper studies muon optimizers, which are a class of popular optimizers widely used in practice.
- The theoretical analysis is correct.

**Weakness**
- It seems that the derived convergence rates do not outperform existing rates. For example, in Table 3, Shen et al (2025) gives convergence rates of order $1/T+1/\sqrt{b}+r\eta$ and the performance is measured under the nuclear norm. The paper gives the rate $1/T+1/\sqrt{b}+n\eta$ and the performance is measured under the Frobenius norm. First, the rate is worse than Shen et al (2025) as there rate involves $r\eta$ and the bound in the paper involves $n\eta$. Second, Shen et al (2025) considers a stronger performance measure as the nuclear norm is larger than the Frobenius norm.

**Audience:**

Yes

**Audience Explanation:**

The paper considers a very popular class of optimization algorithms which use the matrix structure of optimization variables in machine learning. The paper gives both theoretical and empirical analyses, which give some practical guidelines in choosing hyperparameters in muons, e.g., the batch size.

**Broader Impact Concerns:**

No broader impact concerns.

**Claims And Evidence:**

Yes

**Claims Explanation:**

The statements of the theoretical results are clear. The theoretical analysis is also clear. While the arguments are standard, these arguments are correct in the theoretical analysis.

The paper includes extensive experimental results to validate the theoretical analysis. Empirical analyses validate the stability condition, the critical batch size and the role of rank.

**Requested Changes:**

The algorithm description is a bit confusing. For example, in Algorithm 1, the update of $M_t$ already uses the nesterov momentum. Then, it is not clear to me whether the update of $C_t$ using the momentum is necessary. Indeed, one can easily write $C_t$ as a weighted average of $M_{t-1}$ and gradients from the current formula.

The convergence rate involves $r$. Note that $r$ is the maximum rank encountered in practice. It may be large if one runs the algorithm for a large enough number of iterations. In this case, the effect of improving the rank dependency may be not significant.

The convergence rate involves $\sqrt{\frac{(1-\beta)r}{b}}$. This term does not involve $\eta$ and $T$. Therefore, one needs to choose a large batch size to ensure convergence. However, a large batch size would incur a large computational cost.

The font size in the Experimental setup is a bit small.

Table 3: some terms show the dependency on $\eta$, e.g., $\eta n$, while other terms hide this dependency, e.g., $1/T$. It would be better to show the dependency on $\eta$ for consistency.

Typos
- "a smaller upper bounds" should be "a smaller upper bound"
- Lemma A.1: $C^2$ should be $\sigma^2$
- Page 19: the identity in the second displayed equation should be an inequality
- Proof of Proposition 3.1: the last two lines in the displayed equation are the same

---

> ### Author Response · Authors · 2026-04-23
>
> We thank you for confirming the correctness of our theoretical analysis and for the constructive suggestions.
>
> ## **Reply to the comparison with Shen et al. (2025):**
>
> We thank the reviewer for drawing attention to the comparison with Shen et al. (2025). Motivated by this observation, we have substantially refined our main theorem (see the top-level official comment).
>
> First, following Shen et al., we now introduce $r_2 := \max_t \operatorname{rank}(O_t)$ and use the tighter bound $\Vert O\_t\Vert_{\rm F} \le \sqrt{r\_2}$ in place of $\sqrt{n}$. The error term therefore improves from $n\eta$ to $\bar{r}\eta$ with $\bar{r} := \max \lbrace r_1, r_2 \rbrace$.
>
> Second, we have upgraded the convergence metric from the Frobenius norm to the nuclear norm of the gradient by keeping the nuclear-norm term tight to the final stage of the proof rather than relaxing it early. Because $\Vert \nabla f(W_t)\Vert_{\rm F} \le \Vert \nabla f(W\_t)\Vert_*$, the new statement is strictly stronger than the Frobenius-norm version. Table 3 has been updated to reflect both refinements, so the comparison with Shen et al. is now on the same nuclear-norm footing, and our analysis additionally covers the Nesterov + weight-decay variant not treated in that prior work.
>
> ## **Reply to Requested Changes:**
> **1. The algorithm description is a bit confusing.**
>
> In the original implementation of the Muon optimizer by Jordan et al., users can choose via an argument between $C\_t := \beta M\_t + (1-\beta)\nabla f\_{\mathcal{S}\_t}(W\_t)$ and $C\_t := M_t$. Algorithm 1 faithfully reflects the implementation by Jordan et al. The update $C_t = \beta M_t + (1-\beta)\nabla f_{\mathcal{S}_t}(W_t)$ is the "accelerated gradient" form of Nesterov momentum, following the reparametrization of Sutskever et al. (2013) that is standard in modern deep learning frameworks (e.g., PyTorch's SGD with `nesterov=True`). It is equivalent to the classical Nesterov form under a change of variables. We have added a clarifying remark in Section 2.1 of the revised manuscript.

---

> ### Author Response · Authors · 2026-04-23
>
> **2. The convergence rate involves $r$. Note that $r$ is the maximum rank encountered in practice. It may be large. The term $\sqrt{\frac{(1-\beta)r}{b}}$ does not involve $\eta$ and $T$. Therefore, one needs to choose a large batch size to ensure convergence.**
>
> We thank the reviewer for flagging these two distinct concerns---the potential growth of $r_1$ and the $b$-independence of the $\sqrt{(1-\beta)r_1/b}$ term. We address each with dedicated controlled FashionMNIST MLP experiments (hidden widths $\in \lbrace 512, 2048\rbrace$, $\beta \in \lbrace0.9, 0.95, 0.99\rbrace$, 3 seeds, up to 8000 steps; full details in the new Appendix I).
>
> *On the magnitude and growth of $r_1$.* Every 150 steps we compute the SVD of $E_t := C_t - \nabla f(W_t)$ for the $2048 \times 784$ fc1 layer, with $\nabla f(W\_t)$ defined *exactly as in the theorem*---the full empirical-risk gradient averaged over all 60,000 training examples, not a mini-batch proxy. The hard rank (singular values $>10^{-3}\sigma_{\max}$) rises during the first $\approx 500$ steps and then stays essentially flat: at $h=2048, \beta=0.95$ it is $186$ at step 600, $198$ at step 1200, $200$ at step 1800, and $246$ at step 3000. Across all six $(\beta, \text{width})$ configurations the late-training hard rank satisfies $r_1 \leq 0.42\,\min \lbrace m,n \rbrace$, and the Frobenius stable rank $\Vert E\_t \Vert_F^2 / \Vert E\_t \Vert_{\mathrm{op}}^2$ remains in the range $1.15$--$1.25$. These measurements show that the worst-case bound $r_1 \leq \min \lbrace m,n \rbrace$ used in the proof is not attained in practice, and they align with Ahn et al. (2025, Dion) that $r \approx n/4$ suffices for Muon up to 3B parameters.
>
> *On the $b$-independence of the $\sqrt{(1-\beta)r_1/b}$ term.* Our theorem predicts that the time-averaged gradient-norm plateau has the structural form $(\text{plateau})^2 = A + B/b$, where $A = \mathcal{O}(\eta L \hat{r})$ is the $\eta$-controllable floor and $B \propto (1-\beta)\,r_1\,\sigma^2$ is the mini-batch variance term. The $\sqrt{(1-\beta)r_1/b}$ term quoted by the reviewer is therefore only the $B/b$ component; the overall plateau is *not* a pure $1/\sqrt{b}$ curve. Fitting this model to runs across $b \in \lbrace 8, 16, 32, 64, 128, 256, 512\rbrace$ at $(\beta, \eta, h) = (0.95, 0.05, 1024)$ yields $\sqrt{A} = 0.349$, $\sqrt{B} = 1.494$, and $R^2 = 0.973$. The $A$-term dominates once $b \gtrsim 64$, so increasing the batch beyond that point yields diminishing returns; convergence is not predicated on a large batch.
>
> *On the lever that controls the remaining floor.* Because $A$ scales with $\eta$ whereas $B/b$ does not, the effective way to push the plateau down at a fixed budget is to decay the step size rather than enlarge the batch. At the most floor-sensitive regime $b=8$, replacing constant $\eta=0.05$ with $\eta_t = 0.1/\sqrt{1+t}$ (3 seeds, 6000 steps) reduces the plateau from $0.535 \pm 0.18$ to $0.408 \pm 0.11$---a $24\%$ drop with $40\%$ lower variance---mirroring the diminishing-step-size argument in Appendix I.3.
>
> As complementary large-scale evidence, on Llama-320M / C4 the end-of-training Gradient Noise Scale is comparable between Muon and AdamW at batch sizes $\leq 1024$ (GNS $\approx 55$--$70$), indicating that Muon's matrix-aware update does not inflate the effective stochastic variance that scales with $r_1$.
>
>
> **3. The font size in the Experimental setup is a bit small.**
>
> We have increased the font size in experimental setup section. Thank you for the suggestion.
>
>
> **4. Table 3: show the dependency on $\eta$ for consistency.**
>
> We have updated Table 3 to ensure consistent notation for $\eta$ across all compared analyses.
>
> **Typos:**
> Thank you for pointing out the typos. We have corrected all of them as suggested.

---

> > ### Comment · Reviewer_hsB9 · 2026-05-08
> > **Response**
> >
> > I thank the authors for their detailed reply, which addresses most of the issues. The remaining issue is the term $\sqrt{(1-\beta)r/b}$ in the convergence rate. One still requires a large $b$ to make sure that this term is very small as it is independent of $\eta$ and $T$.

---

> > > ### Author Response · Authors · 2026-05-09
> > >
> > > We thank the reviewer for the continued engagement and for confirming that most of the issues have been addressed. As discussed in the paragraph "Interpretation of the role of rank $r_1$" below Theorem 3.1, the term $\sqrt{(1-\beta)r/b}$ can also be made small by setting $\beta$ close to 1, without requiring a large batch size $b$. For example, using $\eta = T^{-3/4}$ and $1-\beta = T^{-1/2}$ yields $\frac{1}{T}\sum_{t=0}^{T-1} \mathbb{E}\left[\Vert \nabla f(W_t) \Vert_{*}\right] = \mathcal{O}\left( T^{-1/4} \right)$.

---

### Review · Reviewer_94g3 · 2026-04-05

**Summary Of Contributions:**

This paper provides theoretical analysis of the Muon optimizer across four variants like Nesterov momentum, and weight decay, derives convergence bounds, and characterizes the critical batch size (CBS)---the batch size minimizing stochastic first-order oracle (SFO) complexity.

**Audience:**

Yes

**Audience Explanation:**

This submission has the following strengths that make people interesting. It provided fundamental theoretical contributions which are solid and well-scoped. The convergence proofs for all four Muon variants fill a genuine gap in the literature. The comparison table (Table~3) honestly situates the work relative to prior analyses, and the paper is notably the first to handle both Nesterov momentum and weight decay jointly.

This submission also presented the weight decay analysis is the most interesting part. Propositions 3.1 and 3.2 show almost-sure boundedness of parameter and gradient norms without assuming bounded gradients---a meaningful technical advantage over prior work. The condition $\eta \leq 1/\lambda$ as a sharp stability boundary is clean and practically actionable, and Figure~1 validates it convincingly.

The paper also discussed limitations of the CBS formula. The authors are appropriately careful that Table~1's formulas are qualitative scaling laws, not absolute predictors, since $\sigma^2$, $r$, and $\varepsilon$ are unobservable. The ratio-cancellation argument that unknown quantities cancel when comparing variants on the same workload is a clever way to make the theory testable.

Finally, the paper provided two-level validation strategy is well-motivated. Separating the controlled full-Muon setting, which matches theory exactly from the practical hybrid experiments, i.e., Muon plus AdamW is methodologically sound and addresses a real gap between theory and practice.

**Broader Impact Concerns:**

No Impact

**Claims And Evidence:**

Yes

**Claims Explanation:**

This is a competent and honest theoretical paper with a clear contribution: convergence analysis of the most practically used Muon variant, i.e., Nesterov acceleration and weight decay. The weight decay boundedness results are the strongest part. The authors have clearly stated all the theoretical theorems for all the convergence rates. The authors provided all the mathematical proof and theoretical analysis in the appendix. The authors also conduct a lot of numerical experiments ad the empirical results from the numerical experiments are consistent with the theoretical results.

**Requested Changes:**

This submission also has the following concerns and weakness that deserve changing. First, the convergence rate is not totally novel. The $\mathcal{O}\!\left(1/(\eta T) + \sqrt{(1-\beta)r/b} + n\eta\right)$ bound matches standard non-convex SGD and Adam-type rates. The paper acknowledges this but does not clearly argue why Muon's bound is tighter in practice. The rank-$r$ argument is suggestive but speculative, and the claim that $r \ll n$ may make Muon's bound tighter is hedged so heavily it loses force.

Also, the gap between theory and the Newton-Schulz approximation is under-addressed. This submission assumes $O_t$ satisfies Eq.~(1) exactly, then briefly argues the approximation error is negligible in a few lines. For a paper whose primary contribution is convergence proofs, this deserves a proper proposition with quantified error propagation, not an informal remark about exponential convergence.

Moreover, the CBS analysis has limited novelty beyond the SFO framework. The critical batch size derivation essentially applies the existing framework to Muon's specific convergence bound. The result in Proposition 4.2 about the lower bound is fairly mechanical once the convergence theorem is established. The paper would benefit from discussing whether the CBS formula reveals anything qualitatively new about Muon that could not be inferred from general principles.

Also, in the numerical experiment section, the LLM results are somewhat inconclusive. Figures 5 and 6 show that on C4/Llama, adding Nesterov or weight decay does not yield consistent gains, which is at odds with the vision results. The explanation is defensible but feels post-hoc. The $\beta \approx 0.95$ sweet spot in Figure~6 is consistent with theory, but the overall C4 results feel less convincing than the CIFAR results.

This submission also has the following minor issues:

First, theorem~3.2 requires $\lambda < \min\{1/\|W_0\|_{\mathrm{op}},\, 1\}$, but thepractical consequence of the operator-norm condition is never discussed. Is this automatically satisfied for standard initializations?

Second, table~2 appears without adequate surrounding discussion---it is unclear what workload or metric it reports at first glance.

Third, the notation $\Delta := \|M_0 - \nabla f(W_0)\|_F$ appears in theorems but its role in the bound (it decays as $1/T$) is not explained intuitively for readers less familiar with momentum analysis.

---

> ### Author Response · Authors · 2026-04-23
>
> We thank you for your thorough and positive assessment, and for recognizing the strengths of our weight decay analysis and two-level validation strategy.
>
> ## **Reply to Requested Changes:**
> **1. The convergence rate is not totally novel.**
>
> As pointed out by the reviewer, while the $\mathcal{O}\left( 1/\eta T + \sqrt{(1-\beta)r/b} + n\eta \right)$ bound matches the convergence rate of existing algorithms, it is important to note that the convergence metric is different. The Euclidean norm $\Vert \cdot \Vert\_2$ is the standard metric for analyzing the convergence of SGD-type optimizers, and its natural extension to matrices is the Frobenius norm $\Vert \cdot \Vert\_F$. Since $\Vert \cdot \Vert\_F \leq \Vert \cdot \Vert\_*$, convergence under the nuclear norm is stronger than that under the Frobenius norm. Our revised theorem holds for the nuclear norm, which suggests that Muon provides a potentially more informative convergence guarantee in practice. (see the updated discussion after Theorem 3.1)
>
> In addition, our main contribution is not a faster asymptotic rate, but rather demonstrating that this rate is **achieved under weaker assumptions when Muon incorporates both Nesterov momentum and weight decay**. Specifically:
>
> 1. Weight decay automatically ensures bounded parameter and gradient norms (Propositions 3.1 and 3.2), eliminating the need for the "bounded gradient" assumption that is common in Adam-type analyses.
> 2. The bound depends on the **rank** $r_1$ rather than the full dimension $d = mn$, reflecting Muon's matrix-aware structure. To move this argument beyond the speculative, we have added direct measurements of $\operatorname{rank}(C\_t - \nabla f(W\_t))$ on controlled FashionMNIST experiments (new Appendix I), where $\nabla f(W_t)$ is computed as the *exact* full empirical-risk gradient over all 60,000 examples rather than a mini-batch proxy. Across $(\beta, \text{width}) \in \lbrace 0.9, 0.95, 0.99 \rbrace \times \lbrace 512, 2048 \rbrace$, the hard rank saturates within $\approx 500$ steps at $r_1 \le 0.42\,\min \lbrace m,n \rbrace$ (often $\le 0.30$) and the Frobenius stable rank remains at $1.15$–$1.25$, indicating that the energy of $E_t$ concentrates on essentially $\mathcal{O}(1)$ directions. These direct measurements corroborate the external evidence of Ahn et al. (2025, Dion)—who report $r \approx n/4$ suffices for Muon up to 3B parameters—and jointly establish that the worst-case $r_1 \le \min \lbrace m,n \rbrace$ is not attained in practice.
> 3. To our knowledge, this is the first convergence analysis to handle both Nesterov momentum and weight decay jointly for Muon.
>
> **2. The gap between theory and the Newton-Schulz approximation is under-addressed.**
>
> To address this issue, we have added a convergence analysis of Muon with Newton-Schulz iteration in Section 3.3. The new Proposition 3.3 quantifies the effect of the approximation error introduced by the Newton-Schulz iteration on the convergence rate. Specifically, the convergence rate includes an additional error term that decreases exponentially with the number of Newton-Schulz iterations $k$. Since the Newton-Schulz iteration converges at a quadratic rate ($\Vert \tilde{O}\_t^{(k)} - O\_t \Vert_{\rm F} = \mathcal{O}(c\_0^{3^k})$), this error is exponentially small and does not affect the asymptotic convergence rate (see Section 3.3).
>
> Empirically, in our controlled toy experiment comparing exact SVD, NS($k{=}3$), and NS($k{=}5$),
> the mean orthogonalization error is $\approx 0.036$ for NS3 and $\approx 0.028$ for NS5. All three orthogonalizers converge to comparable gradient norms (e.g., at $b{=}128$: SVD $7.11 \times 10^{-3}$, NS5 $4.12 \times 10^{-3}$, NS3 $2.77 \times 10^{-3}$; the differences reflect slightly different optimization trajectories, not degradation). On ResNet-18/CIFAR-10, the test accuracy gap between $k{=}3$ and $k{=}5$ is at most $+0.15$ percentage points (Table 8 in Appendix F).

---

> ### Author Response · Authors · 2026-04-23
>
> **3. The CBS analysis has limited novelty beyond the SFO framework.**
>
> To address this issue, we have added a discussion of our novelty at the end of Section 4. Our CBS analysis reveals two features specific to Muon that do not arise from general SFO-framework arguments:
>
> 1. **Rank-dependent scaling:** $b^\star \propto r_1$, where $r_1$ is the rank of the momentum error matrix, which does not appear in the SGD analysis. This structural difference directly reflects Muon's matrix-aware update.
> 2. **Explicit $\beta$ dependence:** The factor $(1-\beta)(1+\sqrt{2}\beta)^2$ for the Nesterov variant links momentum tuning to the optimal batch size---a relationship not captured by prior CBS results for SGD or Adam.
> 3. **Empirical confirmation on ResNet-18/CIFAR-10:** We have added a five-variant CBS experiment (66 runs) on the same practical workload. By widening the hyperparameter range ($\lambda \in \lbrace 0, 0.2, 0.5\rbrace$, Nesterov on/off), the CBS shifts from $\approx 128$ (w/o Nesterov) to $\approx 4096$ (w/ Nesterov + $\lambda{=}0.5$), with the weight-decay CBS ratio matching the predicted $1/(1{-}\lambda)^2$ factor quantitatively (see our reply to Reviewer ApvR, Requested Change 3, and the new Appendix table).
>
>
> **4. The LLM results are somewhat inconclusive.**
>
> The reviewer correctly observes that adding Nesterov momentum or weight decay does not yield consistent loss gains on C4. We agree that this observation deserves a clearer explanation, which we provide below.
>
> Theorems 3.1 and 3.2 predict the same asymptotic convergence rate for all four Muon variants. The theory does not predict that any variant achieves a lower final loss under a fixed token budget; it predicts that Nesterov and weight decay shift the critical batch size and provide boundedness guarantees (Propositions 3.1 and 3.2). The inter-variant loss differences on C4 are indeed small (typically $< 0.05$), which is consistent with the same-rate prediction rather than a negative finding.
>
> We have expanded the Llama-320M / C4 experiments (batch sizes 16 to 8192, $\beta \in \lbrace 0.7, 0.8, 0.9, 0.95, 0.99, 0.999\rbrace$, all trained for 3.2B tokens) and validate two testable predictions from the CBS analysis using the gradient-norm stopping criterion, which directly corresponds to the quantity bounded by the theorems:
>
> 1. **The $\beta$--CBS relationship holds on C4.** Using the gradient-norm threshold $\Vert \nabla f\Vert \le 0.20$, the SFO at $\beta = 0.95$ is $1.30 \times 10^5$ (CBS $= 32$), compared to $4.02 \times 10^5$ at $\beta = 0.80$ (CBS $= 128$) and $1.37 \times 10^6$ at $\beta = 0.70$ (CBS $\ge 1024$, SFO nearly flat across all batch sizes). This confirms Figure 6: as $\beta$ decreases, the CBS shifts rightward and the SFO increases, consistent with the theoretical prediction.
>
> 2. **CBS shifts with the target threshold, consistent with $b^\star \propto 1/\varepsilon^2$.** For Muon with Nesterov at $\beta = 0.95$, the SFO-minimizing batch size shifts from CBS $= 32$ at loose thresholds ($\Vert \nabla f\Vert \le 0.40$) to CBS $= 128$ at tighter thresholds ($\Vert \nabla f\Vert \le 0.25$). The full SFO-vs-batch-size curves (spanning GBS $= 16$ to $8192$) exhibit a clear U-shape at every tested threshold, with the SFO at GBS $= 4096$ being $10$--$40\times$ larger than at the CBS. A new table is included in the Appendix.
>
> We have revised the LLM discussion in Section 5 accordingly.
>
>
> **5. Minor Issues**
>
> **Reply to Minor Issue 1:**
>
> Under He initialization, the expected operator norm $\Vert W_0\Vert\_{\rm op}$ is approximately $\sqrt{2}$ for a square matrix. We verified this empirically on ResNet-18: the operator norms of 3x3 conv layers are $\approx 0.76$ and 1x1 shortcut layers $\approx 1.41$, giving $1/\max_\ell \Vert W\_0^{(\ell)}\Vert\_{\rm op} \approx 0.71$. Since our experiments use $\lambda \le 0.5$, the condition $\lambda < 1/\Vert W\_0\Vert\_{\rm op}$ is comfortably satisfied. This discussion has been added to the manuscript.
>
> **Reply to Minor Issue 2:**
>
> We have added surrounding text to clarify the workload and metric. Thank you for pointing this out.
>
> **Reply to Minor Issue 3:**
>
>  This term $\Delta$ arises from the initialization $\boldsymbol{m}\_{-1} := \boldsymbol{0}$ used in Algorithm 1, and its effect diminishes as the number of iterations increases. From a theoretical perspective, initializing $\boldsymbol{m}\_{0} := \boldsymbol{0}$ or $\boldsymbol{m}_0 := \nabla f\_{\mathcal{S}\_0}(W\_0)$ would yield a more refined upper bound for $\Delta$. However, note that the Muon implementation adopts $\boldsymbol{m}\_{-1} := \boldsymbol{0}$. We have added a similar explanation following Theorem 3.1.

---

### Review · Reviewer_ApvR · 2026-04-09

**Summary Of Contributions:**

# Paper summary

This paper proposes a theoretical study of the Muon optimizer in four variants. Namely, with and without Nesterov momentum, with and without weight decay. In particular, the authors derive nonconvex convergence bounds, boundedness results under simple assumptions, and a lower bound on the critical batch size. Practical experiments on a toy dataset and more classical datasets, such as CIFAR, validate the proposed theory to some extent.

# Strenghts
- The topic is relevant. Muon still lacks a theoretical grounding, and such attempts are valuable
- The proposed study not only proposes a convergence proof but also a practically meaningful critical batch size
- The empirical section, aiming to go beyond pure theory, is appreciated
- The authors explicitly discuss the limitations

# Weaknesses
- My main concern is the claim regarding the tightness of the bound derived in Theorem 3.1 and 3.2. The authors claim, in p. 10, that the leading constant should be smaller due to $\frac{1}{1-\lambda}$ factor. While this is confirmed in practice in the experiments of Figure 2, the theoretical argument does not hold since $\lambda \leq 1$ the factor should **increase** the bound. Therefore, such claim, repetitively made throughout the paper, seems to be supported by experiments only but contradicted by the presented theory. This theoretical point being one of the major claims of the paper, this seems to be a major issue.
- The empirical validation does not test the theorem directly. The theorem is about the average expected norm of the full gradient. The experiments measure something else: the norm of the mini-batch gradient, then smooth it with an exponential moving average, and then record the first training step at which this smoothed quantity goes below a chosen threshold. This may be a useful practical proxy, but it is not the same quantity as the one analyzed in the theorem. I understand that full gradient is impractical to compute; however, experiments on the toy experiments to show the difference between theoretical and practical quantities would help clarify this discrepancy between theorem and experiments at larger scale.
- Although the critical batch size is defined theoretically as the batch size that minimizes SFO complexity, the empirical CBS analysis in the main text is instantiated with several different stopping targets. In section 5, the paper introduces a theory-aligned proxy $T_\epsilon(b)$, defined as the first step at which the EMA-smoothed mini-batch gradient norm falls below a threshold $\epsilon$, and later gives $\epsilon = 0.08$ as an example in the controlled full-Muon discussion. By contrast, Figure 3 on ResNet18/CIFAR10 uses steps to $90%$ test accuracy and SFO to $95%$ training accuracy, while Figure 5 on C4/llama uses steps and SFO to reach a target training loss (captioned as training loss $< 4.0$). Since the empirical CBS depends on the chosen stopping target, the authors should explain how these thresholds were selected and discuss how sensitive their conclusions are to this choice.
-  The experiments support the existence of a critical batch size and some qualitative trends, but they provide only limited support for the finer ordering among Muon variants predicted by the CBS formulas of Table 1. In Figure 3 the SFO-minimizing batch sizes of the four Muon variants are very close, and on C4 the paper itself states that Nesterov momentum and weight decay do not yield gains. As a result, the empirical evidence is directionally consistent with the theory, but it does not strongly demonstrate that these hyperparameter-dependent CBS differences are large or consequential.

**Audience:**

Yes

**Audience Explanation:**

The paper gives a theoretical analysis of Muon, a popular optimizer, especially for training LLMs. This would be of interest to a major audience of TMLR.

**Broader Impact Concerns:**

I do not have any broader impact concerns outside the ordinary computing cost.

**Claims And Evidence:**

No

**Claims Explanation:**

My main concern is the claim regarding the tightness of the bound, which, to my understanding, is incorrect. If the author can clarify this point, then other claims are overall convincing.

**Requested Changes:**

- Critical: Clarify the bounds tightness theory interpretation. It seems incorrect to me at the moment.
- clarify the mini-batch gradient norm as an unbiased proxy for the full gradient norm and include experiments showing how such a quantity might differ in practice
- clarify the definition of the critical batch size

---

> ### Author Response · Authors · 2026-04-23
>
> We are grateful for your recognition that the topic is relevant and the CBS analysis is practically meaningful. We address your three requested changes below.
>
> ## Reply to Requested Change 1:
> **Clarify the bounds tightness theory interpretation.**
>
> You are correct. Our previous claim that the bound for Muon with weight decay is improved due to the $1/(1-\lambda)$ factor was incorrect---this factor in fact slightly *loosens* the upper bound. We have corrected the corresponding statements in the revised manuscript.
>
> The corrected interpretation is as follows. The asymptotic convergence rate (Theorems 3.1 and 3.2) is almost the same for all four Muon variants. The practical advantage of incorporating weight decay is not a tighter convergence rate, but rather the **boundedness guarantees** established in Propositions 3.1 and 3.2 and Lemma C.1: when $\eta \le 1/\lambda$, both the parameter norm $\Vert W\_t\Vert_{\rm F}$ and the gradient norm $\Vert \nabla f(W_t)\Vert_{\rm F}$ remain bounded almost surely throughout training. These boundedness results do not require the commonly imposed "bounded gradient" assumption, which is a meaningful technical advantage.
>
> The empirical advantage observed in Figure 2---where the weight-decay variant converges faster---is therefore explained by the implicit regularization from bounded norms, rather than by a smaller leading constant in the convergence bound. We have revised all relevant passages (Abstract, Introduction, Sections 3 and 5) to reflect this corrected interpretation.
>
>
> ## Reply to Requested Change 2:
> **Clarify the mini-batch gradient norm as an unbiased proxy for the full gradient norm and include experiments showing how such a quantity might differ in practice.**
>
> We agree that the full-batch gradient norm (the theoretical quantity) and the EMA-smoothed mini-batch gradient norm (the experimental proxy) are not the same. We have also corrected the manuscript to describe this as a "practical surrogate" rather than an "unbiased proxy," since the norm of an unbiased gradient estimator is not itself an unbiased estimator of the true gradient norm.
>
> To directly quantify the gap, we conducted a controlled experiment where computing the full-batch gradient is affordable.
>
> **Setup.** On the Teacher-Student Tanh Regression task (Section 5; Appendix F), we run full-Muon on a single $512 \times 256$ matrix with 6 seeds and batch sizes $\lbrace 8, 16, \dots, 1024\rbrace$. At every training step, we record: (i) the full-batch gradient Frobenius norm $\Vert\nabla f(W_t)\Vert\_{\rm F}$, (ii) the mini-batch gradient norm $\Vert \nabla f\_{\mathcal{S}\_t}(W\_t)\Vert\_{\rm F}$, and (iii) the EMA-smoothed mini-batch gradient norm.
>
> **Results.**
> - The Pearson correlation between the full-gradient norm and the mini-batch gradient norm exceeds **0.99** across all tested batch sizes (ranging from 0.991 at $b{=}8$ to 1.000 at $b{=}1024$), with negligible standard deviation across seeds ($< 0.001$).
> - The correlation between the full-gradient norm and the EMA proxy exceeds **0.97** (ranging from 0.978 to 0.997).
> - We verified this across two noise levels ($\sigma{=}0.1$ and $\sigma{=}0.2$), and the correlations remain consistently above 0.97 (mini-batch) and 0.95 (EMA) in all cases.
>
> We note that the absolute CBS point estimate can differ between the full-gradient metric and the stochastic proxy, because the proxy includes mini-batch noise. However, the high correlation confirms that both metrics track the same underlying optimization dynamics, and the qualitative CBS behavior (U-shaped SFO curve, variant ordering direction) is preserved. A full correlation table is provided in the new Appendix (Table 10, Section F).

---

> ### Author Response · Authors · 2026-04-23
>
> ## Reply to Requested Change 3:
> **Clarify the definition of the critical batch size (stopping target sensitivity).**
>
> We acknowledge that different sections of the paper use different stopping criteria, which may create ambiguity. We now provide a unified explanation.
>
> The choice of stopping criterion reflects the purpose of each experiment:
> - **Practitioner-facing experiments** (ResNet/CIFAR in Figures 3--4; Llama/C4 in Figures 5--6) use accuracy or loss targets, which are the metrics practitioners naturally use to evaluate training efficiency.
> - **Theory-aligned experiments** (controlled full-Muon in Appendix F) use the gradient-norm threshold $\tilde{g}_t \le \varepsilon$, which directly corresponds to the quantity bounded by our theorems.
>
> **Sensitivity analysis.** To verify that the CBS conclusions are robust to this choice, we conducted a comprehensive sensitivity study on FashionMNIST (2-layer MLP, hidden dim 1024, full Muon, **16 seeds**) under **15 different stopping criteria** spanning four metric types and multiple threshold levels (Appendix, Table). The key results are:
>
> | Metric | Threshold | CBS ($b^\star$) |
> |--------|-----------|:---------------:|
> | Test accuracy | $\ge 0.80$ | 32 |
> | Test accuracy | $\ge 0.82$ | 64 |
> | Test accuracy | $\ge 0.84$ | 64 |
> | Test accuracy | $\ge 0.85$ | 128 |
> | Train accuracy | $\ge 0.85$ | 64 |
> | Train accuracy | $\ge 0.88$ | 128 |
> | Train loss | $\le 0.50$ | 32 |
> | Train loss | $\le 0.45$ | 64 |
> | Train loss | $\le 0.40$ | 128 |
> | EMA grad norm | $\le 0.60$ | 32 |
> | EMA grad norm | $\le 0.55$ | 64 |
> | EMA grad norm | $\le 0.45$ | 64 |
> | EMA grad norm | $\le 0.40$ | 128 |
> | EMA grad norm | $\le 0.35$ | 128 |
> | EMA grad norm | $\le 0.32$ | 256 |
>
>
> Two key observations emerge: (1) Across all four metric types, the CBS falls consistently in the range **32--256**, confirming that the qualitative CBS conclusion is robust to the choice of stopping criterion. (2) Within each metric type, **tighter thresholds systematically shift the CBS to larger values**, supporting the theoretical prediction $b^\star \propto 1/\varepsilon^2$. The EMA gradient norm sweep provides the most direct evidence, with the CBS shifting monotonically as $32 \to 64 \to 128 \to 256$ as $\varepsilon$ decreases from $0.60$ to $0.32$.
>
> Additionally, our target-precision sensitivity analysis (Section 5) confirms that varying $\varepsilon$ across four levels yields $b^\star \propto 1/\varepsilon^2$ with $R^2 = 0.984$, and the EMA window sensitivity test shows that windows of 50, 100, and 200 steps yield CBS estimates at the same power-of-two grid point.
>
> We acknowledge that the CBS near its minimum is flat---this is why the four Muon variants appear close in Figure 3. This flatness reflects the known behavior of SFO landscapes near the optimum, as demonstrated by our CBS-neighborhood analysis (Figure 7 in Appendix).
>
> To provide stronger evidence for the inter-variant CBS separation, we conducted an additional experiment on the same ResNet-18/CIFAR-10 setup in which the hyperparameter range is deliberately widened: we sweep Nesterov on/off, $\beta \in \lbrace 0.90, 0.95 \rbrace$, and $\lambda \in \lbrace 0, 0.2, 0.5\rbrace$ across batch sizes $b \in \lbrace 32, \ldots, 4096\rbrace$ (66 runs, 1--3 seeds per cell; new Appendix, Table). The resulting CBS estimates are:
>
> | Configuration | CBS (train 95%) |
> |--------------|:---------------:|
> | w/o Nesterov, $\lambda{=}0$ | $\approx 128$ |
> | w/ Nesterov, $\beta{=}0.95$, $\lambda{=}0$ | $\approx 1024$ |
> | w/ Nesterov, $\beta{=}0.90$, $\lambda{=}0$ | $\approx 1024$ |
> | w/ Nesterov, $\beta{=}0.95$, $\lambda{=}0.2$ | $\approx 2048$ |
> | w/ Nesterov, $\beta{=}0.95$, $\lambda{=}0.5$ | $\approx 4096$ |
>
> The empirical CBS ordering---w/o Nesterov $<$ w/ Nesterov $<$ w/ Nesterov+WD$\_{0.2}$ $<$ w/ Nesterov+WD$\_{0.5}$---matches the direction predicted by Table 1. In particular, the CBS ratio between Nesterov+WD$\_{0.5}$ and Nesterov equals $4096/1024 = 4.0$, which agrees with the predicted factor $1/(1{-}0.5)^2 = 4.0$. The Nesterov-to-non-Nesterov shift ($128 \to 1024$) exceeds the lower-bound prediction of $\approx 2.75\times$, which is consistent with Table 1 providing a lower bound rather than an equality.
>
> These results complement the controlled full-Muon experiments and demonstrate that the hyperparameter-dependent CBS predictions of Table 1 produce consequential and empirically verifiable differences when the hyperparameters are varied over a sufficiently wide range. A new table has been added to the revised manuscript (Section 5 and Appendix).

---

> > ### Comment · Reviewer_ApvR · 2026-05-08
> > **Thank you.**
> >
> > Thank you for your reply and clarifications. Most of my concerns have been addressed.

---

### Review · Reviewer_SkLK · 2026-04-15

**Summary Of Contributions:**

This paper presents a theoretical analysis of the Muon optimizer. Specifically, it provides the convergence rates of the Muon optimizer under different settings, including Nesterov momentum and weight decay. The paper then analyzes the critical batch size of the optimizer and conducts experiments to justify the theoretical findings.

**Strengths**
1. Muon is a successful and promising optimizer for neural network training, especially for large models. Providing a solid theoretical analysis for it is an important and valuable contribution to the community.
2. The theoretical analysis is sound and provides useful insights regarding the critical batch size, which could be very helpful for practical applications.
3. The manuscript is well-organized, clearly written, and easy to follow.

**Weaknesses**
1. While the experimental evaluation successfully validates the general trends aligned with the theory (e.g., Figure 1), it lacks a more precise quantitative analysis. For instance, it would be helpful to explicitly measure certain terms from the theoretical results, such as the rank coefficient r, to calculate the numerical theoretical convergence rate and compare it directly with empirical observations. This is particularly important for the critical batch size, where a quantitative comparison—rather than just a trend analysis—would strengthen the paper.

2. The main difference in the convergence bound between Muon and standard SGD seems to be the variance term, which relies on the rank coefficient r (as discussed in Section 3.1). Although the authors explain this difference and state that r is smaller than n, it would be highly beneficial to actually estimate or measure r in the numerical experiments and then compare the resulting convergence rate with SGD.

3. In Figure 2, the models do not appear to have fully converged yet. It would be better to run the evaluation for more iterations so the overall convergence trend becomes clearer.

**Audience:**

Yes

**Audience Explanation:**

The TMLR audience has a strong interest in optimizers for neural network training. Given Muon's  practicability , these theoretical guarantees are very helpful.

**Broader Impact Concerns:**

There are no significant ethical concerns regarding this work.

**Claims And Evidence:**

Yes

**Claims Explanation:**

Sections 3 and 4 clearly support the theoretical claims regarding the convergence and critical batch size of Muon. The experiments in Section 5 are generally clear and support the theory, though they could be improved by adding more precise quantitative comparisons (as noted in the weaknesses).

**Requested Changes:**

1. Measure or estimate the rank coefficient r in the numerical experiments.
1.1 Use this value to calculate the numerical theoretical convergence rate and compare it directly with your empirical results.
1.2 For the critical batch size, provide a quantitative comparison between the theoretical lower bounds and the experimental outcomes, rather than just showing that the qualitative trends align.
1.3 Based on the empirical measurement of r, provide a more concrete comparison between the convergence rates of Muon and standard SGD to back up the discussion in Section 3.1.

2. Run the experiments shown in Figure 2 for more iterations to ensure the models reach full convergence, which will make the overall trend much clearer.

---

> ### Author Response · Authors · 2026-04-23
>
> We thank the reviewer for the positive assessment and for pushing us toward a more quantitative validation of the theory. The two requested changes led to a substantial strengthening of the empirical section, summarized below.
>
> ## Reply to Requested Change 1 (and 1.1–1.3):
> **Measure $r_1$ and give a quantitative theory-vs-experiment comparison.**
>
> We have added a dedicated appendix (new Appendix I) that directly measures the rank coefficient $r_1 := \max\_t \operatorname{rank}(C\_t - \nabla f(W\_t))$ and uses it to give a quantitative comparison between the theoretical convergence bound and the empirical observations. We address the three sub-points in turn.
>
> ### 1.1 — Direct measurement of $r_1$ and numerical comparison with the convergence bound.
>
> On a controlled 2-layer FashionMNIST MLP (hidden widths $\lbrace 512, 2048 \rbrace$, $\beta \in \lbrace 0.9, 0.95, 0.99\rbrace$, 3 seeds), we compute $E\_t := C\_t - \nabla f(W\_t)$ every 150 steps, where $\nabla f(W_t)$ is the *exact* full empirical-risk gradient over all 60,000 training examples---not a mini-batch proxy. The hard rank (singular values $>10^{-3}\sigma\_{\max}$) saturates within $\approx 500$ steps at $r\_1 \le 0.42\,\min \lbrace m, n \rbrace$ across all configurations, and the Frobenius stable rank $\Vert E\_t\Vert\_F^2 / \Vert E\_t\Vert\_{\mathrm{op}}^2$ remains in the range $1.15$–$1.25$. Detailed trajectories and summary statistics are in Table 27 and Figure 15.
>
> Using these measurements, we evaluate the theoretical prediction that the time-averaged gradient-norm plateau has the form $(\mathrm{plateau})^2 = A + B/b$, where $A = \mathcal{O}(\eta L \hat{r})$ is the step-size-controllable floor and $B \propto (1-\beta)\,r_1\,\sigma^2$ is the mini-batch variance. Fitting this model at $(\beta, \eta, h) = (0.95, 0.05, 1024)$ across $b \in \lbrace 8, 16, \dots, 512\rbrace$ yields $\sqrt{A} = 0.349$, $\sqrt{B} = 1.494$, $R^2 = 0.973$ (Table 28 in Appendix I.2), with relative prediction errors below $12\%$ across the swept batch sizes. Back-substituting the measured $\sigma^2$ into $\sqrt{B} = \sqrt{(1-\beta)\,r\_1\,\sigma^2}$
> recovers an effective $r\_1$ in the range O(1)–O(10),
> which should be interpreted as an empirical proxy rather than the exact worst-case quantity defined in the theory.
> This provides a direct numerical link between the measured rank and the theoretical convergence rate.
>
> ### 1.2 — Quantitative comparison of theoretical CBS lower bounds and empirical CBS.
>
> We distinguish two levels of quantitative comparison:
>
> *Ratio-level comparison (exact and testable).* The theoretical CBS formula $b^\star \propto r_1 \sigma^2 / \varepsilon^2$ contains the workload-dependent constants $(r\_1, \sigma^2, \varepsilon)$. When comparing variants of Muon on the *same workload*, these unknowns cancel in the ratio. For example, Table 1 predicts that the Nesterov variant shifts the CBS by a factor of exactly $(1+\sqrt{2}\beta)^2 / 2$, which depends only on $\beta$. We now report this ratio-based quantitative comparison explicitly: on the controlled full-Muon Teacher–Student task at $\beta = 0.9$, the predicted Nesterov-to-non-Nesterov CBS ratio is $\approx 2.54$, which agrees with the observed shift within one power-of-two grid point. This ratio-level test is the strongest *quantitative* comparison permitted by the structure of the bound.
>
> *Absolute-level comparison (fundamentally limited).* An absolute prediction of $b^\star$ additionally requires $L$, $\sigma^2$, and a specific $\varepsilon$. In the controlled Teacher–Student setting where $L$ and $\sigma^2$ can be estimated, we fit $\hat{Y}, \hat{Z}$ from the proxy $X/T + Y/\sqrt{b} + Z$ and compute a predicted CBS via $\hat{b}^\star = 9\hat{Y}^2/(4\varepsilon^2)$ (Appendix F); the resulting value ($\approx 1500$–$2000$ depending on $\varepsilon$) falls beyond the empirical sweep range and is therefore consistent with the absence of an observed CBS transition within $b \le 2048$, rather than an observed mismatch. We are explicit that the absolute CBS is *not* directly predictable from the bound, and that the CBS formulas of Table 1 should be read as ratio-level scaling laws—a point now strengthened by the new ratio-level comparison above (see end of Section 4.2).

---

> ### Author Response · Authors · 2026-04-23
>
> ### 1.3 — Numerical comparison of Muon vs. SGD using the measured $r_1$.
>
> Standard SGD-type analyses yield a variance term $\sqrt{d/b}$ with $d = mn$, whereas the Muon bound has $\sqrt{r_1 / b}$ (up to $(1-\beta)$ and the matrix-Hölder constants). For the fc1 layer in our FashionMNIST experiment ($m = 2048, n = 784$), $d = mn \approx 1.6 \times 10^6$ while the late-training measured $r_1 \leq 329$ (worst-case across all seeds and $\beta$). The ratio $d / r_1 \geq 4.9 \times 10^3$ translates, via the $\sqrt{\cdot}$ dependence, into a Muon-over-SGD variance-term reduction factor of $\gtrsim 70\times$. Equivalently, to achieve the same variance contribution, SGD would require a batch roughly $70^2 \approx 5{,}000\times$ larger than Muon. We have added this quantitative comparison to the discussion following Theorem 3.1, replacing the previous hedged "$r_1 \ll n$ in practice" statement with a concrete numerical gap derived from direct measurement.
>
> ## Reply to Requested Change 2:
> **Run Figure 2 experiments for more iterations to reach full convergence.**
>
> We agree that at large batch sizes (notably $b{=}2048$ in the previous Figure 2) the standard schedule $E_B = 100 \times (512/B)$ leaves only $25$ epochs, and the gradient-norm curves have not fully entered their late-training plateau.
>
> **1. Main figure replaced with a fully-converged $b{=}128$ panel.** The previous Figure 2 used $b{=}2048$ with only $25$ epochs. The revised figure uses the $b{=}128$ panel from the same experiment, which trains for $400$ epochs ($\approx 1.57\times 10^{5}$ steps). Under this schedule, all four Muon variants reach a clear late-training plateau (training loss $\approx 3 \times 10^{-3}$, EMA gradient norm $\approx 2$--$4 \times 10^{-2}$), and are essentially indistinguishable in loss, consistent with Theorems 3.1-3.2.
>
> **2. Extended-budget $b{=}2048$ re-run.** We also re-ran the $b{=}2048$ experiment with $100$ epochs ($4\times$ the original budget) on the same six optimizers and learning-rate sweeps (30 runs total). All Muon variants reach train loss $< 0.015$ and test accuracy $\sim$92\%, confirming full convergence at the original batch size. A summary table is provided in the revised Appendix (Table 23).

---

### Author Response · Authors · 2026-04-23

We sincerely thank the reviewers for their thoughtful comments and feedback. We would like to announce two specific refinements to our main theorem, which further clarify our contribution and improve the theoretical results.

**1. Refinement of the Convergence Rate**

Initially, we used the bound $\Vert O_t \Vert\_{\rm{F}} \leq \sqrt{n}$ based on the assumption that $\textup{rank}(O\_t) \leq n$ (given $m \geq n$). Following the approach of Shen et al., we have now introduced $r_2$ as the maximum rank of $O_t$ during the optimization process. This allows for a tighter bound: $\Vert O_t \Vert\_{\rm{F}} \leq \sqrt{r_2}$. Consequently, the convergence rate for Muon is refined to $\mathcal{O}\left( \frac{1}{\eta T} + \sqrt{\frac{\bar{r}(1-\beta)}{b}} + \eta\bar{r} \right)$ where $\bar{r} := \max\lbrace r_1, r_2 \rbrace$, $r_1 := \max_{t} \textup{rank}(C_t - \nabla f(W_t))$, and $r_2 := \max_{t} \textup{rank}(O_t)$. This modification improves the error term from $n\eta$ to $\bar{r}\eta$. We note that this refinement does not require substantial changes to the underlying proof structure.

**2. Update of the Convergence Metric**

To better highlight our contribution and facilitate a clearer comparison with prior work, we have updated our convergence criterion from the Frobenius norm to the nuclear norm of the gradient. This was achieved by maintaining the tighter nuclear norm term until the final stage of the proof, rather than relaxing it to the Frobenius norm early on as in the previous version. Since $\Vert\nabla f(W_t)\Vert\_{\rm{F}} \leq \Vert\nabla f(W_t)\Vert\_*$, this provides a strictly stronger convergence guarantee.

---

### Decision · Action_Editor_1XxC · 2026-06-20

**Recommendation:** Reject

**Audience:**

Yes

**Audience Explanation:**

This submission has some interesting contributions in optimization community.

**Claims And Evidence:**

No

**Claims Explanation:**

This paper presents a theoretical analysis of the Muon optimizer with smoothness and bounded variance assumptions. That is, the main contribution of this submission is its theoretical results. However, the main contribution regarding bound tightness was wrong, and the authors acknowledge that the theoretical claim can be revised as follows: The asymptotic convergence rate in Theorems 3.1 and 3.2 is almost the same as in the four variants of Muon.

As suggested by the Reviewers, I am recommending rejection.